# Adapting to Continuous Covariate Shift via Online Density Ratio Estimation

**Yu-Jie Zhang**[1], **Zhen-Yu Zhang**[2], **Peng Zhao**[3], **Masashi Sugiyama**[2,1]

[1] The University of Tokyo, Chiba, Japan
[2] RIKEN AIP, Tokyo, Japan
[3] National Key Laboratory for Novel Software Technology, Nanjing University, Nanjing, China

## Abstract

Dealing with distribution shifts is one of the central challenges for modern machine learning. One fundamental situation is the *covariate shift*, where the input distributions of data change from the training to testing stages while the input-conditional output distribution remains unchanged. In this paper, we initiate the study of a more challenging scenario — *continuous* covariate shift — in which the test data appear sequentially, and their distributions can shift continuously. Our goal is to adaptively train the predictor such that its prediction risk accumulated over time can be minimized. Starting with the importance-weighted learning, we theoretically show the method works effectively if the time-varying density ratios of test and train inputs can be accurately estimated. However, existing density ratio estimation methods would fail due to data scarcity at each time step. To this end, we propose an online density ratio estimation method that can appropriately reuse historical information. Our method is proven to perform well by enjoying a dynamic regret bound, which finally leads to an excess risk guarantee for the predictor. Empirical results also validate the effectiveness.

## 1 Introduction

How to deal with distribution shifts is one of the central challenges for modern machine learning [1, 2], which is also the key requirement of robust artificial intelligence in open and dynamic environments [3, 4]. *Covariate shift* is a representative problem setup and has attracted much attention [5, 6], where the input density $\mathcal{D}(\mathbf{x})$ varies from training data to test data but their input-conditional output $\mathcal{D}(y \mid \mathbf{x})$ remains unchanged. On the one hand, covariate shift serves as a foundational setup for theoretically understanding general distribution shifts [7], and on the other hand, it can encompass many real-world tasks such as brain-computer interface [8], speaker identification [9] and audio processing [10].

Existing works of coping with covariate shift mainly focused on the "one-step" adaptation, where the learner aims to train a model well-performed on a *fixed* testing distribution. To mitigate the distribution discrepancy, a common and classic solution is the importance-weighting framework [6], where one assigns an appropriate importance weight to each labeled training sample and then conducts weighted empirical risk minimization. The importance weight, also known as the density ratio of test and training inputs, usually needs to be estimated via a reasonable amount of unlabeled data sampled from the testing distribution [11, 12]. However, the one-step adaptation can be insufficient in many real-world covariate shift scenarios, especially when data are accumulated in an online fashion such that testing environments *continuously shift* and only *a few unlabeled samples* are observed at each time. For instance, in the speaker identification task, the speech features vary over time due to session

---

*Correspondence: Yu-Jie Zhang <yujie.zhang@ms.k.u-tokyo.ac.jp> and Peng Zhao <zhaop@lamda.nju.edu.cn>

dependent variation, the recording environment change, and physical conditions and emotions of the speakers [9]. Therefore, it is necessary to perform a prompt adaptation to the changes.

Motivated by such demands, we initiate the study of *continuous covariate shift*, where the learner's goal is to adaptively update the model to minimize the risk accumulated over time (see the formal definition in Section 3). To achieve this goal, we propose the ACCOUS approach (Adapt to Continuous Covariate shift with Unlabeled Stream) equipped with sound theoretical guarantees. Our approach is based on the classic importance-weighting framework yet requires innovations to make it applicable to the continuous shift scenario. Indeed, we theoretically identify that the importance-weighting works effectively if the cumulative estimation error of the time-varying density ratios of train and test inputs can be reduced. However, at each time step, a direct application of one-step density ratio estimation would lead to a high variance due to data scarcity; whereas reusing all previous data can be highly biased when testing distributions change dramatically. Thus, it is crucial to design an accurate estimation of *time-varying* test-train density ratios by appropriately reusing historical information.

To combat the difficulty, we propose a generic reduction of the time-varying density ratio estimation problem to the online convex optimization [13]: one can immediately obtain high-quality time-varying density ratios by a suitable online process to optimize its *dynamic regret* over a certain sequence of loss functions. Our reduction is based on the Bregman divergence density ratio matching framework [14] and applicable to various existing density ratio estimation models with specific configurations of the divergence functions. To minimize the dynamic regret, our key algorithmic ingredient of this online optimization is the *online ensemble* structure [15, 16], where a group of base-learners is maintained to perform density ratio estimation with different lengths of historical data, and a meta-algorithm is employed to combine the predictions. As such, we can properly reuse historical information without knowing the cumulative intensity of the covariate shift. We further instantiate the reduction framework with the logistic regression model [17]. Letting $V_T = \sum_{t=2}^{T} \|\mathcal{D}_t(\mathbf{x}) - \mathcal{D}_{t-1}(\mathbf{x})\|_1$, we prove an $\widetilde{\mathcal{O}}(T^{1/3}V_T^{2/3})$ dynamic regret bound for the density ratio estimator, which finally leads to an $\widetilde{\mathcal{O}}(T^{-1/3}V_T^{1/3})$ averaged excess risk of the predictor trained by importance-weighted learning. The rate can hardly be improved even if one receives labels of the testing stream after prediction (see more elaborations below Theorem 3 and Appendix D.6). Finally, we conduct experiments to evaluate our approach, and the empirical results validate the theoretical findings.

**Technical Contributions.** We note that online ensemble was also employed by [18] to handle continuous *label shift*, another typical distribution shift assuming the change happens on the class prior $\mathcal{D}_t(y)$. However, their method crucially relies on the *unbiasedness* of weights in the risk estimator, whereas density ratio estimators for covariate shift adaptation cannot satisfy the condition in general. Instead of pursuing unbiasedness, we delicately design an online optimization process to learn the density ratio estimator, which is versatile to be implemented with various models. Our methodology is very general and might be of broader use, such as relaxing the unbiasedness requirement in the continuous label shift. Besides, the $\widetilde{\mathcal{O}}(T^{1/3}V_T^{2/3})$ dynamic regret bound for the logistic regression density ratio estimator is obtained non-trivially. Our bound essentially holds for online learning with exp-concave functions under noisy feedback. The only one achieving this [19] holds in expectation and requires *complicated* analysis on the Karush-Kuhn-Tucker (KKT) condition [20] of comparators. By contrast, our bound holds in high probability, and the analysis is greatly *simplified* by virtue of exploiting the structure of comparators in our problem that they are essentially the minimizers of expected functions (hence without analyzing the KKT condition).

## 2 Preliminaries

This section introduces the preliminaries and related work on the continuous covariate shift adaptation. We discuss the related work on non-stationary online learning in Appendix B

**One-Step Covariate Shift Adaptation.** Let $\mathcal{D}_0(\mathbf{x}, y)$ and $\mathcal{D}_1(\mathbf{x}, y)$ be the training and test distributions. The "one-step" adaptation problem studies how to minimize the testing risk $R_1(\mathbf{w}) = \mathbb{E}_{(\mathbf{x},y)\sim\mathcal{D}_1(\mathbf{x},y)}[\ell(\mathbf{w}^\top\mathbf{x}, y)]$ by the model trained with a labeled dataset $S_0 = \{\mathbf{x}_i, y_i\}_{i=1}^{N_0}$ sampled from $\mathcal{D}_0(\mathbf{x}, y)$ and unlabeled dataset $S_1 = \{\mathbf{x}_i\}_{i=1}^{N_1}$ sampled from $\mathcal{D}_1(\mathbf{x})$. We call this one-step adaptation since the test distribution is *fixed* and a reasonable number of unlabeled data $S_1$ is available.

**Importance-Weighted ERM.** A classic solution for the one-step covariate shift adaptation is the importance-weighted empirical risk minimization (IWERM), which mitigates the distribution shift

by minimizing the weighted empirical risk $\widetilde{R}_1(\mathbf{w}) = \mathbb{E}_{(\mathbf{x},y)\sim S_0}[r_1^*(\mathbf{x})\ell(\mathbf{w}^\top\mathbf{x}, y)]$, where $r_1^*(\mathbf{x}) = \mathcal{D}_1(\mathbf{x})/\mathcal{D}_0(\mathbf{x})$ is the importance weight. The risk $\widetilde{R}_1(\mathbf{w})$ is unbiased to $R_1(\mathbf{w})$ and thus the learned model is consistent over the test distribution [5]. The importance weighted learning was studied from the lens of variance-bias trade off [5, 21], cross validation [22], model misspecification [23], and deep neural network implementation [24]. All those are conducted under the one-step adaptation scenario.

**Density Ratio Estimation.** The importance weighted estimation, also known as the density ratio estimation (DRE) [12], aims to estimate the density ratio $r_1^*(\mathbf{x})$ using datasets input of $S_0$ and $S_1$. Various methods were proposed with different statistical models [14, 17, 25, 26, 27, 28, 29]. All those methods focused on the one-step adaptation, and it is challenging to extend them to the continuous shift due to the limited unlabeled data at each time step. The work [30] studied how to update the density ratio estimator with streaming data, but the ground-truth density ratio is assumed to be fixed. Recently, the time-varying density ratio was investigated in [31]. Their problem setup fundamentally differs from ours as they assume sufficient online data at each iteration, while our challenge is dealing with limited data per iteration. The work [32] proposed to learn the time-varying density ratios using all past data at each time. Although learning theory insights are provided in the paper, it is still unclear how the learned density ratios balance sample complexity with environmental shift intensity. In contrast, our density ratio estimator updates in an online fashion with dynamic regret guarantees.

**Continuous Distribution Shift.** For broader continuous distribution shift problems, the study [33] focused on the label shift case and provided the first feasible solution. Then, the work [18] introduced modern non-stationary online learning techniques to the problem, developing the first method with dynamic regret guarantees. As shown in Remark 2, our solution refines the previous methods [33, 18] by decoupling density ratio estimation from predictor training, paving the way for a theoretically-grounded method for continuous covariate shift. Similarly, the subsequent work [34] developed a method for continuous label shift, which allows for training the predictor with various models by separately estimating the label probability through an online regression oracle. Another research [35] studied the *change detection* for continuous covariate shift. However, they only focused on identifying differences between the current and initial offline distributions, and it is still unclear how to update the model adaptively with the online data. Moreover, the method [35] can only detect at a given time granularity, while ours can perform the model update at each round.

## 3 Adapting to Continuous Covariate Shift

In this section, we formulate the problem setup of continuous covariate shift and then introduce our approach based on the IWERM framework and online density ratio estimation.

### 3.1 Problem Setup

There are two stages in continuous covariate shift. The first one is the *offline initialization stage*, where the learner can collect a reasonable number of label data $S_0 = \{\mathbf{x}_n, y_n\}_{n=1}^{N_0}$ from the initial distribution $\mathcal{D}_0$. Then, we come to the *online adaptation stage*, where the unlabeled testing data arrive sequentially, and the underlying distributions can continuously shift. Consider a $T$-round online adaptation. At each round $t \in [T] \triangleq \{1, \ldots, T\}$, the learner will receive an unlabeled dataset $S_t = \{\mathbf{x}_n\}_{n=1}^{N_t}$ sampled from the underlying distribution $\mathcal{D}_t$. Without loss of generality, we consider $N_t = 1$. We have the following continuous covariate shift condition.

**Assumption 1** (Continuous Covariate Shift)**.** For all $\mathbf{x} \in \mathcal{X}$ in the feature space, $y \in \mathcal{Y}$ in the label space, and any $t \in [T]$, we have

$$\mathcal{D}_t(y \mid \mathbf{x}) = \mathcal{D}_0(y \mid x) \text{ and } r_t^*(\mathbf{x}) = \mathcal{D}_t(\mathbf{x})/\mathcal{D}_0(\mathbf{x}) \leq B < \infty.$$

We note that there are emerging discussions on the necessity of covariate shift adaptation [7, 23]. When there are infinite number of training samples, a well-specified large and over-parametrized model can be trained to approximate $\mathcal{D}_0(y \mid \mathbf{x})$ and there is no need to perform covariate shift adaptation. However, in the finite-sample cases, one would prefer to train a model with constraint complexity to ensure its generalization ability, which leads to model misspecification and covariate shift adaptation is indeed necessary. In this paper, we study the continuous covariate shift under a misspecified hypothesis space $\mathcal{W}$, which generalizes the standard one-step covariate shift problem (see [6] and reference therein) to the continuous shift case. Our goal is to train a sequence of model

$\{\mathbf{w}_t\}_{t=1}^T$ that are comparable with the best model $\mathbf{w}_t^* \in \arg\min_{\mathbf{w}\in\mathcal{W}} R_t(\mathbf{w})$ in the hypothesis space at each time. Consequently, we take the following average excess risk as the performance measure:

$$\mathfrak{R}_T(\{\widehat{\mathbf{w}}_t\}_{t=1}^T) \triangleq \frac{1}{T}\left(\sum_{t=1}^T R_t(\widehat{\mathbf{w}}_t) - \sum_{t=1}^T R_t(\mathbf{w}_t^*)\right). \tag{1}$$

We end this part by listing several common notations used throughout the paper. We denote by $R = \max_{\mathbf{x}\in\mathcal{X}}\|\mathbf{x}\|_2$ the maximum norm of the input and by $D = \max_{\mathbf{w}_1,\mathbf{w}_2\in\mathcal{W}}\|\mathbf{w}_1 - \mathbf{w}_2\|_2$ the diameter of the hypothesis space $\mathcal{W}\subseteq\mathbb{R}^d$. The constant $G = \max_{\mathbf{x}\in\mathcal{X},y\in\mathcal{Y},\mathbf{w}\in\mathcal{W}}\|\nabla\ell(\mathbf{w}^\top\mathbf{x},y)\|_2$ is the maximum gradient norm and $L = \max_{\mathbf{x}\in\mathcal{X},y\in\mathcal{Y},\mathbf{w}\in\mathcal{W}}|\ell(\mathbf{w}^\top\mathbf{x},y)|$ is the upper bound of the loss values. We use the $\widetilde{\mathcal{O}}(\cdot)$-notation to hide dependence on logarithmic factors of $T$.

## 3.2 Importance Weighted ERM for Continuous Shift

Our algorithm is based on the importance weighted learning. Consider the scenario where the learner has a predefined $\widehat{r}_t : \mathcal{X} \to [0, B]$ as an estimation for the true density ratio $r_t^*(\mathbf{x}) = \mathcal{D}_t(\mathbf{x})/\mathcal{D}_0(\mathbf{x})$ for each time $t \in [T]$. Then, the predictor can be trained by the importance weighted ERM method:

$$\widehat{\mathbf{w}}_t = \arg\min_{\mathbf{w}\in\mathcal{W}} \mathbb{E}_{\mathbf{x}\sim S_0}[\widehat{r}_t(\mathbf{x})\ell(\mathbf{w}^\top\mathbf{x},y)]. \tag{2}$$

The averaged excess risk of the predictor is closely related to the quality of the density ratio estimator.

**Proposition 1.** *For any $\delta \in (0, 1]$, with probability at least $1 - \delta$, IWERM (2) with the estimator $\widehat{r}_t(\mathbf{x})$ ensures $\mathfrak{R}_T(\{\widehat{\mathbf{w}}_t\}_{t=1}^T) \leq 2\sum_{t=1}^T \mathbb{E}_{\mathbf{x}\sim S_0}\left[|\widehat{r}_t(\mathbf{x}) - r_t^*(\mathbf{x})|\right]/T + \mathcal{O}(\log(T/\delta)/\sqrt{N_0})$.*

In above, the $\mathcal{O}(\log T/\sqrt{N_0})$ term measures the *generalization gap* of the predictor since it is trained over the empirical data $S_0$ instead of $\mathcal{D}_0$. Such a rate is tight up to a logarithmic factor in $T$. Indeed, considering a stationary environment (i.e., $\mathcal{D}_t(\mathbf{x}, y) = \mathcal{D}_1(\mathbf{x}, y)$) and an exact density ratio estimation (i.e., $\widehat{r}_t(\mathbf{x}) = r_t^*(\mathbf{x})$), Proposition 1 indicates that the averaged model $\overline{\mathbf{w}}_T = \sum_{t=1}^T \widehat{\mathbf{w}}_t/T$ enjoys an excess risk bound of $R_1(\overline{\mathbf{w}}_T) - R_1(\mathbf{w}_1^*) \leq \mathcal{O}(\log T/\sqrt{N_0})$, matching the lower bound for importance weighted learning [36, Proposition 2] up to an $\mathcal{O}(\log T)$ factor.

Our main focus lies on the $\sum_{t=1}^T \mathbb{E}_{\mathbf{x}\sim S_0}[|\widehat{r}_t(\mathbf{x}) - r_t^*(\mathbf{x})|]/T$ term, which is the averaged estimation error of the density ratio estimator $\widehat{r}_t$ to the time-varying ground truth $r_t^*$. To minimize the estimation error, a fundamental challenge comes from the *unknown non-stationarity* exhibited in the online environments — how to select a right amount of historical data to reuse for each iteration? Intuitively, if the environments change slowly, it would be preferable to reuse all historical data to construct the density ratio estimator $\widehat{r}_t$. However, when distribution shifts occur frequently, earlier data could be useless even potentially harmful. In such scenarios, training the density ratio estimator with the most recent data would be a more rational strategy. Furthermore, even if we could capture a roughly stationary period, it remains unclear how to update the density ratio estimator in an online manner with guarantees. To our knowledge, online density ratio estimation is underexplored in the literature, even for the stationary testing streaming, not to mention the more challenging non-stationary setup.

## 3.3 Online Density Ratio Estimation

Here, we present a generic reduction of the online density ratio estimation problem to a *dynamic regret minimization* problem. This novel perspective paves our way for designing an algorithm in tackling the non-stationarity of environments, as mentioned in Section 3.2.

**Bregman Divergence Density Ratio Matching.** Our reduction is based on the Bregman divergence density ratio matching [14], a general framework that unifies various existing DRE methods. Specifically, in the framework, the discrepancy between the ground-truth density ratio function $r_t^*$ and any density ratio function $r$ is measured by the expected Bregman divergence over $\mathcal{D}_0(\mathbf{x})$ defined by

$$\mathrm{EB}_\psi(r_t^*\|r) = \mathbb{E}_{\mathbf{x}\sim\mathcal{D}_0(\mathbf{x})}\left[\mathcal{B}_\psi\left(r_t^*(\mathbf{x})\|r(\mathbf{x})\right)\right], \tag{3}$$

where $\mathcal{B}_\psi(a\|b) \triangleq \psi(a) - \psi(b) - \partial\psi(b)(a - b)$ is the Bregman divergence and $\psi : \mathrm{dom}\,\psi \to \mathbb{R}$ is the associated divergence function. The expected Bregman divergence can be equally rewritten as $\mathrm{EB}_\psi(r_t^*\|r) = L_t^\psi(r) - L_t^\psi(r_t^*)$, where $L_t^\psi$ is the loss for the density ratio function defined by

$$L_t^\psi(r) = \mathbb{E}_{\mathbf{x}\sim\mathcal{D}_0(\mathbf{x})}\left[\partial\psi(r(\mathbf{x}))r(\mathbf{x}) - \psi(r(\mathbf{x}))\right] - \mathbb{E}_{\mathbf{x}\sim\mathcal{D}_t(\mathbf{x})}\left[\partial\psi(r(\mathbf{x}))\right]. \tag{4}$$

As a consequence, one can train the density ratio estimator by minimizing the loss $L_t^\psi(r)$. We have $r_t^* \in \arg\min_{r \in \mathcal{H}} L_t^\psi(r)$ when $\mathcal{H}$ is the set of all measurable functions.

The Bregman divergence-based density ratio matching framework takes various DRE methods as special cases. For instance, choosing the divergence function as $\psi_{\mathsf{LS}}(t) = (t-1)^2/2$ yields LSIF [28] and KMM [26]; choosing $\psi_{\mathsf{LR}}(t) = t \log t - (t+1) \log(t+1)$ leads to the logistic regression method [17]; and one can also recover the UKL [37] and KLLEP [25] with $\psi_{\mathsf{KL}}(t) = t \log t - t$.

**Online DRE via Dynamic Regret Minimization.** For a single-round density ratio estimation, the Bregman divergence-based framework suggests to train the density ratio estimator $\widehat{r}_t$ by minimizing the gap $L_t^\psi(\widehat{r}_t) - L_t^\psi(r_t^*)$. Therefore, to derive an estimator sequence that performs well over time, we utilize the cumulative loss gap $\sum_{t=1}^T L_t^\psi(\widehat{r}_t) - \sum_{t=1}^T L_t^\psi(r_t^*)$ as a performance measure. Indeed, the cumulative loss gap serves as an upper bound of the estimation error of density ratio estimators.

**Proposition 2.** *Let $\psi$ be a $\mu$-strongly convex function. For any density ratio estimator sequence $\{\widehat{r}_t\}_{t=1}^T$, we have $\frac{1}{T}\sum_{t=1}^T \mathbb{E}_{\mathbf{x} \sim \mathcal{D}_0}[|\widehat{r}_t(\mathbf{x}) - r_t^*(\mathbf{x})|] \le \sqrt{2\big(\sum_{t=1}^T L_t^\psi(\widehat{r}_t) - \sum_{t=1}^T L_t^\psi(r_t^*)\big)/(\mu T)}$.*

Proposition 2 indicates that it suffices to optimize the cumulative loss gap to perform online density ratio estimation. Such a measure quantifies the performance difference between the online algorithm (that yields the estimated ratio sequence $\{\widehat{r}_t\}_{t=1}^T$) and a sequence of *time-varying* comparators (the true ratio sequence $\{r_t^*\}_{t=1}^T$). This is exactly the *dynamic regret* in online learning literature [38, 39]. As such, we have reduced the online density ratio estimation to a problem of dynamic regret minimization over loss functions $\{L_t^\psi\}_{t=1}^T$ against comparators $\{r_t^*\}_{t=1}^T$.

Since the loss function involves the expectation over the underlying distribution $\mathcal{D}_0(\mathbf{x})$ (see the definition in (4)), we need a counterpart result with respect to empirical data $S_0$.

**Theorem 1.** *Let $\psi$ be a $\mu$-strongly convex function satisfying $t\partial^3\psi(t) \le 0$ and $\partial^3\psi(t) \le 0$ for all $t \in \operatorname{dom} \psi$. Let $\mathcal{H}_\theta = \{\mathbf{x} \mapsto h(\mathbf{x}, \theta) \mid \theta \in \Theta\}$ be a hypothesis space of density ratio functions parameterized by a finite-dimensional bounded set $\Theta \triangleq \{\theta \in \mathbb{R}^d \mid \|\theta\|_2 \le S\}$ with a certain link function $h : \mathcal{X} \times \Theta \mapsto \mathbb{R}$. Denote by $[z]_+ \triangleq \max\{z, 0\}$. Then, for any density ratio estimator $\widehat{r}_t \in \mathcal{H}_\theta$, the empirical estimation error is bounded by*

$$\frac{1}{T}\sum_{t=1}^T \mathbb{E}_{\mathbf{x} \sim S_0(\mathbf{x})}\big[|r_t^*(\mathbf{x}) - \widehat{r}_t(\mathbf{x})|\big] \le \sqrt{\frac{4}{\mu T}\left[\sum_{t=1}^T \widetilde{L}_t^\psi(\widehat{r}_t) - \sum_{t=1}^T \widetilde{L}_t^\psi(r_t^*)\right]_+ + \mathcal{O}\left(\frac{\sqrt{d}\log(T/\delta)}{\mu\sqrt{N_0}}\right)},$$

*provided that $h(\mathbf{x}, \theta)$ is bounded for any $\mathbf{x} \in \mathcal{X}$ and $\theta \in \Theta$ and Lipschitz continuous. In the above,*

$$\widetilde{L}_t^\psi(r) = \mathbb{E}_{\mathbf{x} \sim S_0}\left[\partial\psi(r(\mathbf{x}))r(\mathbf{x}) - \psi(r(\mathbf{x}))\right] - \mathbb{E}_{\mathbf{x} \sim \mathcal{D}_t(\mathbf{x})}\left[\partial\psi(r(\mathbf{x}))\right] \tag{5}$$

*is the empirical approximation of the expected loss $L_t^\psi$ using the data $S_0$.*

**Remark 1** (assumptions on $\psi$)**.** We imposed certain assumptions on the divergence function $\psi$ in Theorem 1. One can check these conditions hold for commonly used $\psi$ in density ratio estimation, including *all* the divergence functions mentioned earlier ($\psi_{\mathsf{LR}}$ and $\psi_{\mathsf{KL}}$ are strongly convex when the inputs are upper bounded, which can be satisfied with suitable choices of $\mathcal{H}_\theta$). Section 4 will present an example with $\psi_{\mathsf{LR}}$ to show how the conditions are satisfied. ◁

Theorem 1 shows that we can immediately obtain a sequence of high-quality density ratio estimators $\{r_t^*\}_{t=1}^T$ by minimizing the dynamic regret with respect to $\{\widetilde{L}_t^\psi\}_{t=1}^T$,

$$\mathbf{Reg}_T^{\mathbf{d}}(\{\widetilde{L}_t^\psi, r_t^*\}_{t=1}^T) = \sum_{t=1}^T \widetilde{L}_t^\psi(\widehat{r}_t) - \sum_{t=1}^T \widetilde{L}_t^\psi(r_t^*). \tag{6}$$

One caveat is that the second term of $\widetilde{L}_t^\psi$ in the definition (5) requires the knowledge of underlying distribution $\mathcal{D}_t$, which is unavailable. Empirically, we can only observe $\widehat{L}_t^\psi$ defined below, building upon the empirical observations $S_t \sim \mathcal{D}_t$,

$$\widehat{L}_t^\psi(r) = \mathbb{E}_{\mathbf{x} \sim S_0}\left[\partial\psi(r(\mathbf{x}))r(\mathbf{x}) - \psi(r(\mathbf{x}))\right] - \mathbb{E}_{\mathbf{x} \sim S_t}\left[\partial\psi(r(\mathbf{x}))\right]. \tag{7}$$

That said, we need to design an online optimization process to minimize the dynamic regret (6) defined over $\widetilde{L}_t^\psi$ based on the observed loss $\{\widehat{L}_t^\psi\}_{t=1}^T$. Since the time-varying comparator $r_t^*$ in (6)

is *not* the minimizer of the observed loss $\widehat{L}_t^\psi$ (but rather the minimizer of the expected loss $L_t^\psi$ defined in (4)), directly minimizing the empirical loss $\widehat{L}_t^\psi$ will lead to a high estimation error. In the next section, we introduce how to optimize the dynamic regret with the *online ensemble* framework developed in recent studies of non-stationary online convex optimization [15, 16, 19].

**Remark 2** (comparison with previous work). For the continuous label shift [18], the online ensemble framework was employed to train the predictor $\mathbf{w}_t$. However, the previous method crucially relies on the construction of an unbiased importance ratio estimator satisfying $\mathbb{E}_{\mathcal{D}_t}[\widehat{r}_t(\mathbf{x})] = r_t^*(\mathbf{x})$. Such a favorable property is hard to be satisfied in the covariate shift case. For example, in the Bregman divergence density ratio matching framework, one can only observe an unbiased loss $\widehat{L}_t^\psi(r)$, whose minimizer $\widehat{r}_t = \arg\min_r \widehat{L}_t^\psi(r)$ is not unbiased to the true value $r_t^*$ in general. To this end, we disentangle the model training and importance weight estimation process in this paper. Our reduction holds for the general Bergman divergence matching framework and thus can be initiated with different models (not necessarily unbiased). It is possible to extend our framework to continuous label shift problem with other importance weight estimators besides the unbiased one [40] used in [18]. ◁

## 4 Instantiation: Logistic Regression Model

When the loss function $\widehat{L}_t^\psi$ is non-convex, it is generally intractable to conduct the online optimization, regardless of minimizing the standard regret or the strengthened dynamic regret. Fortunately, the attained loss functions are convex or enjoy even stronger curvature with the properly chosen hypothesis space and divergence function. In this section, we instantiate our framework with the logistic regression model. A corresponding online DRE method is presented with dynamic regret guarantees.

**Example 1** (Logistic regression model). Consider the function $\psi = \psi_{\mathsf{LR}} \triangleq t\log t - (t+1)\log(t+1)$ and the hypothesis space $H_\theta^{\mathsf{LR}} = \{\mathbf{x} \mapsto \exp\left(-\theta^\top \phi(\mathbf{x})\right) \mid \|\theta\|_2 \le S\}$. Here, $\phi : \mathcal{X} \mapsto \mathbb{R}^d$ represents a specific basis function with bounded norm $\|\phi(\mathbf{x})\|_2 \le R$, which could be, for instance, the feature representation extractor from a deep neural network. Then, the loss function $\widehat{L}_t^\psi(\theta)$ as per (7) becomes

$$\widehat{L}_t^\psi(\theta) = \frac{1}{2}\Big(\mathbb{E}_{S_0}[\log(1 + e^{-\phi(\mathbf{x})^\top \theta})] + \mathbb{E}_{S_t}[\log(1 + e^{\phi(\mathbf{x})^\top \theta})]\Big).$$

Let $\beta = \exp(SR)$. Then, the output of any density ratio function $r \in \mathcal{H}_\theta$ is bounded by $[1/\beta, \beta]$. It can be validated that $\psi_{\mathsf{LR}}$ is $1/(\beta + \beta^2)$-strongly convex and satisfies the condition $\partial^3 \psi_{\mathsf{LR}}(z) \le 0$ required by Theorem 1. Besides, the logistic loss is a $1/(1(1+\beta))$-*expconcave function* and $R^2/2$-*smooth function*, presenting a favorable function properties for online convex optimization [13].

We note that the other two divergence functions $\psi_{\mathsf{LS}}$ and $\psi_{\mathsf{KL}}$ also exhibit desirable function properties as $\psi_{\mathsf{LR}}$. When choosing the hypothesis space $H_\theta = \{\mathbf{x} \mapsto \theta^\top \phi(\mathbf{x}) \mid \theta \in \Theta\}$, the methods recover the the uLISF [28] and UKL [37] density ratio estimators equipped with the generalized linear model. Our analysis is also applicable in the two cases. More details are provided in Appendix D.7.

### 4.1 Density Ratio Estimation via Online Ensemble

In this part, we introduce our online ensemble method for online DRE with the logistic regression model. Since the logistic regression loss is exp-concave, [19] shows that one can employ the following-the-leading-history (FLH) method [41] to minimize the dynamic regret. However, a caveat is that the observed loss $\widehat{L}_t$ is only an empirical estimation of $\widetilde{L}_t$ established on few online data $S_t$. The result of [19] only implies an *expected* bound for our problem.

We twisted the FLH algorithm to achieve a high probability bound. As shown in Figure 1, our algorithm maintains multiple base-learners, each learning over different intervals of the time-horizon, and then employs a meta-learner to aggregate their predictions. We modify the meta-learner in FLH from Hedge [42] to Adapt-ML-Prod [43], which ensures that the meta-learner can track each base-learner on the associated time interval with high probability. This strategy allows us to selectively reuse the historical information to handle the non-stationary environments. We introduce ingredients of our algorithms as follows. A more detailed algorithm descriptions and comparison with the dynamic regret minimization literature in OCO can be found in Appendices D.1 and B respectively.

**Base-learner.** Our algorithm runs multiple base-learners that are active on different intervals of the time horizon. Let $\mathcal{C} = \{I_i = [s_i, e_i] \subseteq [T]\}$ be a interval set. Each base-learner $\mathcal{E}_i$ will only

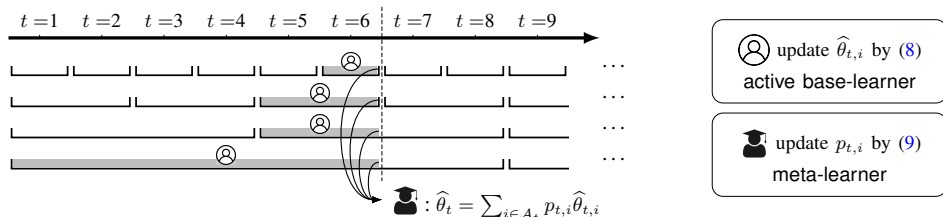

**Figure 1:** An illustration of our online ensemble method, where we employ a meta-learner to aggregate the predictions from base-learners running over different intervals of the time horizon.

update and submit the model $\widehat{\theta}_{t,i}$ to the meta-learner if $t \in I_i$. We use the online Newton step (ONS) for the model update. For an exp-concave loss function, the method ensures the base-learner's model is comparable with the best fixed model over the interval up to a logarithmic factor in $T$ [44]. Specifically, for each base-learner $\mathcal{E}_i$ running on the interval $[s_i, e_i]$, ONS updates the model by

$$\widehat{\theta}_{t+1,i} = \Pi_{\Theta}^{A_{t,i}}\left[\widehat{\theta}_{t,i} - \gamma A_{t,i}^{-1}\nabla\widehat{L}_t(\widehat{\theta}_{t,i})\right], \ \forall t \in [s_i, e_i] \,, \tag{8}$$

where the matrix $A_{t,i} = \lambda I + \sum_{\tau=s_i}^{t}\nabla\widehat{L}_\tau(\widehat{\theta}_{\tau,i})\nabla\widehat{L}_s(\widehat{\theta}_{\tau,i})^\top$. In the above, the projection function is defined as $\Pi_{\Theta}^{A_{t,i}}[\theta_1] = \arg\min_{\theta\in\Theta}\|\theta - \theta_1\|_{A_{t,i}}$ and $\Theta = \{\theta \in \mathbb{R}^d \mid \|\theta\|_2 \leq S\}$ is the parameter space. The constant $\lambda > 0$ and $\eta > 0$ are the regularizer parameter and step size to be specified latter.

**Meta-learner.** As shown by Figure 1, the meta-learner's role is to aggregate the models produced by the base-learners using a weighting scheme. At each iteration $t$, the meta-learner will maintain a weight $p_{t,i}$ for each "active" base-learner $\mathcal{E}_i$, defined as the one whose associated interval contains $t$. We update the weights for active base-learners based on the Adapt-ML-Prod method [43]. Specifically, for every active base-learner $\mathcal{E}_i$, our algorithm maintains a "potential" $v_{t,i} \in \mathbb{R}_+$ at iteration $t$, which reflects the historical performance of the base-leaner until time $t$. Then, denoting by $\mathcal{A}_t$ the index set of the active base-learners at time $t$, their weights and the output model are obtained by

$$p_{t,i} \propto \varepsilon_{t-1,i}v_{t-1,i} \text{ for all } i \in \mathcal{A}_t \quad\text{and}\quad \widehat{\theta}_{t+1} = \sum_{i\in\mathcal{A}_{t+1}} p_{t+1,i}\widehat{\theta}_{t+1,i} \,. \tag{9}$$

In the above $\varepsilon_{t,i} > 0$ is a step size that can be automatically tuned along the learning process. A noteworthy ingredient of our algorithm is the construction of the potential $v_{t,i}$, using the linearized loss $\langle\nabla\widehat{L}_t(\widehat{\theta}_t), \widehat{\theta}_t - \widehat{\theta}_{t,i}\rangle/(SR)$. Then, the generalization gap between $\widehat{L}_t$ and $\widetilde{L}_t$ can be controlled with the negative term introduced by the exp-concave loss function. As a result, we can establish a high probability bound to ensure that $\widehat{\theta}_t$ is competitive with any base-learner's model $\widehat{\theta}_{t,i}$ on the corresponding interval $I_i$. The detailed configurations of $v_{t,i}$ and $\varepsilon_{t,i}$ are deferred to Appendix D.1.

**Schedule of Intervals.** As shown in Figure 1, we specify the intervals with the geometric covering scheme [41], where the interval set is defined as $\mathcal{C} = \bigcup_{k\in\mathbb{N}\cup\{0\}}\mathcal{C}_k$ with $\mathcal{C}_k = \{[i \cdot 2^k, (i+1) \cdot 2^k - 1] \mid i \in \mathbb{N} \text{ and } i \cdot 2^k \leq T\}$. One can check that $|\mathcal{C}|$ is at most $T$, and the number of active base-learner is bounded as $|\mathcal{A}_t| \leq \lceil\log t\rceil$. Thus, we only need to maintain at most $\mathcal{O}(\log t)$ base-learners at time $t$. The intervals specified by the geometric covering are informative to capture the non-stationarity of the environments, leading to the following dynamic regret guarantees.

### 4.2 Theoretical Guarantees

This part presents the theoretical guarantees of our method. The estimator $\widehat{r}_t(\mathbf{x}) = \exp(-\phi(\mathbf{x})^\top\widehat{\theta}_t)$ established on $\widehat{\theta}_t$ returned by the online ensemble method (9) achieves the dynamic regret guarantee.

**Theorem 2.** *Assume the true density ratio $r_t^*(\mathbf{x}) = \mathcal{D}_t(\mathbf{x})/\mathcal{D}_0(\mathbf{x})$ is contained in the hypothesis space as $r_t^* \in H_\theta^{\mathsf{LR}} \triangleq \{\mathbf{x} \mapsto \exp(-\phi(\mathbf{x})^\top\theta) \mid \theta \in \Theta\}$ for any $t \in [T]$. Then, with probability at least $1 - \delta$, the dynamic regret of the density ratio estimator $\widehat{r}_t(\mathbf{x}) = \exp(-\phi(\mathbf{x})^\top\widehat{\theta}_t)$ is bounded by*

$$\mathbf{Reg}_T^{\mathbf{d}}(\{\widetilde{L}_t, r_t^*\}_{t=1}^T) \leq \widetilde{\mathcal{O}}\left(\max\{T^{\frac{1}{3}}V_T^{\frac{2}{3}}, 1\} + T/N_0\right),$$

*when the parameters are set as $\gamma = 3(1 + \beta)$ and $\lambda = 1$. In the above, $V_T = \sum_{t=2}^T\|\mathcal{D}_t(\mathbf{x}) - \mathcal{D}_{t-1}(\mathbf{x})\|_1$ measures the variation of input densities.*

Theorem 2 imposes a realizable assumption for the true density ratio function such that $r_t^* \in H_\theta^{\mathsf{LR}}$. Such an assumption is required since we train the density ratio estimator on a given hypothesis space while the true density ratio function could be arbitrary. We note that the realizability assumption is frequently used in the analysis for the density ratio estimation problem [29, 45] and other related topics, including active learning [46] and contextual bandit [47], where the density ratio (or density) estimation is required. A possible direction to relax the assumption is to consider a richer function class, e.g., a neural network or a non-parametric model. We leave this as a future work.

The $\widetilde{\mathcal{O}}(T^{\frac{1}{3}} V_T^{\frac{2}{3}})$ rate in our bound exhibits the same rate as the minimax optimal dynamic regret bound for the squared loss function with the noisy feedback [48]. Since the squared loss function enjoys even stronger curvature than the logistic loss, our result is hard to be improved. To achieve this fast-rate result, the key was to show a "squared" formulation $\mathcal{O}\big(\max\{1, |I| V_I^2\}\big)$ of the dynamic regret for the base-leaner on each interval $I$ (Lemma 3 in Appendix D.2). [19] achieved this by a complicated analysis with the KKT condition to capture the structure of the comparators, while we greatly simplified the analysis by exploiting the structure that the comparator $\theta_t^*$ is essentially the minimizers of *expected* functions $L_t$ (hence avoiding analyzing the KKT condition). Our analysis is applicable to the case where the minimiers lie in the interior of the decision set and only requires the smoothness of the loss function, which can be of independent interest in online convex optimization.

**Averaged Excess Risk Bound for Continuous Covariate Shift.** After obtaining the density ratio estimator, we can train the predictive model by IWERM (2), leading to the following guarantee.

**Theorem 3.** *Under the same condition as Theorem 2 and letting* $V_T = \sum_{t=2}^{T} \|\mathcal{D}_t(\mathbf{x}) - \mathcal{D}_{t-1}(\mathbf{x})\|_1$, *running IWERM* (2) *with the estimated density ratio function* $\widehat{r}_t(\mathbf{x}) = \exp(-\phi(\mathbf{x})^\top \widehat{\theta}_t)$ *yields*

$$\mathfrak{R}_T(\{\widehat{\mathbf{w}}_t\}_{t=1}^T) \leq \widetilde{\mathcal{O}}\Big(N_0^{-\frac{1}{2}} + \max\big\{T^{-\frac{1}{3}} V_T^{\frac{1}{3}}, T^{-\frac{1}{2}}\big\}\Big).$$

Theorem 3 shows that our learned predictor $\widehat{\mathbf{w}}_t$ adapts to the environment with a converged average excess risk when compared with the per-round best predictor $\mathbf{w}_t^*$. By the discussion below Proposition 1, the $\widetilde{\mathcal{O}}(N_0^{-1/2})$ generalization error over the offline data $S_0$ is hard to improve. We focus on the error in the online learning part. When the environment is nearly-stationary, i.e., $V_T \leq \mathcal{O}(T^{-1/2})$, our bound implies an $\widetilde{\mathcal{O}}(N_0^{-1/2} + T^{-1/2})$ average excess risk, matching the same rate for the one-step adaptation [49], even if the unlabeled data appear sequentially and the comparator $\mathbf{w}_t^*$ could change over time. When the environments shift quickly, the $\mathcal{O}(T^{-1/3} V_T^{1/3})$ rate still exhibits a diminishing error for covariate shift adaptation once $V_T = o(T)$. Indeed, our result has the same rate as that for continuous label shift [18] (with a slightly different definition of $V_T$ though). In Appendix D.6, we further provide evidence to show that the $\mathcal{O}(\max\{T^{-1/3} V_T^{1/3}, T^{-1/2})$ rate can hardly be improved, even if one can receive labels of the testing stream after prediction.

**Discussion on Assumptions.** We end this section by a discussion on possible future directions to relax the linear model assumption for DRE and covariate shift assumption. Since our analysis is based on the online convex optimization framework, we employed the (generalized) linear model for DRE to ensure the convexity. To go beyond the linear model while still having theoretical guarantees, one might extend the online ensemble framework to learn within the Reproducing Kernel Hilbert Space, leveraging advances in online kernel learning [50]. As for the step towards handling the general distribution shift, it is possible to consider the sparse joint shift model [51] where both covariate and label distribution could shift. Besides, studying how to handle the joint distribution shift with few online labeled data also presents an interesting future direction. Our research on the covariate shift might serve as a basic step towards addressing more complex real-world distribution shifts.

## 5 Experiments

**Setups.** We generate continuous covariate shift by combining two fixed distributions with a time-varying mixture proportion $\alpha_t \in [0, 1]$. Specifically, given two fixed distributions $\mathcal{D}'(\mathbf{x})$ and $\mathcal{D}''(\mathbf{x})$, we generate samples from $\mathcal{D}_t(\mathbf{x}) = (1 - \alpha_t)\mathcal{D}'(\mathbf{x}) + \alpha_t \mathcal{D}''(\mathbf{x})$ at each round. The mixture proportion $\alpha_t$ shifts in four patterns: in `Sin` Shift and `Squ` Shift, $\alpha_t$ changes periodically, following sine and square waves; in `Lin` Shift, the environment changes slowly from $\alpha_1 = 1$ to $\alpha_T = 0$ linearly over $T$ rounds, while in `Ber` Shift, the proportion $\alpha_t$ flips quickly between 0 and 1 with a certain probability. For parameterization in the Accous implementation, we set $R$ by directly calculating the data norm

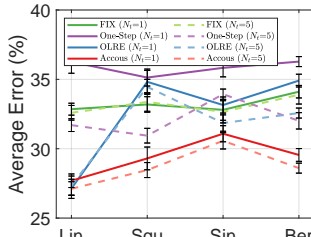 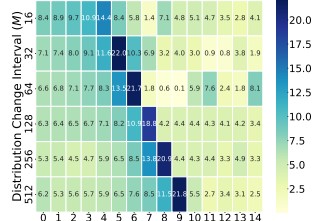 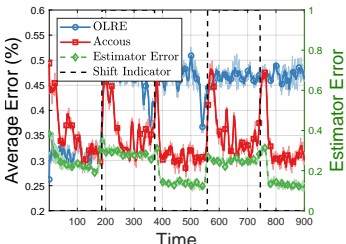

**Figure 2:** Performance comparison on the four kinds of shifts.

**Figure 3:** Weight(%) heatmap of base-learners in squared shift.

**Figure 4:** Average error and estimator loss in squared shift.

and set $S = d/2$ for all experiments. We repeat all experiments five times and evaluate the algorithms by the average classification error over the unlabeled test stream for 10,000 rounds.

**Contenders.** We compare our method with six algorithms, which can be divided into three groups. The first is a baseline approach that predicts directly using the model trained on initial offline data (*FIX*). The second group consists of the one-step covariate shift adaptation methods that do not reuse historical information: *DANN* [52] handles the shifts by learning an invariant feature space, *IW-KMM* [26], *IW-KEIEP* [25] and *IW-uLSIF* [28] equip the one-step IWERM method (2) with different density ratio estimators. The third is *OLRE*, which serves as an ablation study for our Accous algorithm. The *OLRE* algorithm estimates the density ratio by running an ONS (8) with all historical data and then performs the IWERM at each round. All algorithm parameters are set by their default.

In the following, we focus on the results of empirical studies. Detailed configurations for the datasets, simulated shifts, and setups for the Accous algorithm and contenders can be found in Appendix A.

## 5.1 Illustrations on Synthetic Data

**Average Classification Error Comparison.** We summarize the comparison results on synthetic data with different types of covariate shifts with all contenders in Figure 2. The Accous algorithm outperforms almost all other methods in the four shift patterns. We observe that the offline model FIX performs poorly compared to the online methods. The best result of DANN, IW-KMM, IW-KLIEP and IW-uLSIF is called One-Step for comparison. By reusing historical data, the Accous algorithm achieves a lower average classification error compared to these One-Step competitors that use only one round of online data. The Accous also outperforms OLRE, especially when the environments change relatively quickly (Squ, Sin, and Ber). These results justify that performing density ratio estimation with either one round of online data or all historical data is inappropriate for non-stationary environments. Moreover, when the number of unlabeled data per round changes from $N_t = 5$ to $N_t = 1$, the one-step methods suffer severe performance degradation. In contrast, our Accous algorithm still performs relatively well, highlighting the need for selective reuse of historical data.

**Effectiveness of Online Density Ratio Estimation.** We then take a closer look at the core component of our algorithm, the online density ratio estimator, in the Squ shift where the distribution of the online data shifts every $M$ rounds. Figure 3 shows the weight assignment for base-learners with different interval lengths, averaged over their active period. The result shows that our meta-learner successfully assigns the largest weight to the base-learner whose interval length matches the switching period $M$, which ensures that the right amount of historical data is reused. Figure 4 also shows the average error of Accous and OLRE over $1,000$ iterations. The results show that Accous (red line) can quickly adapt to new distributions after the covariate shift occurs. In contrast, OLRE (blue line) struggles because it uses all the historical data generated from different distributions. Furthermore, we denote the loss of the density ratio estimator $\widehat{L}_t(\widehat{\theta}_t)$ by the green line, which is quickly minimized after the covariate shift occurs. The similar tendency between the average error (red line) and the loss of the estimator (green line) confirms our theoretical finding in Theorem 1: we can minimize the excess risk of the predictor by minimizing the dynamic regret of the density ratio estimator.

## 5.2 Comparison on Real-world Data

**Performance Comparison on Benchmarks.** In Table 1 and Table 2, we present the results of the average classification error comparison with the contenders on four benchmark datasets. Our

**Table 1:** Average classification error (%) of different algorithms on various real-world datasets with `Lin` and `Ber` shifts. We report the mean accuracy and standard deviation over five runs. The best algorithms are emphasized in bold. The number of data in each round is set as $N_t = 5$.

| | Lin | | | | | | | | Ber | | | | | | | |
|---|---|---|---|---|---|---|---|---|---|---|---|---|---|---|---|---|
| | FIX | DANN | KMM | KLIEP | uLSIF | LR | OLRE | Accous | FIX | DANN | KMM | KLIEP | uLSIF | LR | OLRE | Accous |
| **Diabetes** | 39.55 | 46.88 | 43.67 | 37.85 | 38.32 | 39.17 | 37.40 | **35.92** | 40.90 | 49.03 | 46.80 | 42.98 | 43.22 | 40.28 | 42.05 | **39.14** |
| | ±6.58 | ±7.09 | ±6.23 | ±6.15 | ±6.00 | ±7.10 | ±6.56 | **±6.44** | ±6.90 | ±6.54 | ±6.67 | ±6.52 | ±6.78 | ±6.25 | ±6.22 | **±4.51** |
| **Breast** | 10.25 | 16.03 | 8.26 | 6.90 | 11.07 | 7.33 | **3.59** | 4.60 | 11.68 | 15.34 | 10.12 | 7.13 | 9.01 | 8.14 | 5.27 | **3.89** |
| | ± 3.00 | ± 4.79 | ± 0.31 | ± 0.77 | ± 2.34 | ± 1.59 | **± 0.25** | ± 0.16 | ± 4.16 | ± 5.23 | ± 0.60 | ± 1.30 | ± 1.91 | ± 2.02 | ± 0.44 | **± 0.37** |
| **MNIST-SVHN** | 8.19 | 13.37 | 6.24 | 7.52 | 6.38 | 6.81 | 6.65 | **5.93** | 11.33 | 16.54 | 10.81 | 11.50 | 12.19 | 11.18 | 13.42 | **10.31** |
| | ±1.39 | ±0.97 | ±1.52 | ±0.33 | ±0.76 | ±0.84 | ±0.93 | **±0.72** | ±0.83 | ±1.10 | ±0.63 | ±0.97 | ±1.20 | ±0.50 | ±0.49 | **±0.51** |
| **CIFAR10-CINIC10** | 29.24 | 31.59 | 29.60 | 29.33 | 29.18 | 29.75 | 28.17 | **27.80** | 33.16 | 42.50 | 33.49 | 31.44 | 32.98 | 33.03 | 31.22 | **30.69** |
| | ±0.51 | ±2.18 | ±1.43 | ±0.97 | ±1.50 | ±0.48 | ±1.24 | **±0.83** | ±0.94 | ±1.29 | ±0.83 | ±1.57 | ±0.67 | ±1.26 | ±1.01 | **±0.62** |

**Table 2:** Average classification error (%) of different algorithms on various real-world datasets with `Squ` and `Sin` shifts. We report the mean accuracy and standard deviation over five runs. The best algorithms are emphasized in bold. The number of data in each round is set as $N_t = 5$.

| | Squ | | | | | | | | Sin | | | | | | | |
|---|---|---|---|---|---|---|---|---|---|---|---|---|---|---|---|---|
| | FIX | DANN | KMM | KLIEP | uLSIF | LR | OLRE | Accous | FIX | DANN | KMM | KLIEP | uLSIF | LR | OLRE | Accous |
| **Diabetes** | 37.85 | 45.69 | 43.57 | 37.00 | 38.03 | 37.83 | 39.17 | **36.45** | 37.30 | 44.96 | 41.38 | 36.40 | 37.23 | 36.99 | 38.58 | **35.17** |
| | ±6.67 | ±6.51 | ±6.97 | ±6.80 | ±6.45 | ±5.87 | ±6.67 | **±5.36** | ±7.02 | ±6.79 | ±6.46 | ±6.93 | ±6.88 | ±6.21 | ±7.45 | **±6.44** |
| **Breast** | 9.92 | 12.54 | 8.53 | 6.52 | 7.92 | 7.39 | 4.67 | **3.68** | 8.68 | 14.34 | 7.18 | 6.11 | 9.59 | 7.50 | 4.16 | **3.29** |
| | ±0.23 | ±1.55 | ±0.30 | ±0.91 | ±1.39 | ±0.83 | ±0.23 | **±0.58** | ±0.56 | ±1.78 | ±0.39 | ±0.92 | ±1.02 | ±0.51 | ±0.42 | **±0.43** |
| **MNIST-SVHN** | 11.28 | 13.98 | 10.21 | 11.01 | 10.49 | 11.72 | 13.68 | **9.54** | 9.98 | 16.73 | 8.91 | 9.41 | 9.30 | 9.00 | 15.38 | **7.74** |
| | ±1.22 | ±0.97 | ±0.53 | ±1.41 | ±0.67 | ±1.89 | ±0.59 | **±0.40** | ±1.29 | ±1.03 | ±0.68 | ±1.01 | ±0.53 | ±1.77 | ±0.88 | **±0.74** |
| **CIFAR10-CINIC10** | 34.75 | 38.13 | 34.08 | 34.27 | 33.80 | 33.56 | 32.61 | **31.08** | 36.44 | 42.53 | 37.00 | 36.58 | 36.53 | 36.21 | 38.24 | **35.42** |
| | ±0.29 | ±1.58 | ±0.85 | ±1.24 | ±1.73 | ±1.59 | ±0.69 | **±0.60** | ±0.47 | ±1.09 | ±0.67 | ±1.52 | ±1.02 | ±1.31 | ±1.86 | **±1.22** |

algorithm Accous outperforms other contenders in the four types of shifts, whether in a slowly evolving environment (`Lin`) or in relatively non-stationary environments (`Squ`,`Sin`,`Ber`). The OLRE algorithm uses all historical data, but does not necessarily outperform contenders that use only current data, suggesting that we should selectively reuse the historical data.

**Case Study on a Real-life Application.** We conducted empirical studies on a real-world application using the yearbook dataset [53], which contains high school front-facing yearbook photos from 1930 to 2013. The photo distributions shift along years due to the changing social norms, fashion styles, and population demographics. Consequently, it is essential to adapt to these changes. We used photos before 1945 as the offline labeled dataset and generated the online unlabeled data stream using photos after 1945, arranged chronologically with 10 photos per round. Figure 5 shows the average error curves. The OLER algorithm performs well when the distributions do not change too much, but the method starts to suffer from large prediction errors after 400 iterations. In contrast, our method outperforms all competitors by selectively reusing historical data.

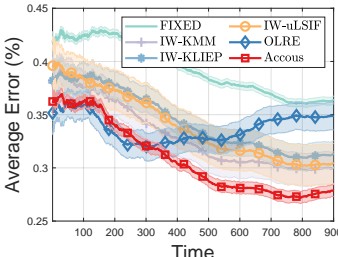

**Figure 5:** Average error on Yearbook dataset with real-life covariate shift.

## 6 Conclusion

This paper initiated the study of the continuous covariate shift with the goal of minimizing the cumulative excess risk of the predictors. We showed that the importance-weighted ERM method works effectively given high-quality density ratio estimators (Proposition 1), whose estimation can be cast as a dynamic regret minimization problem (Theorem 1). Instantiating with the logistic regression density ratio estimation model, we proposed a novel online that can adaptively reuse the historical data and enjoy a tight dynamic regret bound (Theorem 2). The regret bound finally implies an $\widetilde{\mathcal{O}}(T^{-1/3}V_T^{1/3})$ averaged excess risk guarantee for the predictor. Experiments validate the effectiveness of the proposed method and demonstrate the need for selective reuse of historical data.

## Acknowledgments

YJZ and MS were supported by the Institute for AI and Beyond, UTokyo. YJZ was also supported by Todai Fellowship. Peng Zhao was supported by NSFC (62206125) and JiangsuSF (BK20220776). The authors thank Yong Bai and Yu-Yang Qian for their assistance in the experiments.

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

# A Omitted Details for Experiments

In this section, we first provide additional details that were omitted in section 5. We then provide the descriptions of our experimental setups, which include details of the datasets used, simulations of continuous covariate shifts, descriptions of the competing algorithms, and the parameter configuration of our proposed Accous algorithm. After that, we supplement the detailed numerical results obtained from the synthetic experiments.

## A.1 Experiment Setup

**Continuous Covariate Shift Simulation.** To simulate the continuous covariate shift, we generate offline and online samples from different marginal distributions and annotate them with the same labeling function. Given two different marginal distributions $\mathcal{D}'(\mathbf{x})$ and $\mathcal{D}''(\mathbf{x})$, the offline dataset $S_0$ is sampled from the distribution $\mathcal{D}_0(\mathbf{x}) = (1 - \alpha_0)\mathcal{D}'(\mathbf{x}) + \alpha_0\mathcal{D}''(\mathbf{x})$, where $\alpha_0 \in [0, 1]$ is a mixture proportion coefficient. The online dataset $S_t$ received at each iteration $t$ is sampled from the distribution $\mathcal{D}_t(\mathbf{x}) = (1 - \alpha_t)\mathcal{D}'(\mathbf{x}) + \alpha_t\mathcal{D}''(\mathbf{x})$, where $\alpha_t \in [0, 1]$ is the time-varying coefficient. By changing the coefficient $\alpha_t$ for $t \geq 1$, we can simulate different types of continuous covariate shifts with different shift patterns.

- `Linear Shift (Lin)`: We set $\alpha_t = t/T$ to simulate a gradually changing environment.

- `Square Shift (Squ)`: In this case, the parameter $\alpha_t$ switches between $1$ and $0$ every $M$ rounds, where $2M$ is the periodic length. We set $M = \Theta(\sqrt{T})$ to simulate a fast-changing environment with periodic patterns.

- `Sine Shift (Sin)`: We set $\alpha_t = \sin\frac{i\pi}{M}$, where $i = t \bmod M$ and $M$ is the periodic length. We set $M = \Theta(\sqrt{T})$ to simulate a fast-changing environment with periodic patterns.

- `Bernoulli Shift (Ber)`: In this case, we keep the $\alpha_t = \alpha_{t-1}$ with probability $p \in [0, 1]$ and otherwise set $\alpha_t = 1 - \alpha_{t-1}$. We set $p = \Theta(1/\sqrt{T})$ to simulate a fast-changing environment but without periodic patterns.

**Datasets.** For the synthetic dataset, we generate $\mathcal{D}'(\mathbf{x})$ and $\mathcal{D}''(\mathbf{x})$ using the multinomial Gaussian distribution, wherein each class is generated from a Gaussian distribution. For $\mathcal{D}'(\mathbf{x})$, the positive class data is generated from $\mathcal{N}(\mathbf{x}; \boldsymbol{\mu}_1^p, \Sigma)$, and the negative class data is generated from $\mathcal{N}(\mathbf{x}; \boldsymbol{\mu}_1^n, \Sigma)$. Here, $\boldsymbol{\mu}_1^p$ is set to $-\mathbf{1.2}$ in $\mathbb{R}^{12}$, and $\boldsymbol{\mu}_1^n$ is set to $-\mathbf{0.8}$ in $\mathbb{R}^{12}$. As for $\mathcal{D}''(\mathbf{x})$, its positive class data is generated from $\mathcal{N}(\mathbf{x}; \boldsymbol{\mu}_2^p, \Sigma)$, and the negative class data is generated from $\mathcal{N}(\mathbf{x}; \boldsymbol{\mu}_2^n, \Sigma)$. Here, $\boldsymbol{\mu}_2^p$ is specified as $\mathbf{1.2}$ in $\mathbb{R}^{12}$, and $\boldsymbol{\mu}_2^n$ as $\mathbf{0.8}$ in $\mathbb{R}^{12}$. The covariance matrix is set as $\Sigma = I_d$ in $\mathbb{R}^{12 \times 12}$, and the prior probability for the positive class data is fixed at $0.5$. Besides, we generated offline data with $\alpha_0 = 0.9$ and introduced a time-varying coefficient, $\alpha_t$, to simulate continuous covariate shifts.

The benchmark datasets are generated from real-world datasets.

- *Diabetes (Dia)* is a tabular UCI dataset. Each sample is an $8$-dimensional vector with a binary label. We split the dataset into two subsets and choose the split such that the classifier trained on one dataset generalizes poorly to the other. We choose one of them as the offline dataset and combine them with a time-varying coefficient to generate the online data.

- *Breast (Bre)* is a tabular UCI dataset. Each sample is a $9$-dimensional vector with a binary label. This dataset is generated similarly to the Diabetes dataset.

- *MNIST-SVHN (M-S)* contains the digital images collected from handwriting or natural scene images. A feature extractor pre-trained on this dataset produces a $512$-dimensional vector with a $10$-class label for each sample. We choose $10\%$ MNIST data and $90\%$ SVHN data as the offline dataset and generate the online data by mixing the MNIST and SVHN datasets with a time-varying proportion coefficient $\alpha_t$.

- *CIFAR10-CINIC10 (C-C)* contains real-life subjective images collected from different sources. This dataset is generated similarly to the MNIST-SVHN dataset.

We also conduct experiments on a real-world dataset where the underlying data distributions exhibit a continuous shift.

- *Yearbook* [53]: This dataset contains the 37,921 frontal-facing American high school yearbook photos from 1930 to 2013 [54]. Each photo is a $32 \times 32 \times 1$ grayscale image associated with a binary label representing the student's gender. This dataset captures the real-life changing of social norms, fashion styles, and population demographics. The offline data is generated with photos before 1945, the online data is generated from 1946, and the data size is 10 per round.

**Contenders.** We compare our proposed approach with six contenders, including:

- *FIX* is the baseline method that predicts with the classifier that are pre-trained on offline data without any updates in the online adaptation stage.

- is a general domain adaptation algorithm that mitigates the discrepancy between the training and test distributions by learning an invariant feature representation. We adapt this method to the online setting by running DANN at each round $t \in [T]$ based on the offline data $S_0$ and the online data $S_t$ available at that round.

- *IW-KMM* is an importance weighting learning method that first estimates the density ratios by the Kernel Mean Matching (KMM) method [26] and then performs the IWERM to train the model. At each iteration $t \in [T]$, the density ratios are estimated with the offline dataset $S_0$ and the online dataset $S_t$ by the KMM. Then the model is trained by running the IWERM method on the offline data $S_0$.

- *IW-KLIEP* is also an importance weighting learning method, where the density ratios are estimated by the KLIEP method [25]. At each iteration $t \in [T]$, after the density ratios have been estimated by KLIEP, the learner performs the IWERM over the offline data $S_0$.

- *IW-uLSIF* is an importance weighting learning method where the density ratio estimated by the uLSIF method is used [28]. At each iteration $t \in [T]$, the learner first estimates the density ratios by the uLSIF and then performs the IWERM on the offline data $S_0$.

- *LR* serves as an ablation study for the proposed Accous algorithm. At each iteration $t \in [T]$, the learner first estimates the density ratios by an ONS with a logistic regression model and then performs the IWERM on the offline data $S_0$.

- *OLRE* also serves as an ablation study for the proposed Accous algorithm. It estimates the density ratios in an on-line manner by running an ONS with a logistic regression model with all the historical data and then performing the IWERM each round.

**Accous Setup.** We implement the Accous algorithm using a logistic regression model. Before running the Accous algorithm, two parameters must be defined. They are the maximum norm of the feature $R$ and the maximum norm of the density functions $S$. We set $R$ by directly calculating the data norm. The covariate shift level determines the parameter $S$, which is difficult to predict in advance. We found that setting $S = d/2$ is a good choice in the experiments. For stable performance, we set a threshold of 100 and truncate the large weights after the density ratio estimation in each round. We also ignore the base models with interval lengths 1, 2 in the density ratio estimation.

For the synthetic, diabetes, and breast tabular datasets, the linear model is used. For the MNIST-SVHN, CIFAR10-CINIC10, and Yearbook datasets, the deep model is used for density ratio estimation and predictive model training. The deep models consist of a backbone and a linear layer. In all experiments, only the linear layer of the neural network is updated, while the backbone parameters trained with offline data remain fixed. Backbone settings for different datasets are presented below.

- *MNIST-SVHN* and *CIFAR10-CINIC10*: The backbone is a pre-trained ResNet34 backbone from torchvision with its weights initialized by training on ImageNet. Both the weight estimator and the classifier share the same backbone and its corresponding parameters.

- *Yearbook*: Our network adopts a similar structure and updating process as in the *MNIST-SVHN* and *CIFAR10-CINIC10* experiments, following the configuration described in [53]. The backbone of our network consists of a CNN model with the default settings.

**Computational Resources.** We run experiments with two Xeon Gold 6248R processors (24 cores, 3.0GHz base, 4.0GHz boost), eight Tesla V100S GPUs (32GB video memory each), 768GB RAM, all managed by the Ubuntu 20.04 operating system.

**Table 3:** Average error (%) over the entire time horizons for different algorithms under various simulated shifts. The best algorithms are emphasized in bold.

| | $N_t = 1$ | | | | | | | | $N_t = 5$ | | | | | | | |
|---|---|---|---|---|---|---|---|---|---|---|---|---|---|---|---|---|
| | FIX | DANN | KMM | KLIEP | uLSIF | LR | OLRE | Accous | FIX | DANN | KMM | KLIEP | uLSIF | LR | OLRE | Accous |
| Lin | 32.85 | 46.93 | 37.97 | 36.26 | 38.34 | 36.51 | **27.13** | 27.69 | 32.59 | 40.33 | 32.68 | 34.13 | 31.70 | 32.21 | 27.34 | **27.11** |
| | ±0.43 | ±2.40 | ±0.56 | ±0.82 | ±0.40 | ±1.12 | ±**0.74** | ±0.49 | ±0.53 | ±2.99 | ±0.86 | ±0.30 | ±0.46 | ±0.72 | ±0.81 | ±**0.69** |
| Squ | 33.19 | 40.59 | 35.50 | 35.13 | 36.44 | 36.21 | 34.82 | **29.30** | 33.37 | 41.24 | 31.52 | 30.94 | 32.07 | 31.63 | 34.46 | **28.46** |
| | ±0.56 | ±3.28 | ±0.33 | ±0.61 | ±0.79 | ±0.46 | ±0.30 | ±**0.82** | ±0.67 | ±4.01 | ±0.46 | ±0.53 | ±0.36 | ±1.08 | ±0.86 | ±**0.54** |
| Sin | 32.80 | 42.65 | 36.25 | 35.83 | 37.01 | 37.42 | 33.15 | **31.08** | 32.66 | 39.94 | 34.15 | 34.47 | 33.93 | 34.62 | 31.85 | **30.57** |
| | ±0.75 | ±4.49 | ±0.37 | ±0.65 | ±0.38 | ±0.73 | ±0.50 | ±**0.58** | ±0.80 | ±2.94 | ±0.45 | ±0.72 | ±0.37 | ±0.56 | ±0.28 | ±**0.59** |
| Ber | 34.11 | 46.20 | 36.49 | 37.07 | 36.28 | 37.42 | 34.91 | **29.55** | 33.89 | 45.33 | 32.89 | 33.52 | 32.01 | 34.13 | 32.57 | **28.60** |
| | ±0.45 | ±4.12 | ±0.42 | ±0.67 | ±0.36 | ±0.85 | ±0.47 | ±**0.46** | ±0.67 | ±2.88 | ±0.36 | ±0.78 | ±0.59 | ±0.72 | ±0.40 | ±**0.36** |

## A.2 Supplementary Numerical Results for Section 5.1

In Section 5.1 we summarize the results of the synthetic experiments, and here in Table 3, we supplement their detailed numerical results. Overall, Accous outperforms almost all other methods in the four shift patterns. Compared to the competitors that use only one round of online data, our proposed method achieves a lower average error. The general domain adaptation method DANN is also inferior in the online setting due to the limited amount of data per round, which makes it difficult to learn a good representation. The proposed Accous algorithm also outperforms the OLRE algorithm, which reuses all historical data. The above evidence demonstrates the need for selective reuse of historical data and validates the effectiveness of our algorithm.

# B More Related Work

## B.1 Related Work on Non-stationary Online Learning

Online convex optimization (OCO) [13] is a powerful paradigm to handle sequential prediction problems. To adapt to the continuously shifting environments, we exploit the online ensemble technique [15, 16] developed in non-stationary online learning literature to hedge the uncertainty. In this part, we will introduce the related works and discuss the difference between our algorithm and the previous work.

Specifically, OCO paradigm considers a $T$-round game between a learner and the environment. At every round $t$, the learner makes a prediction $\theta_t \in \Theta$, and, at the same time, the environments reveal the convex loss function $L_t : \Theta \to \mathbb{R}$. Then, the learner suffers from the loss $L_t(\theta_t)$ and observes certain information about $L_t$ to update the model for the next iteration. A classical performance measure for OCO framework in non-stationary environments is the dynamic regret,

$$\mathbf{Reg}_T^{\mathbf{d}}(\{\widetilde{L}_t, \theta_t^*\}_{t=1}^T) = \sum_{t=1}^T L_t(\theta_t) - \sum_{t=1}^T L_t(\theta_t^*), \tag{10}$$

which compares the learner's prediction with the function minimizer $\theta_t^*$ at every iteration. Over the decades, a variety of online learning algorithms have been proposed to optimize this dynamic regret measure [38, 39, 55, 56, 57, 58, 59, 60, 61, 62, 63] in the online convex optimization problem.

**Worst-case Dynamic Regret.** Many works focus on the full information setting [57, 58, 59, 60, 63], where the learner can observe the loss functions entirely. We call the dynamic regret in such a case as "worse-case" dynamic regret in the sense that the comparator $\theta_t^*$ is exactly the minimizer of the observed function $L_t$. According to different complexity measures for quantifying the environmental non-stationarity, there are two lines of results. The first is the path length of comparators $V_T^\theta = \sum_{t=2}^T \|\theta_t^* - \theta_{t-1}^*\|_2$ measuring the variation of the function minimizers; and the second one is the variation of the function values $V_T^L = \sum_{t=2}^T \max_{\theta \in \Theta} |L_t(\theta) - L_{t-1}(\theta)|$. For the full information setting, the prior art [63] showed that a simple greedy strategy can already be already well-behaved and achieves an $\mathcal{O}(\min\{V_T^\theta, V_T^L\})$ dynamic regret bound that minimizes the two kinds of non-stationarity measure simultaneously.

**Universal Dynamic Regret.** Competing to the sequences of observed functions' minimizers can be sometimes too pessimistic for the online optimizing, which may lead to a severe overfitting. A more appropriate performance measure is the universal dynamic regret [38], which has drawn more and more attention in recent years. Universal dynamic regret compares the learner's prediction with an

*arbitrary* comparator sequence $\{\nu_t\}_{t=1}^T$,

$$\mathbf{Reg}_T^{\mathbf{d}}(\{\widetilde{L}_t, \nu_t\}_{t=1}^T) = \sum_{t=1}^T L_t(\theta_t) - \sum_{t=1}^T L_t(\nu_t). \tag{11}$$

Let $V_T^{\nu} = \sum_{t=1}^T \|\nu_t - \nu_{t-1}\|_2$ be the path length of the comparator sequence $\{\nu_t\}_{t=1}^T$. [38] showed that the online gradient descent algorithm achieves an $\mathcal{O}((1 + V_T^{\nu})\sqrt{T})$ universal dynamic regret bound for convex function, whereas there is still a gap to the $\Omega(\sqrt{T(1 + V_T^{\nu})})$ lower bound established by [15], who further proposed an algorithm attaining an $\mathcal{O}(\sqrt{T(1 + V_T^{\nu})})$ dynamic regret bound and thus closed the gap. Their key algorithmic ingredient is the online ensemble structure [64], which hedges the non-stationarity of the environments by employing a meta-learner to ensemble the predictions from a group of base-learners. When, the loss function is convex and smooth, the $\mathcal{O}(\sqrt{T(1 + V_T^{\nu})})$ can be improved to $\mathcal{O}(\sqrt{(1 + \min\{F_T, G_T\} + V_T^{\nu})(1 + V_T^{\nu})})$ [16], where $F_T = \sum_{t=1}^T L_t(\nu_t)$ and $G_T = \sum_{t=2}^T \max_{\theta \in \Theta} \|\nabla L_t(\theta) - \nabla L_{t-1}(\theta)\|_2^2$ are two problem-dependent quantities capturing different kinds of easiness of the environments. For the exp-concave and smooth functions, [19] showed an $\mathcal{O}(T^{1/3}(V_T^{\nu})^{2/3})$ universal dynamic regret by an improper algorithm (i.e., the output decisions can be slightly out of the feasible domain, while comparators have to locate in the domain), which is proven to be minimax optimal for exp-concave functions. Note that the worst-case dynamic regret (10) is clearly a special case of the universal dynamic regret defined in (11), so a universal dynamic regret upper bound directly implies an upper bound for the worst-case dynamic regret, by substituting comparators $\{\nu_t\}_{t=1}^T$ as functions' minimizer $\{\theta_t^*\}_{t=1}^T$. This method yields a path-length type bound. Furthermore, we mention that it is also possible to obtain a function-variation type worst-case dynamic regret bound from the universal dynamic regret guarantee, which is revealed in [61, Appendix A.2].

**An Intermediate Dynamic Regret.** There is also an intermediate case between the worst-case and universal dynamic regret. We refer to it the "intermediate" dynamic regret in this paper. In this intermediate case, the online learner still aims to optimize the worst-case dynamic regret of the loss function $L_t(\theta)$, but she can only observe a *noisy* feedback $\widehat{L}_t(\theta)$ at each iteration. Since the comparator $\theta_t^* = \arg\min_{\theta \in \Theta} L_t(\theta)$ is not the minimizer of the observed loss $\widehat{L}_t(\theta)$, such a kind of problem is more challenging than the worst-case dynamic regret minimization problem given the uncertainty of the comparator sequence. [39] showed a restarted online gradient descent algorithm can achieve an $\mathcal{O}(T^{1/3}(V_T^L)^{2/3})$ dynamic regret bound for convex function and an $\mathcal{O}(\sqrt{T V_T^L})$ bound for strongly convex function under the noisy feedback. However, their algorithm requires the knowledge of the non-stationarity measure $V_T^L = \sum_{t=2}^T \max_{\theta \in \Theta} |L_t(\theta) - L_{t-1}(\theta)|$ defined over $L_t$, which is generally unknown. [48] showed that an $\mathcal{O}(T^{1/3}(V_T^{\theta})^{2/3})$ bound is achievable with the noisy feedback when $L_t$ is a one-dimensional squared loss. The proposed method is based on the de-noising technique and are free from the knowledge of $V_T^{\theta}$. Evidently, this intermediate dynamic regret is still a feasible realization of the universal dynamic regret, so one can achieve the intermediate dynamic regret bound by the universal dynamic regret. However, this reduction will only provide an expected guarantee. More detailed discussions are provided below.

**Comparison of Our Results and Earlier Works.** Our $\mathbf{Reg}_T^{\mathbf{d}}(\{\widetilde{L}_t, \theta_t^*\}_{t=1}^T) \leq \mathcal{O}(T^{1/3}(V_T^{\theta})^{2/3})$ high-probability dynamic regret bound for logistic regression (Theorem 2) falls into the intermediate case, where we aim to minimize the worst-case dynamic regret bound in terms of $L_t$ with noisy feedback. The closest related work is by [19], who also exploited an online ensemble structure and achieved a universal dynamic regret in the form of $\mathbf{Reg}_T^{\mathbf{d}}(\{\widetilde{L}_t, \nu_t\}_{t=1}^T) \leq \mathcal{O}(T^{1/3}(V_T^{\nu})^{2/3})$ universal dynamic regret by taking $\widehat{L}_t$ as the input. Further choosing $\nu_t = \theta_t^*$ and taking the expectation over $S_t$ for each round, their result implies an expectation bound $\mathbb{E}[\mathbf{Reg}_T^{\mathbf{d}}] \leq \mathcal{O}(T^{1/3}(V_T^{\theta})^{2/3})$. One main advantage of our result is that the bound holds with high probability. To achieve the high-probability bound, we twist the meta-learner from Hedge [42] to Adapt-ML-Prod [43], which helps control the generalization error between $\widehat{L}_t$ and $\widetilde{L}_t$ with the negative term introduced by the exp-concavity of the loss function. Besides, similar to [19], our analysis crucially relies on providing a squared formulation of the dynamic regret $\mathcal{O}(\max\{1, |I|(V_T^{\theta})^2\})$ on each interval $I$. However, to obtain such a result, [19] crucially rely on a complicated analysis on the KKT condition to depict the structure of comparators. We greatly simplify the analysis by exploiting the fact that in our case the comparator $\theta_t^*$ is actually the minimizer of expected functions $L_t$, hence avoiding analyzing KKT condition. Our

analysis is applicable to the case when the minimizers lie in the interior of the feasible domain and require the smoothness of the online functions only (which does not rely on the particular properties of logistic loss). This result might be of independent interest in other online learning studies.

## C   Omitted Details for Section 3

This section provides the proofs for Section 3 in the main paper.

### C.1   Proof of Proposition 1

*Proof of Proposition 1.* We first provide an excess risk bound for the model $\widehat{\mathbf{w}}_t$ at each time $t$ and then combine them for $T$ rounds. For each time $t \in [T]$, let $\widetilde{R}_t(\mathbf{w}) = \mathbb{E}_{(\mathbf{x},y) \sim S_0}\left[r_t^*(\mathbf{x})\ell(\mathbf{w}^\top \mathbf{x}, y)\right]$ be the risk built upon the true density ratio function $r_t^*$ and $\widehat{R}_t(\mathbf{w}) = \mathbb{E}_{(\mathbf{x},y) \sim S_0}\left[\widehat{r}_t(\mathbf{x})\ell(\mathbf{w}^\top \mathbf{x}, y)\right]$ be that built upon the empirical density ratio function $\widehat{r}_t$. We can decompose the excess risk of the trained model $\widehat{\mathbf{w}}_t$ as

$$R_t(\widehat{\mathbf{w}}_t) - R_t(\mathbf{w}_t^*) = \underbrace{R_t(\widehat{\mathbf{w}}_t) - \widetilde{R}_t(\widehat{\mathbf{w}}_t)}_{\texttt{term (a)}} + \underbrace{\widetilde{R}_t(\widehat{\mathbf{w}}_t) - \widetilde{R}_t(\mathbf{w}_t^*)}_{\texttt{term (b)}} + \underbrace{\widetilde{R}_t(\mathbf{w}_t^*) - R_t(\mathbf{w}_t^*)}_{\texttt{term (c)}}.$$

For term (a), by a standard generalization error analysis as shown in Lemma 2, with probability at least $1 - \delta/(2T)$, the gap between $R_t(\widehat{\mathbf{w}}_t)$ and $\widetilde{R}_t(\widehat{\mathbf{w}}_t)$ is bounded by

$$\texttt{term (a)} = R_t(\widehat{\mathbf{w}}_t) - \widetilde{R}_t(\widehat{\mathbf{w}}_t) \leq \frac{2BGDR}{\sqrt{N_0}} + 4BL\sqrt{\frac{2\ln((8T)/\delta)}{N_0}}.$$

Then, we proceed to bound term (b).

$$
\begin{aligned}
\texttt{term (b)} &= \widetilde{R}_t(\widehat{\mathbf{w}}_t) - \widetilde{R}_t(\mathbf{w}_t^*) \\
&= \widetilde{R}_t(\widehat{\mathbf{w}}_t) - \widehat{R}_t(\widehat{\mathbf{w}}_t) + \widehat{R}_t(\widehat{\mathbf{w}}_t) - \widehat{R}_t(\mathbf{w}_t^*) + \widehat{R}_t(\mathbf{w}_t^*) - \widetilde{R}_t(\mathbf{w}_t^*) \\
&\leq \widetilde{R}_t(\widehat{\mathbf{w}}_t) - \widehat{R}_t(\widehat{\mathbf{w}}_t) + \widehat{R}_t(\mathbf{w}_t^*) - \widetilde{R}_t(\mathbf{w}_t^*) \\
&= \mathbb{E}_{(\mathbf{x},y) \sim S_0}\left[(r_t^*(\mathbf{x}) - \widehat{r}_t(\mathbf{x}))\ell(\widehat{\mathbf{w}}_t^\top \mathbf{x}, y)\right] + \mathbb{E}_{(\mathbf{x},y) \sim S_0}\left[(\widehat{r}_t(\mathbf{x}) - r_t^*(\mathbf{x}))\ell((\mathbf{w}_t^*)^\top \mathbf{x}, y)\right] \\
&\leq 2L\mathbb{E}_{\mathbf{x} \sim S_0}[|r_t^*(\mathbf{x}) - \widehat{r}_t(\mathbf{x})|],
\end{aligned}
$$

where the first inequality is due to the optimality of $\widehat{\mathbf{w}}_t$ and the second inequality is by the condition $\max_{\mathbf{x} \in \mathcal{W}, y \in \mathcal{Y}, \mathbf{w} \in \mathcal{W}}|\ell(\mathbf{w}^\top \mathbf{x}, y)| \leq L$.

Finally, as for term (c), since $\mathbf{w}_t^*$ is independent of $S_0$, a direct application of the Hoeffding's inequality [65, Lemma B.6] implies,

$$\texttt{term (c)} = \widetilde{R}_t(\mathbf{w}_t^*) - R_t(\mathbf{w}_t^*) \leq BL\sqrt{\frac{2\ln((8T)/\delta)}{N_0}}$$

with probability at least $1 - \delta/(2T)$. Combining the upper bounds for term (a), term (b) and term (c) and taking the summation over $T$ rounds, we get

$$\mathfrak{R}_T(\{\widehat{\mathbf{w}}_t\}_{t=1}^T) \leq \frac{2BGDRT}{\sqrt{N_0}} + 5BLT\sqrt{\frac{2\ln((8T)/\delta)}{N_0}} + 2L\sum_{t=1}^T \mathbb{E}_{\mathbf{x} \sim S_0}[|r_t^*(\mathbf{x}) - \widehat{r}_t(\mathbf{x})|]$$

with probability at least $1 - \delta$, which completes the proof. $\qquad\square$

### C.2   Proof of Proposition 2

*Proof of Proposition 2.* We begin with converting the estimation error to the squared formulation,

$$\sum_{t=1}^T \mathbb{E}_{\mathbf{x} \sim \mathcal{D}_0(\mathbf{x})}[|r_t^*(\mathbf{x}) - \widehat{r}_t(\mathbf{x})|] \leq \sum_{t=1}^T \sqrt{\mathbb{E}_{\mathbf{x} \sim \mathcal{D}_0(\mathbf{x})}[|r_t^*(\mathbf{x}) - \widehat{r}_t(\mathbf{x})|^2]}$$

$$\leq \sqrt{T \left( \sum_{t=1}^{T} \mathbb{E}_{\mathbf{x} \sim \mathcal{D}_0(\mathbf{x})}[|r_t^*(\mathbf{x}) - \widehat{r}_t(\mathbf{x})|^2] \right)}, \qquad (12)$$

where the first inequality is by Jensen's inequality and the last inequality is due to the Cauchy-Schwarz inequality. Then, we can bound the squared density ratio estimation error as

$$\frac{\mu}{2} \mathbb{E}_{\mathbf{x} \sim \mathcal{D}_0(\mathbf{x})} \left[ |\widehat{r}_t(\mathbf{x}) - r_t^*(\mathbf{x})|^2 \right] \leq \mathbb{E}_{\mathbf{x} \sim \mathcal{D}_0(\mathbf{x})} \left[ \psi \left( r_t^*(\mathbf{x}) \right) - \psi \left( \widehat{r}_t(\mathbf{x}) \right) - \partial \psi \left( \widehat{r}_t(\mathbf{x}) \right) \left( r_t^*(\mathbf{x}) - \widehat{r}_t(\mathbf{x}) \right) \right]$$

$$= \mathrm{EB}_\psi(r_t^* \| \widehat{r}_t) = L_t^\psi(\widehat{r}_t) - L_t^\psi(r_t^*), \qquad (13)$$

where the first inequality is due to the strong convexity of the function $\psi$. The last equality is by the definition of the loss function $L_t^\psi$. Combining (12) and (13), we have

$$\sum_{t=1}^{T} \mathbb{E}_{\mathbf{x} \sim \mathcal{D}_0(\mathbf{x})}[|\widehat{r}_t(\mathbf{x}) - r_t^*(\mathbf{x})|] \leq \sqrt{\frac{2T}{\mu} \left( \sum_{t=1}^{T} L_t^\psi(\widehat{r}_t) - \sum_{t=1}^{T} L_t^\psi(r_t^*) \right)}.$$

$\square$

### C.3   Proof of Theorem 1

*Proof of Theorem 1.* Following the same arguments in obtaining (12), the empirical cumulative estimation error of the density ratio estimators can be bounded as

$$\sum_{t=1}^{T} \mathbb{E}_{\mathbf{x} \sim S_0(\mathbf{x})}[|r_t^*(\mathbf{x}) - \widehat{r}_t(\mathbf{x})|] \leq \sqrt{T \left( \sum_{t=1}^{T} \mathbb{E}_{\mathbf{x} \sim S_0(\mathbf{x})} \left[ |r_t^*(\mathbf{x}) - \widehat{r}_t(\mathbf{x})|^2 \right] \right)}. \qquad (14)$$

Then, we proceed to bound the squared cumulative estimation error by the regret of the loss function $\widetilde{L}_t^\psi(r)$ defined in (5). For notation simplicity, we define

$$f_1(z) = \partial \psi(z) z - \psi(z) \quad \text{and} \quad f_2(z) = \partial \psi(z).$$

In such a case, the loss function $\widetilde{L}_t^\psi(r) = \mathbb{E}_{\mathbf{x} \sim S_0} [f_1(r(\mathbf{x}))] - \mathbb{E}_{\mathbf{x} \sim \mathcal{D}_t(\mathbf{x})} [f_2(r(\mathbf{x}))]$ and the regret over $\widetilde{L}_t^\psi$ can be written as

$$\widetilde{L}_t^\psi(\widehat{r}_t) - \widetilde{L}_t^\psi(r_t^*) = \mathbb{E}_{\mathbf{x} \sim S_0}[f_1(\widehat{r}_t(\mathbf{x})) - f_1(r_t^*(\mathbf{x}))] - \mathbb{E}_{\mathbf{x} \sim \mathcal{D}_t}[f_2(\widehat{r}_t(\mathbf{x})) - f_2(r_t^*(\mathbf{x}))]. \qquad (15)$$

For the first term of R.H.S. of (15), by the Taylor's Theorem, for each $\mathbf{x} \in \mathcal{X}$, we have

$$\mathbb{E}_{\mathbf{x} \sim S_0}[f_1(\widehat{r}_t(\mathbf{x})) - f_1(r_t^*(\mathbf{x}))]$$

$$= \mathbb{E}_{\mathbf{x} \sim S_0} \left[ \partial f_1(r_t^*(\mathbf{x}))(\widehat{r}_t(\mathbf{x}) - r_t^*(\mathbf{x})) + \frac{\partial^2 f_1(\xi_{\mathbf{x}})}{2}(\widehat{r}_t(\mathbf{x}) - r_t^*(\mathbf{x}))^2 \right]$$

$$\geq \mathbb{E}_{\mathbf{x} \sim S_0} \left[ \partial^2 \psi \big( r_t^*(\mathbf{x}) \big) r_t^*(\mathbf{x}) \big( \widehat{r}_t(\mathbf{x}) - r_t^*(\mathbf{x}) \big) \right] + \frac{\mu}{2} \mathbb{E}_{\mathbf{x} \sim S_0} \left[ (\widehat{r}_t(\mathbf{x}) - r_t^*(\mathbf{x}))^2 \right]. \qquad (16)$$

In above $\xi_{\mathbf{x}} \in \operatorname{dom} \psi$ is a certain point on the line collecting $\widehat{r}_t(\mathbf{x})$ and $r_t^*(\mathbf{x})$. The second inequality is due to the definition of $f_1$ such that,

$$\partial^2 f_1(\xi_{\mathbf{x}}) = \partial^2 \psi(\xi_{\mathbf{x}}) - \partial^3 \psi(\xi_{\mathbf{x}}) \xi_{\mathbf{x}} \geq \mu,$$

where the inequality holds due to the $\mu$-strong convexity of $\psi(z)$ and the condition $\xi_{\mathbf{x}} \partial^3 \psi(\xi_{\mathbf{x}}) \leq 0$ for all $\xi_{\mathbf{x}} \in \operatorname{dom}\psi$.

Then, we proceed to bound the second term of R.H.S. of (15),

$$\mathbb{E}_{\mathbf{x} \sim \mathcal{D}_t}[f_2(\widehat{r}_t(\mathbf{x})) - f_2(r_t^*(\mathbf{x}))]$$

$$= \mathbb{E}_{\mathbf{x} \sim \mathcal{D}_t} \left[ \partial f_2 \big( r_t^*(\mathbf{x}) \big) (\widehat{r}_t(\mathbf{x}) - r_t^*(\mathbf{x})) \right] + \mathbb{E}_{\mathbf{x} \sim \mathcal{D}_t} \left[ \frac{\partial^2 f_2(\xi_{\mathbf{x}})}{2} (\widehat{r}_t(\mathbf{x}) - r_t^*(\mathbf{x}))^2 \right]$$

$$\leq \mathbb{E}_{\mathbf{x} \sim \mathcal{D}_t} \left[ \partial^2 \psi \big( r_t^*(\mathbf{x}) \big) (\widehat{r}_t(\mathbf{x}) - r_t^*(\mathbf{x})) \right] = \mathbb{E}_{\mathbf{x} \sim \mathcal{D}_0} \left[ \partial^2 \psi \big( r_t^*(\mathbf{x}) \big) r_t^*(\mathbf{x}) (\widehat{r}_t(\mathbf{x}) - r_t^*(\mathbf{x})) \right], \quad (17)$$

In above $\xi_{\mathbf{x}} \in \mathrm{dom}\,\psi$ is a certain point on the line connecting $\widehat{r}_t(\mathbf{x})$ and $r_t^*(\mathbf{x})$. The first inequality is due to the definition of $f_2$ and the condition on $\psi$, such that $\partial^2 f_2(\xi_{\mathbf{x}}) = \partial^3 \psi(\xi_{\mathbf{x}}) \leq 0$. Plugging (16) and (17) into (15), and rearranging the terms, we arrive

$$\frac{\mu}{2}\mathbb{E}_{\mathbf{x}\sim S_0}\left[(\widehat{r}_t(\mathbf{x}) - r_t^*(\mathbf{x}))^2\right] \leq \widetilde{L}_t^\psi(\widehat{r}_t) - \widetilde{L}_t^\psi(r_t^*) + U_t, \tag{18}$$

where the term $U_t$ is defined as

$$U_t = \mathbb{E}_{\mathbf{x}\sim\mathcal{D}_0}\left[\partial^2\psi\big(r_t^*(\mathbf{x})\big)r_t^*(\mathbf{x})\big(\widehat{r}_t(\mathbf{x}) - r_t^*(\mathbf{x})\big)\right] - \mathbb{E}_{\mathbf{x}\sim S_0}\left[\partial^2\psi\big(r_t^*(\mathbf{x})\big)r_t^*(\mathbf{x})\big(\widehat{r}_t(\mathbf{x}) - r_t^*(\mathbf{x})\big)\right].$$

It remains to analyze the generalization gap $U_t$ for every iteration. The main challenge is that the learned model $\widehat{r}_t$ depends on the initial dataset $S_0$ and the generalization error is related to the complexity of the hypothesis of the density ratio estimator. In the following lemma, we use the covering number [65, Chapter 27] to measure the complexity of the hypothesis space. It is possible to provide an upper bound of $U_t$ with other advanced complexity measure.

**Lemma 1.** *Let $\mathcal{H}_r \triangleq \{r : \mathcal{X} \to \mathbb{R} \mid \|r\|_\infty \leq B_r'\}$ be a hypothesis space of the density ratio function whose covering number is $N(\mathcal{H}_r, \varepsilon, \|\cdot\|_\infty)$. Then, for all model $r \in \mathcal{H}_r$ and $\delta \in (0,1)$, we have the following upper bound*

$$U_t = \mathbb{E}_{\mathbf{x}\sim\mathcal{D}_0}\left[\partial^2\psi\big(r_t^*(\mathbf{x})\big)r_t^*(\mathbf{x})\big(r(\mathbf{x}) - r_t^*(\mathbf{x})\big)\right] - \mathbb{E}_{\mathbf{x}\sim S_0}\left[\partial^2\psi\big(r_t^*(\mathbf{x})\big)r_t^*(\mathbf{x})\big(r(\mathbf{x}) - r_t^*(\mathbf{x})\big)\right]$$
$$\leq \frac{C_r(\mu)\log\left(2N(\mathcal{H}_r, \varepsilon, \|\cdot\|_\infty)/\delta\right)}{N_0} + \frac{\mu}{4}\mathbb{E}_{S_0}\left[(r(\mathbf{x}) - r_t^*(\mathbf{x}))^2\right] + 2A_r\varepsilon + \frac{\mu\varepsilon^2}{4},$$

*with probability at least $1 - \delta$ for any $\varepsilon > 0$, where $C_r(\mu) = 4A_rB_r/3 + 16A_r^2/\mu + 3\mu B_r^2 = \mathcal{O}(\max\{\mu, 1/\mu\})$ with constants $A_r = |\max_{\mathbf{x}\in\mathcal{X}} \partial^2\psi(r_t^*(\mathbf{x}))r_t^*(\mathbf{x})|$, and $B_r = \max\{B, B_r'\}$.*

Combining (18) and Lemma 1 with the hypothesis space $\mathcal{H}_r = \mathcal{H}_\theta \triangleq \{\mathbf{x} \mapsto h(\mathbf{x}, \theta) \mid \theta \in \Theta\}$ and rearranging the terms yields,

$$\frac{\mu}{4}\mathbb{E}_{\mathbf{x}\sim S_0}\left[(\widehat{r}_t(\mathbf{x}) - r_t^*(\mathbf{x}))^2\right] \leq \widetilde{L}_t^\psi(\widehat{r}_t) - \widetilde{L}_t^\psi(r_t^*) + \frac{C_r(\mu)\log\left(2N(\mathcal{H}_\theta, \varepsilon, \|\cdot\|_\infty)/\delta\right)}{N_0} + 2A_r\varepsilon + \frac{\mu\varepsilon^2}{4}.$$

Further choosing $\varepsilon = 1/T$ and taking the summation of the $T$ iterations, we have

$$\frac{\mu}{4}\sum_{t=1}^{T}\mathbb{E}_{\mathbf{x}\sim S_0}\left[(\widehat{r}_t(\mathbf{x}) - r_t^*(\mathbf{x}))^2\right]$$
$$\leq \sum_{t=1}^{T}\widetilde{L}_t^\psi(\widehat{r}_t) - \sum_{t=1}^{T}\widetilde{L}_t^\psi(r_t^*) + \frac{C_r(\mu)T\log\left(2TN(\mathcal{H}_\theta, 1/T, \|\cdot\|_\infty)/\delta\right)}{N_0} + D_r \tag{19}$$

where $D_r = 2A_r + \mu/(4T) = \mathcal{O}(1)$ is a constant. Then, we proceed to bound the covering number of the hypothesis space $\mathcal{H}_\theta$. Let $\theta, \theta' \in \Theta$ be the corresponding parameters for the two density ratio function $r, r' \in \mathcal{H}_\theta$. Then, we can show that for any $\|\theta - \theta'\|_2 \leq \varepsilon$, we have

$$\|r - r'\|_\infty = \max_{\mathbf{x}\in\mathcal{X}}|h(\mathbf{x}, \theta) - h(\mathbf{x}, \theta')| \leq G_h\|\theta - \theta'\|_2,$$

where $G_h = \max_{\mathbf{x}\in\mathcal{X}, \theta\in\Theta}\|\nabla h(\mathbf{x}, \theta)\|_2$ is the Lipschitz continuity constant. Then, we can check that the covering number of $\mathcal{H}_\theta$ in terms of $\|\cdot\|_\infty$ can be bounded by that of $\Theta$ in terms of $\|\cdot\|_2$ as

$$N(\mathcal{H}_\theta, 1/T, \|\cdot\|_\infty) \leq N(\Theta, 1/(G_hT), \|\cdot\|_2) \leq (3SG_hT)^d. \tag{20}$$

In the above, the last inequality holds because the parameter space $\Theta$ is essentially a $L_2$-ball with radius $S$, whose covering number is bounded by $(3S/\varepsilon)^d$ [66]. Then, plugging (20) into (19), we arrive

$$\sum_{t=1}^{T}\mathbb{E}_{\mathbf{x}\sim S_0}\left[(\widehat{r}_t(\mathbf{x}) - r_t^*(\mathbf{x}))^2\right] \leq \frac{4}{\mu}\sum_{t=1}^{T}\left(\widetilde{L}_t^\psi(\widehat{r}_t) - \widetilde{L}_t^\psi(r_t^*)\right) + \frac{4dTC_r(\mu)\log\left(6SG_hT/\delta\right)}{\mu N_0} + \frac{4D_r}{\mu}.$$

Then, we complete the proof by plugging this inequality into (14),

$$\sum_{t=1}^{T}\mathbb{E}_{\mathbf{x}\sim S_0(\mathbf{x})}[|\widehat{r}_t(\mathbf{x}) - r_t^*(\mathbf{x})|]$$

$$\leq \sqrt{\frac{4T}{\mu}\left(\sum_{t=1}^{T}L_t^{\psi}(\widehat{r}_t) - \sum_{t=1}^{T}L_t^{\psi}(r_t^*)\right) + \frac{4dT^2C_r(\mu)\log\left(6SG_hT/\delta\right)}{\mu N_0} + \frac{4TD_r}{\mu}}$$

$$= \sqrt{\frac{4T}{\mu}\max\left\{\sum_{t=1}^{T}L_t^{\psi}(\widehat{r}_t) - \sum_{t=1}^{T}L_t^{\psi}(r_t^*), 0\right\}} + \mathcal{O}\left(\max\left\{1, \frac{1}{\mu}\right\} \cdot \frac{T\sqrt{d}\log(T/\delta)}{\sqrt{N_0}}\right).$$

$\square$

### C.4 Useful Lemmas

**Lemma 2.** *Let* $\widetilde{R}_t(\mathbf{w}) = \mathbb{E}_{(\mathbf{x},y)\sim S_0}\left[r_t^*(\mathbf{x})\ell(\mathbf{w}^\top\mathbf{x}, y)\right]$. *With probability at least* $1 - \delta$ *over the drawn of* $S_0$, *all predictions* $\mathbf{w} \in \mathcal{W}$ *satisfy*

$$|R_t(\mathbf{w}) - \widetilde{R}_t(\mathbf{w})| \leq \frac{2BGDR}{\sqrt{N_0}} + 4BL\sqrt{\frac{2\ln(4/\delta)}{N_0}}.$$

*Proof of Lemma 2.* Since the risk function $\widetilde{R}_t(\mathbf{w})$ is bounded by $BL$ for all $\mathbf{w} \in \mathcal{W}$, the standard analysis on generalization bound with Rademacher complexity (e.g. [65, Theorem 26.5]) shows that

$$|R_t(\mathbf{w}) - \widetilde{R}_t(\mathbf{w})| \leq 2\widehat{\mathfrak{R}}_{S_0}(\mathcal{L}_t) + 4BL\sqrt{\frac{2\ln(4/\delta)}{N_0}}$$

for any $\mathbf{w} \in \mathcal{W}$. In above, the function space $\mathcal{L}_t$ is defined as $\mathcal{L}_t = \{(\mathbf{x}, y) \mapsto r_t^*(\mathbf{x})\ell(\mathbf{w}^\top\mathbf{x}, y) \mid \mathbf{w} \in \mathcal{W}\}$ and $\widehat{\mathfrak{R}}_{S_0}(\mathcal{L}_t) = \frac{1}{N_0}\mathbb{E}_{\boldsymbol{\sigma}\sim\{\pm 1\}^{N_0}}\left[\sup_{\mathbf{w}\in\mathcal{W}}\sum_{n=1}^{N_0}\sigma_i r_t(\mathbf{x}_n)\ell(\mathbf{w}^\top\mathbf{x}_n, y_n)\right]$ is the empirical Rademacher complexity of $\mathcal{L}_t$.

Note that the function $\phi_n : \mathbb{R} \to \mathbb{R}$ defined as $\phi_n(z) = r_t^*(\mathbf{x}_n)\ell(z, y_n)$ is $BG$-Lipschitz, such that $|\phi_n(\mathbf{w}_1^\top\mathbf{x}_n) - \phi_n(\mathbf{w}_2^\top\mathbf{x}_n)| \leq BG|(\mathbf{w}_1 - \mathbf{w}_2)^\top\mathbf{x}_n|$ for any $\mathbf{w}_1, \mathbf{w}_2 \in \mathcal{W}$. Then, according to the Talagrand's lemma [65, Lemma 26.9], we have

$$\widehat{\mathfrak{R}}_{S_0}(\mathcal{L}_t) \leq BG\widehat{\mathfrak{R}}_{S_0}(\widetilde{\mathcal{W}}),$$

where $\widetilde{\mathcal{W}} = \{\mathbf{x} \mapsto \mathbf{w}^\top\mathbf{x} \mid \mathbf{w} \in \mathcal{W}\}$ is a hypothesis space for the linear functions. By [65, Lemma 26.10], the Rademacher complexity of the linear function class is bounded by

$$\widehat{\mathfrak{R}}_{S_0}(\widetilde{\mathcal{W}}) \leq \frac{D\max_{\mathbf{x}_n\in S_0}\|\mathbf{x}_n\|_2}{\sqrt{N_0}} \leq \frac{DR}{\sqrt{N_0}}.$$

We complete the proof by combining the above displayed inequalities. $\square$

*Proof of Lemma 1.* We begin with the analysis for the generalization gap for a fixed model $r \in \mathcal{H}_r$. For notation simplicity, we denote by

$$f_r(\mathbf{x}) = \partial^2\psi\left(r_t^*(\mathbf{x})\right)r_t^*(\mathbf{x})\left(r(\mathbf{x}) - r_t^*(\mathbf{x})\right),$$

and let the random variable $Z_i = \mathbb{E}_{\mathbf{x}\sim\mathcal{D}_0}[f_r(\mathbf{x})] - f_r(\mathbf{x}_i)$ for any $\mathbf{x}_i \in S_0$. Since $\mathbf{x}_i$ is i.i.d. sampled from $\mathcal{D}_0$ and $f_r$ is independent of $S_0$, we have $\mathbb{E}[Z_i] = 0$ and $|Z_i| \leq 2A_rB_r$ with the constants $A_r = \max_{\mathbf{x}\in\mathcal{X}}|\partial^2\psi(r_t^*(\mathbf{x}))r_t^*(\mathbf{x})|$ and $B_r = \max\{B, \max_{\mathbf{x}\in\mathcal{X}, r\in\mathcal{H}_r}|r(\mathbf{x})|\}$. Besides, the variance of $Z_i$ is bounded by

$$\mathbb{E}[Z_i^2] = \mathbb{E}_{\mathbf{x}\sim\mathcal{D}_0}[(f_r(\mathbf{x}))^2] - (\mathbb{E}_{\mathbf{x}\sim\mathcal{D}_0}[f_r(\mathbf{x})])^2 \leq \mathbb{E}_{\mathbf{x}\sim\mathcal{D}_0}\left[(f_r(\mathbf{x}))^2\right],$$

Then, by the Bernstein's inequality [65, Lemma B.9], with probability at least $1 - \delta$, we have

$$\mathbb{E}_{\mathbf{x}\sim D_0}[f_r(\mathbf{x})] - \mathbb{E}_{\mathbf{x}\sim S_0}[f_r(\mathbf{x})] \leq \frac{4A_rB_r\log(1/\delta)}{3N_0} + \sqrt{\frac{2\log(1/\delta)\mathbb{E}_{\mathcal{D}_0}[(f_r(\mathbf{x}))^2])}{N_0}}$$

$$\leq \left(\frac{4A_rB_r}{3} + \frac{16A_r^2}{\mu}\right)\frac{\log(1/\delta)}{N_0} + \frac{\mu}{8A_r^2}\mathbb{E}_{\mathcal{D}_0}[(f_r(\mathbf{x}))^2]$$

$$\leq \left(\frac{4A_r B_r}{3} + \frac{16A_r^2}{\mu}\right)\frac{\log(1/\delta)}{N_0} + \frac{\mu}{8}\mathbb{E}_{\mathcal{D}_0}\left[(r(\mathbf{x}) - r_t^*(\mathbf{x}))^2\right], \tag{21}$$

where the first inequality is due to the AM-GM inequality. The second inequality is due to the fact that $|f_r(\mathbf{x})| \leq |\partial^2\psi(r_t^*(\mathbf{x}))r_t^*(\mathbf{x})| \cdot |r(\mathbf{x}) - r_t^*(\mathbf{x})| \leq A_r|r(\mathbf{x}) - r_t^*(\mathbf{x})|$ for any $\mathbf{x} \in \mathcal{X}$.

Then, since $r$ is a fixed model in $\mathcal{H}_r$, we can further bound $\mathbb{E}_{\mathcal{D}_0}\left[(r(\mathbf{x}) - r_t^*(\mathbf{x}))^2\right]$ by $\mathbb{E}_{S_0}\left[(r(\mathbf{x}) - r_t^*(\mathbf{x}))^2\right]$ by applying the Bernstein's inequality to the random variable $Y_i = (r(\mathbf{x}_i) - r_t^*(\mathbf{x}_i))^2/(4B_r^2)$. Specifically, after checking $\mathbb{E}[Y_i] = \mathbb{E}_{\mathcal{D}_0}\left[(r(\mathbf{x}) - r_t^*(\mathbf{x}))^2\right]/(4B_r^2)$ and $Y_i \in [0, 1]$, a direct application of [65, Lemma B.10] implies

$$\mathbb{E}_{\mathcal{D}_0}\left[(r(\mathbf{x}) - r_t^*(\mathbf{x}))^2\right]$$
$$\leq \mathbb{E}_{S_0}\left[(r(\mathbf{x}) - r_t^*(\mathbf{x}))^2\right] + \frac{16B_r^2\log(1/\delta)}{N_0} + \sqrt{\frac{8B_r^2\mathbb{E}_{S_0}\left[(r(\mathbf{x}) - r_t^*(\mathbf{x}))^2\right]\log(1/\delta)}{N_0}}$$
$$\leq \frac{24B_r^2\log(1/\delta)}{N_0} + \mathbb{E}_{S_0}\left[(r(\mathbf{x}) - r_t^*(\mathbf{x}))^2\right], \tag{22}$$

where the last inequality is due to the AM-GM inequality. Plugging (22) into (21), for a fixed $r \in \mathcal{H}_r$, we have

$$\mathbb{E}_{\mathbf{x}\sim D_0}[f_r(\mathbf{x})] - \mathbb{E}_{\mathbf{x}\sim S_0}[f_r(\mathbf{x})]$$
$$\leq \left(\frac{4A_r B_r}{3} + \frac{16A_r^2}{\mu} + 3\mu B_r^2\right)\frac{\log(2/\delta)}{N_0} + \frac{\mu}{8}\mathbb{E}_{S_0}\left[(r(\mathbf{x}) - r_t^*(\mathbf{x}))^2\right],$$

with probability at least $1 - \delta$.

Then, based on the notion of covering number, we show a union bound of $U_t$ over all $r \in \mathcal{H}_r$. Let $\mathcal{N}(\mathcal{H}_r, \varepsilon, \|\cdot\|_\infty)$ be the $\varepsilon$-net of $\mathcal{H}_r$ such that for any $r \in \mathcal{H}_r$ one can find a $r' \in \mathcal{N}(\mathcal{H}_r, \varepsilon, \|\cdot\|_\infty)$ satisfying $\|r - r'\|_\infty \leq \varepsilon$. The covering number $N(\mathcal{H}_r, \varepsilon, \|\cdot\|_\infty)$ is defined as the minimal cardinality of an $\varepsilon$-net of $\mathcal{H}_r$. By taking the union bound over all $r' \in \mathcal{N}(\mathcal{H}_r, \varepsilon, \|\cdot\|_\infty)$, with probability at least $1 - \delta$, the following holds for any $r' \in \mathcal{N}(\mathcal{H}_r, \varepsilon, \|\cdot\|_\infty)$,

$$\mathbb{E}_{\mathbf{x}\sim D_0}[f_{r'}(\mathbf{x})] - \mathbb{E}_{\mathbf{x}\sim S_0}[f_{r'}(\mathbf{x})]$$
$$\leq \frac{C_r(\mu)\log\left(\frac{2N(\mathcal{H}_r, \varepsilon, \|\cdot\|_\infty)}{\delta}\right)}{N_0} + \frac{\mu}{8}\mathbb{E}_{S_0}\left[(r'(\mathbf{x}) - r_t^*(\mathbf{x}))^2\right], \tag{23}$$

where $C_r(\mu) = 4A_r B_r/3 + 16A_r^2/\mu + 3\mu B_r^2 = \mathcal{O}(\max\{\mu, 1/\mu\})$. Then, for any model $r \in \mathcal{H}_r$, we can show

$$\mathbb{E}_{\mathbf{x}\sim D_0}[f_r(\mathbf{x})] - \mathbb{E}_{\mathbf{x}\sim S_0}[f_r(\mathbf{x})]$$
$$= \mathbb{E}_{\mathbf{x}\sim D_0}[f_{r'}(\mathbf{x})] - \mathbb{E}_{\mathbf{x}\sim S_0}[f_{r'}(\mathbf{x})] + \mathbb{E}_{\mathbf{x}\sim \mathcal{D}_0}[f_r(\mathbf{x}) - f_{r'}(\mathbf{x})] - \mathbb{E}_{\mathbf{x}\sim S_0}[f_r(\mathbf{x}) - f_{r'}(\mathbf{x})]$$
$$\leq \mathbb{E}_{\mathbf{x}\sim D_0}[f_{r'}(\mathbf{x})] - \mathbb{E}_{\mathbf{x}\sim S_0}[f_{r'}(\mathbf{x})] + 2\varepsilon A_r$$
$$\overset{(23)}{\leq} \frac{C_r(\mu)\log\left(2N(\mathcal{H}_r, \varepsilon, \|\cdot\|_\infty)/\delta\right)}{N_0} + \frac{\mu}{8}\mathbb{E}_{S_0}\left[(r'(\mathbf{x}) - r_t^*(\mathbf{x}))^2\right] + 2A_r\varepsilon$$
$$\leq \frac{C_r(\mu)\log\left(2N(\mathcal{H}_r, \varepsilon, \|\cdot\|_\infty)/\delta\right)}{N_0} + \frac{\mu}{4}\mathbb{E}_{S_0}\left[(r(\mathbf{x}) - r_t^*(\mathbf{x}))^2\right] + 2A_r\varepsilon + \frac{\mu\varepsilon^2}{4},$$

where the first inequality is due to the property of the $\varepsilon$-net such that

$$\mathbb{E}_{\mathbf{x}\sim\mathcal{D}_0}[f_r(\mathbf{x}) - f_{r'}(\mathbf{x})] \leq \mathbb{E}_{\mathbf{x}\sim\mathcal{D}_0}[|\partial^2\psi(r_t^*(\mathbf{x}))r'(\mathbf{x})| \cdot |r(\mathbf{x}) - r_t^*(\mathbf{x})|] \leq A_r\|r - r'\|_\infty \leq A_r\varepsilon.$$

The last inequality is by

$$\frac{\mu}{8}\mathbb{E}_{S_0}\left[(r'(\mathbf{x}) - r_t^*(\mathbf{x}))^2\right] \leq \frac{\mu}{4}\mathbb{E}_{S_0}\left[(r(\mathbf{x}) - r_t^*(\mathbf{x}))^2\right] + \frac{\mu}{4}\mathbb{E}_{S_0}\left[(r(\mathbf{x}) - r'(\mathbf{x}))^2\right]$$
$$\leq \frac{\mu}{4}\mathbb{E}_{S_0}\left[(r(\mathbf{x}) - r_t^*(\mathbf{x}))^2\right] + \frac{\mu}{4}\|r - r'\|_\infty^2 \leq \frac{\mu}{4}\mathbb{E}_{S_0}\left[(r(\mathbf{x}) - r_t^*(\mathbf{x}))^2\right] + \frac{\mu\varepsilon^2}{4}.$$

Thus, we complete the proof. $\qquad\square$

---

**Algorithm 1** Base-learner $\mathcal{E}_i$

---

**Input:** Active interval $I_i = [s_i, e_i]$, regularization parameter $\lambda$ and step size $\eta$, feasible domain $\Theta$.

1: Initialize the model $\widehat{\theta}_{s_i,i}$ as any point in the domain $\Theta$ and the matrix $A_{s_i-1} = \lambda I_d$.
2: **for** $t \in I_i = [s_i, e_i]$ **do**
3:   Submit the model $\widehat{\theta}_{t,i}$ to the meta-learner.
4:   Update the matrix by $A_{t,i} = A_{t-1,i} + \nabla \widehat{L}_t(\widehat{\theta}_{t,i}) \nabla \widehat{L}_t(\widehat{\theta}_{t,i})^\top$.
5:   Update the model for next iteration by

$$\widehat{\theta}_{t+1,i} = \Pi_\Theta^{A_{t,i}} \left[ \widehat{\theta}_{t,i} - \gamma A_{t,i}^{-1} \nabla \widehat{L}_t(\widehat{\theta}_{t,i}) \right],$$

   where $\Pi_\Theta^{A_{t,i}}[\nu] = \arg\min_{\theta \in \Theta} \|\theta - \nu\|_{A_{t,i}}$ is the projection operator.
6: **end for**

---

## D Omitted Details for Section 4

This section presents the omitted details for Section 4. We will first introduce the algorithm details for our online ensemble algorithm, characterized by its meta-base structure, in Appendix D.1. Then, we present the theoretical guarantees alongside their proofs for the base-learner (Lemma 3) and meta-learner (Lemma 10) in Appendix D.2 and Appendix D.3 respectively. These elements collectively contribute to the dynamic regret guarantee of the overall algorithm. The dynamic regret of the online ensemble algorithm (Theorem 2) is presented in Appendix D.4. We provide the proof for the average excess risk (Theorem 3) in Appendix D.5. Further discussions regarding the tightness of the bound and alternative choices for the Bregman divergence function are located in Appendices D.6 and D.7.

### D.1 Omitted Algorithm Details

Here, we introduce the algorithmic details for the online ensemble method omitted in Section 4.1.

**Interval Schedules.** We run multiple based learners on the geometric covering:

$$\mathcal{C} = \bigcup_{k \in \mathbb{N} \cup \{0\}} \mathcal{C}_k \quad \text{and} \quad \mathcal{C}_k = \left\{ [i \cdot 2^k, (i+1) \cdot 2^k - 1] \mid i \in \mathbb{N} \text{ and } i \cdot 2^k \leq T \right\} \tag{24}$$

and then employ a meta-learner to combine them by weights. For any interval $I_i = [s_i, e_i] \in \mathcal{C}$, which starts at time $t = s_i$ and end at $t = e_i$, we denote by $\mathcal{E}_i$ the base-learner running over $I_i$.

**Base-learner.** The base-learner $\mathcal{E}_i$ updates his/her model with the ONS algorithm [44] in the associated active period $I_i$. The algorithmic details of the base-learner are summarized in Algorithm 1. At every iteration $t \in I_i$, the base-learner $\mathcal{E}_i$ will first submit his/her model $\widehat{\theta}_{t,i}$ to the meta-learner and then update the model for the next iteration by Line 4 and Line 5 as shown in Algorithm 1.

**Meta-learner.** The meta-learner aggregates the predictions of each active base-learner by a weighted combination scheme. To obtain the weight $p_{t,i}$ for each active base-learner $\mathcal{E}_i$, we maintain three terms $m_{t,i}$, $v_{t,i}$ and $\varepsilon_{t,i}$ for each base-learner $\mathcal{E}_i$ when he/she is active, i.e., $t \in I_i$. Let $g_t(\theta) = \langle \nabla \widehat{L}_t(\widehat{\theta}_t), \theta - \widehat{\theta}_t \rangle$ be a linearized loss for the original function $\widehat{L}_t(\theta)$. The first term is defined by

$$m_{t,i} \triangleq \frac{g_t(\widehat{\theta}_t) - g_t(\widehat{\theta}_{t,i})}{SR} = \frac{\langle \nabla \widehat{L}_t(\widehat{\theta}_t), \widehat{\theta}_t - \widehat{\theta}_{t,i} \rangle}{SR},$$

which measures the performance gap between the meta-learner's prediction $\widehat{\theta}_t$ and the $i$-th base-learner's prediction $\widehat{\theta}_{t,i}$ over the linearized loss $g_t(\theta)$. The second term $v_{t,i}$ is the "potential" of the base-learner $\mathcal{E}_i$, which measures the historical performance of $\mathcal{E}_i$ over the interval. When invoking the base-learner $\mathcal{E}_i$ at $t = s_i$, we initialize $v_{s_i-1,i} = 1/K$ and the term is updated by

$$v_{t,i} = \left( v_{t-1,i} \cdot \left( 1 + \varepsilon_{t-1,i} m_{t,i} \right) \right)^{\frac{\varepsilon_{t,j}}{\varepsilon_{t-1,j}}}, \forall t \in I_i. \tag{25}$$

The last term $\varepsilon_{t,i}$ is the learning rate, which is set as

$$\varepsilon_{t,i} = \min \left\{ \frac{1}{2}, \sqrt{\frac{\ln K}{1 + \sum_{\tau=s_i}^t m_{\tau,i}^2}} \right\}. \tag{26}$$

---

**Algorithm 2** Meta-learner

---

**Input:** Interval set for base-learners $\mathcal{C}$.
1: Iitialize the set for active base-learner $\mathcal{A}_0 = \emptyset$ and let $K = |\mathcal{C}|$.
2: **for** $t = 1, \ldots, T$ **do**
3:  **for** $I_i = [s_i, e_i] \in \mathcal{C}$ **do**
4:    **if** $s_i$ is equal to $t$ **then**
5:      Create a base-leaner $\mathcal{E}_i$ by Algorithm 1 and initialize the potential $v_{s_i-1,i} = 1/K$ and step size $\varepsilon_{s_i-1,i} = \min\{1/2, \sqrt{\ln K}\}$.
6:      Add the index to the active set $\mathcal{A}_t = \mathcal{A}_{t-1} \cup \{i\}$.
7:    **else if** $e_i + 1$ is equal to $t$ **then**
8:      Remove the index from the active set $\mathcal{A}_t = \mathcal{A}_{t-1}/\{i\}$.
9:    **end if**
10:  **end for**
11:  Receive the $\widehat{\theta}_{i,t}$ from active base-learners $\mathcal{E}_i \in \mathcal{A}_t$ and update their weights $p_{t,i}$ by (27).
12:  Submit the model by $\widehat{\theta}_t = \sum_{i \in \mathcal{A}_t} p_{t,i} \widehat{\theta}_{t,i}$.
13:  **for** $i \in \mathcal{A}_t$ **do**
14:    Update the step size $\varepsilon_{t+1,i}$ for all current active base-learner by (26).
15:    Update the potential $v_{t+1,i}$ for all current active base-learner by (25).
16:  **end for**
17: **end for**

---

After obtaining the $v_{t,i}$ and $\varepsilon_{t,i}$ for each active base-learner, the weight for the next iteration $t + 1$ is updated by

$$p_{t+1,i} = \frac{\varepsilon_{t,i} v_{t,i}}{\sum_{i \in \mathcal{A}_{t+1}} \varepsilon_{t,i} v_{t,i}}, \text{ for all } i \in \mathcal{A}_{t+1} . \tag{27}$$

The algorithm is summarized in Algorithm 2. At the beginning of each iteration $t$, the algorithm will update the index set for the active base-learner (Line 3-Line 10). Then, the meta-learner's prediction is generated by the weighted combination rule (Line 11). After that, the meta-learner will update the potential $v_{t+1,i}$ and the step $\varepsilon_{t+1,i}$ for the next iteration.

### D.2 Regret Guarantee for Base-Algorithm (Lemma 3)

By suitable configurations of the parameters $\gamma, \lambda > 0$, we show that the ONS algorithm enjoys the following dynamic regret guarantees,

**Lemma 3.** *Let $\theta_t^* = \arg\min_{\theta \in \mathbb{R}^d} L_t(\theta)$ and all the minimizer $\theta_t^*$'s lie in the interior of $\Theta$. Set $\gamma = 6(1 + \beta)$ and $\lambda = 1$. Then, with probability at least $1 - \delta$, the ONS running on the interval $I_i = [s_i, e_i] \subseteq [T]$ ensures*

$$\sum_{t \in I_i} \widetilde{L}_t(\widehat{\theta}_{t,i}) - \sum_{t \in I_i} \widetilde{L}_t(\theta_t^*) \leq \mathcal{O}\left( d\beta \log\left(|I_i|/\delta\right) + |I_i|\left(V_{I_i}^\theta\right)^2 + \frac{|I_i| \log(dT/\delta)}{N_0} \right)$$

*where $V_{I_i}^\theta = \sum_{t=s_i+1}^{e_i} \|\theta_t^* - \theta_{t-1}^*\|_2$ is the path length measuring the fluctuation of the minimizers.*

#### D.2.1 Main Proof

This section provides the proof of the ONS algorithm. Since the analysis of the single base-learner is independent from the online ensemble structure, for notation simplicity, we omit the subscribe of the base-learner in the analysis. Specifically, we abbreviate the prediction $\widehat{\theta}_{t,i}$, matrix $A_{t,i}$ and the interval $I_i$ as $\widehat{\theta}_t$, matrix $A_t$ and the interval $I$ respectively.

*Proof of Lemma 3.* We begin the proof of the dynamic regret with a lemma on the static regret of the ONS algorithm, whose proof is proved in Appendix D.2.2. For notation simplicity, we omit the subscribes of the base-learner, where the prediction $\theta_{t,i}$, matrix $A_{t,i}$ and interval $I_i$ is abbreviated as $\theta_t$, $A_t$ and $I$, respectively.

**Lemma 4.** *Set $\gamma = 6(1 + \beta)$ and $\lambda = 1$. With probability at least $1 - 2\delta$, for any $\delta \in (0, 1]$, the ONS algorithm* (8) *running on the interval $I = [s, e] \subseteq [T]$ ensures*

$$\sum_{t \in I} \widetilde{L}_t(\widehat{\theta}_t) - \sum_{t \in I} \widetilde{L}_t(\theta) \leq C_I^{(1)}(\delta) = \mathcal{O}\left(d\beta \log(|I|/\delta)\right)$$

*for any $\theta \in \Theta$, where $C_I^{(1)}(\delta) = 12d\beta' \log\left(1 + |I|R^2\right) + \frac{S^2}{3\beta'} + (48\beta' + \frac{6S^2R^2}{\beta'})\log\frac{\sqrt{2|I|+1}}{\delta} + (4SR + \frac{S^2R^2}{\beta'})\sqrt{\log\frac{2|I|+1}{\delta^2}}$ and $\beta' = \beta + 1$.*

Then, we can decompose the dynamic regret as follows,

$$\sum_{t \in I} \widetilde{L}_t(\widehat{\theta}_t) - \sum_{t \in I} \widetilde{L}_t(\theta_t^*) = \sum_{t \in I} \widetilde{L}_t(\widehat{\theta}_t) - \sum_{t \in I} \widetilde{L}_t(\bar{\theta}_I^*) + \sum_{t \in I} \widetilde{L}_t(\bar{\theta}_I^*) - \sum_{t \in I} \widetilde{L}_t(\theta_t^*)$$

$$\leq C_I^{(1)}(\delta) + \sum_{t \in I} \widetilde{L}_t(\bar{\theta}_I^*) - \sum_{t \in I} \widetilde{L}_t(\theta_t^*), \tag{28}$$

where $\bar{\theta}_I^* = \frac{1}{|I|}\sum_{t \in I}\theta_t^*$ is an averaged prediction of the function minimizers over $T$ rounds. Under the condition all function minimizers lie in the interior of $\Theta$, we have $\bar{\theta}_I^* \in \Theta$. As a consequence, the last inequality holds due to Lemma 4.

Then, we proceed to bound the last two terms of the R.H.S. of the above inequality. Since the first-order derivative of the sigmoid function $\sigma'(z) = \exp(-z)/(1 + \exp(-z))^2 \leq 1/4$ for any $z \in \mathbb{R}$. Let $I_d \in \mathbb{R}^{d \times d}$ be the identity matrix. we can check $\widetilde{L}_t$ is a $R^2/4$-smooth function such that $\nabla^2 \widetilde{L}_t(\theta) = \frac{1}{2}\mathbb{E}_{\mathbf{x} \sim S_0}[\sigma'(\mathbf{x}^\top \theta)\phi(\mathbf{x})\phi(\mathbf{x})^\top] + \frac{1}{2}\mathbb{E}_{\mathbf{x} \sim \mathcal{D}_t}[\sigma'(\mathbf{x}^\top \theta)\phi(\mathbf{x})\phi(\mathbf{x})^\top] \preccurlyeq \frac{R^2}{4}I_d$ for any $\theta \in \Theta$. Then, with probability at least $1 - 2\delta$, we have

$$\widetilde{L}_t(\bar{\theta}_I^*) - \widetilde{L}_t(\theta_t^*) \leq \langle \nabla \widetilde{L}_t(\theta_t^*), \bar{\theta}_I^* - \theta_t^* \rangle + \frac{R^2}{8}\|\bar{\theta}_I^* - \theta_t^*\|_2^2$$

$$\leq \frac{8}{R^2}\|\nabla \widetilde{L}_t(\theta_t^*)\|_2^2 + \frac{R^2}{4}\|\bar{\theta}_I^* - \theta_t^*\|_2^2$$

$$\leq \frac{16\ln((d+1)/\delta)}{N_0} + \frac{R^2}{4}\|\bar{\theta}_I^* - \theta_t^*\|_2^2 + o\left(\frac{\log(d/\delta)}{N_0}\right)$$

$$\leq \frac{16\ln((d+1)/\delta)}{N_0} + \frac{R^2}{4}(V_I^\theta)^2 + o\left(\frac{\log(d/\delta)}{N_0}\right) \tag{29}$$

where $V_I^\theta = \sum_{t=s+1}^{e}\|\theta_{t-1}^* - \theta_t^*\|_2$ with $s$ being the start time of the interval $I$ and $e$ being the ending time. In above, the first inequality is due to the smoothness of the loss function and the second inequality comes from the AM-GM inequality. The third inequality is due to Lemma 5. The last inequality can be obtained by the definition of $\bar{\theta}_I^*$ such that

$$\|\bar{\theta}_I^* - \theta_t^*\|_2^2 = \left(\left\|\frac{1}{|I|}\sum_{\tau \in I}\theta_\tau^* - \theta_t^*\right\|_2\right)^2 \leq \left(\frac{1}{|I|}\sum_{\tau \in I}\|\theta_\tau^* - \theta_t^*\|_2\right)^2 \leq (V_I^\theta)^2.$$

Plugging (29) into (28) and taking the summation over $T$ rounds, we obtain

$$\sum_{t \in I} \widetilde{L}_t(\widehat{\theta}_t) - \sum_{t \in I} \widetilde{L}_t(\theta)$$

$$\leq C_I^{(1)}(\delta) + \frac{16|I|\ln((d+1)|I|/\delta)}{N_0} + \frac{R^2}{4}|I|(V_I^\theta)^2 + o\left(\frac{|I|\log(d|I|/\delta)}{N_0}\right)$$

$$= \mathcal{O}\left(d\beta \log(|I|/\delta) + \frac{|I|\ln(d|I|/\delta)}{N_0} + |I|(V_I^\theta)^2\right)$$

with probability at least $1 - 3\delta$. We have completed the proof. $\qquad\square$

### D.2.2 Useful Lemmas

**Lemma 4.** *Set $\gamma = 6(1 + \beta)$ and $\lambda = 1$. With probability at least $1 - 2\delta$, for any $\delta \in (0, 1]$, the ONS algorithm* (8) *running on the interval $I \subseteq [T]$ ensures,*

$$\sum_{t \in I} \widetilde{L}_t(\widehat{\theta}_t) - \sum_{t \in I} \widetilde{L}_t(\theta) \leq C_I^{(1)}(\delta) = \mathcal{O}\left(d\beta \log(|I|/\delta)\right)$$

*for any $\theta \in \Theta$, where $C_I^{(1)}(\delta) = 12d\beta' \log\left(1 + |I|R^2\right) + \frac{S^2}{3\beta'} + (48\beta' + \frac{6S^2R^2}{\beta'})\log\frac{\sqrt{2|I|+1}}{\delta} + (4SR + \frac{S^2R^2}{\beta'})\sqrt{\log\frac{2|I|+1}{\delta^2}}$ and $\beta' = \beta + 1$.*

*Proof of Lemma 4.* The main difference between the proof of Lemma 4 and the standard analysis for the ONS algorithm (e.g., proof of [13, Theorem 4.3]) is that the later one is for the full information online learning setting, in which the learner can exactly observe the loss function $\widetilde{L}_t$. However, in the continuous covariate shift problem, we can only observe the empirical loss $\widehat{L}_t$, which poses the challenge to bound a generalization gap between the expected loss $\widetilde{L}_t$ and the empirical observation $\widehat{L}_t$ (e.g. term (b) in (35)). We manage to show such a gap can be bounded by $\mathcal{O}(\log T)$ with the self-normalized concentration inequality for the martingale different sequence.

We start with the analysis of the instantaneous regret. By the Taylor's theorem, we have

$$\widehat{L}_t(\widehat{\theta}_t) - \widehat{L}_t(\theta) = \langle \nabla \widehat{L}_t(\widehat{\theta}_t), \widehat{\theta}_t - \theta \rangle - \frac{1}{2}(\widehat{\theta}_t - \theta)^\top \nabla^2 \widehat{L}_t(\boldsymbol{\xi}_t)(\widehat{\theta}_t - \theta), \tag{30}$$

where $\boldsymbol{\xi}_t \in \mathbb{R}^d$ is a certain point on the line connecting $\theta$ and $\widehat{\theta}_t$. In above, the second order derivative $\nabla^2 \widetilde{L}_t$ can be further lower bounded by

$$\nabla^2 \widehat{L}_t(\boldsymbol{\xi}_t) = \frac{1}{2}\mathbb{E}_{\mathbf{x} \sim S_0}[\sigma'(\mathbf{x}^\top \boldsymbol{\xi}_t)\phi(\mathbf{x})\phi(\mathbf{x})^\top] + \frac{1}{2}\mathbb{E}_{\mathbf{x} \sim S_t}[\sigma'(\mathbf{x}^\top \boldsymbol{\xi}_t)\phi(\mathbf{x})\phi(\mathbf{x})^\top]$$

$$\succeq \frac{1}{4(1 + e^{SR})}\left(\mathbb{E}_{\mathbf{x} \sim S_0}[\phi(\mathbf{x})\phi(\mathbf{x})^\top] + \mathbb{E}_{\mathbf{x} \sim S_t}[\phi(\mathbf{x})\phi(\mathbf{x})^\top]\right)$$

$$\succeq \frac{1}{2(1 + \beta)}\nabla \widehat{L}_t(\widehat{\theta}_t)\nabla \widehat{L}_t(\widehat{\theta}_t)^\top, \tag{31}$$

where $\sigma'(z) = \exp(-z)/(1 + \exp(-z))^2$ is the first-order derivative of the sigmoid function $\sigma(z) = 1/(1 + \exp(-z))$ and $A \succeq B$ indicates $A - B$ is a positive semi-definite matrix for any matrix $A, B \in \mathbb{R}^{d \times d}$. The first equality is due to the definition of $\widetilde{L}_t$ and the second inequality is due to Lemma 6.

Plugging (31) into (30), we have

$$\widehat{L}_t(\widehat{\theta}_t) - \widehat{L}_t(\theta) \leq \langle \nabla \widehat{L}_t(\widehat{\theta}_t), \widehat{\theta}_t - \theta \rangle - \frac{1}{4(1 + \beta)}\left(\nabla \widehat{L}_t(\widehat{\theta}_t)^\top(\widehat{\theta}_t - \theta)\right)^2. \tag{32}$$

Let $\mathbb{E}_t[\cdot] = \mathbb{E}_{S_t \sim \mathcal{D}_t}[\cdot \mid S_0, S_1, \ldots, S_{t-1}]$ be the expectation taken over the draw of $S_t$ conditioned on the randomness until round $t - 1$. Since $\widehat{\theta}_t$ is independent of $S_t$, we have $\mathbb{E}_t[\widehat{L}_t(\widehat{\theta}_t)] = \widetilde{L}_t(\widehat{\theta}_t)$ and $\mathbb{E}_t[\nabla \widehat{L}_t(\widehat{\theta}_t)] = \nabla \widetilde{L}_t(\widehat{\theta}_t)$. Taking the expectation over both sides of (32) yields

$$\widetilde{L}_t(\widehat{\theta}_t) - \widetilde{L}_t(\theta)$$

$$\leq \langle \nabla \widetilde{L}_t(\widehat{\theta}_t), \widehat{\theta}_t - \theta \rangle - \frac{1}{4(1 + \beta)}\mathbb{E}_t\left[\left(\nabla \widehat{L}_t(\widehat{\theta}_t)^\top(\widehat{\theta}_t - \theta)\right)^2\right]$$

$$= \langle \nabla \widehat{L}_t(\widehat{\theta}_t), \widehat{\theta}_t - \theta \rangle + \langle \nabla \widetilde{L}_t(\widehat{\theta}_t) - \nabla \widehat{L}_t(\widehat{\theta}_t), \widehat{\theta}_t - \theta \rangle - \frac{1}{4(1 + \beta)}\mathbb{E}_t\left[\left(\nabla \widehat{L}_t(\widehat{\theta}_t)^\top(\widehat{\theta}_t - \theta)\right)^2\right]$$
$$\tag{33}$$

where the first term of the R.H.S. can be further bounded by

$$\langle \nabla \widehat{L}_t(\widehat{\theta}_t), \widehat{\theta}_t - \theta \rangle$$

$$= \langle \nabla \widehat{L}_t(\widehat{\theta}_t), \widehat{\theta}_{t+1} - \theta \rangle + \langle \nabla \widehat{L}_t(\widehat{\theta}_t), \widehat{\theta}_t - \widehat{\theta}_{t+1} \rangle$$

$$\leq \frac{1}{2\gamma}\left(\|\widehat{\theta}_t - \theta\|^2_{A_t} - \|\widehat{\theta}_{t+1} - \theta\|^2_{A_t} - \|\widehat{\theta}_t - \widehat{\theta}_{t+1}\|^2_{A_t}\right) + \langle \nabla \widehat{L}_t(\widehat{\theta}_t), \widehat{\theta}_t - \widehat{\theta}_{t+1}\rangle$$

$$\leq \frac{1}{2\gamma}\left(\|\widehat{\theta}_t - \theta\|^2_{A_t} - \|\widehat{\theta}_{t+1} - \theta\|^2_{A_t} - \|\widehat{\theta}_t - \widehat{\theta}_{t+1}\|^2_{A_t}\right) + 2\gamma\|\nabla \widehat{L}_t(\widehat{\theta}_t)\|^2_{A_t^{-1}} + \frac{1}{2\gamma}\|\widehat{\theta}_t - \widehat{\theta}_{t+1}\|^2_{A_t}$$

$$= 2\gamma\|\nabla \widehat{L}_t(\widehat{\theta}_t)\|^2_{A_t^{-1}} + \frac{1}{2\gamma}\left(\|\widehat{\theta}_t - \theta\|^2_{A_t} - \|\widehat{\theta}_{t+1} - \theta\|^2_{A_t}\right)$$

$$= 2\gamma\|\nabla \widehat{L}_t(\widehat{\theta}_t)\|^2_{A_t^{-1}} + \frac{1}{2\gamma}\left(\|\widehat{\theta}_t - \theta\|^2_{A_{t-1}} - \|\widehat{\theta}_{t+1} - \theta\|^2_{A_t}\right) + \frac{1}{2\gamma}(\nabla \widehat{L}_t(\widehat{\theta}_t)^\top(\widehat{\theta}_t - \theta))^2. \quad (34)$$

In above, the first inequality is due to the update rule of the ONS algorithm and the Bregman proximal inequality [67, Lemma 5]. The second inequality is due to the AM-GM inequality. The last equality is due the definition of $A_t$ such that $\|\widehat{\theta}_t - \theta\|^2_{A_t} - \|\widehat{\theta}_t - \theta\|^2_{A_{t-1}} = (\nabla \widehat{L}_t(\widehat{\theta}_t)^\top(\widehat{\theta}_t - \theta))^2$.

Plugging (34) into (33) and taking the summation over the interval $I = [s, e]$, we have

$$\sum_{t\in I}\widetilde{L}_t(\widehat{\theta}_t) - \sum_{t\in I}\widetilde{L}_t(\theta)$$

$$\leq \underbrace{2\gamma\sum_{t\in I}\|\nabla \widehat{L}_t(\widehat{\theta}_t)\|^2_{A_t^{-1}} + \frac{1}{2\gamma}\|\widehat{\theta}_s - \theta\|^2_{A_s}}_{\texttt{term (a)}} + \frac{1}{2\gamma}\sum_{t\in I}(\nabla \widehat{L}_t(\widehat{\theta}_t)^\top(\widehat{\theta}_t - \theta))^2 \quad (35)$$

$$+ \underbrace{\sum_{t\in I}\langle\nabla \widetilde{L}_t(\widehat{\theta}_t) - \nabla \widehat{L}_t(\widehat{\theta}_t), \widehat{\theta}_t - \theta\rangle - \frac{1}{4(1+\beta)}\sum_{t\in I}\mathbb{E}_t\left[(\nabla \widehat{L}_t(\widehat{\theta}_t)^\top(\widehat{\theta}_t - \theta))^2\right]}_{\texttt{term (b)}}.$$

Now, we proceed to bound term (a) and term (b), respectively. Firstly, according to Lemma 7, we can show term (a) is bounded by

$$\texttt{term (a)} \leq 2d\gamma\log\left(1 + \frac{|I|R^2}{\lambda}\right).$$

Then, we handle term (b) with the Bernstein-type self-normalized concentration inequality [68, Theorem 4] . Specifically, let $\mathbb{E}_t[\cdot] = \mathbb{E}_{S_t\sim\mathcal{D}_t}[\cdot \mid S_0, S_1, \ldots, S_{t-1}]$ be the expectation taken over the draw of $S_t$ conditioned on the randomness until round $t - 1$. Denoting by $Z_t = \langle\nabla \widetilde{L}_t(\widehat{\theta}_t) - \nabla \widehat{L}_t(\widehat{\theta}_t), \widehat{\theta}_t - \theta\rangle$, it is easy to check $\{Z_t\}_{t=s}^e$ is a martingale difference sequence such that $\mathbb{E}_t[Z_t] = 0$ and $|Z_t| \leq 4SR$. Lemma 8 with the choice $\nu = 1/(48(1+\beta))$ indicates,

$$\texttt{term (b)} \leq \frac{1}{24(1+\beta)}\sum_{t\in I}\left((\nabla \widehat{L}_t(\widehat{\theta}_t)^\top(\widehat{\theta}_t - \theta))^2 + \mathbb{E}_t\left[(\nabla \widehat{L}_t(\widehat{\theta}_t)^\top(\widehat{\theta}_t - \theta))^2\right]\right)$$

$$+ 48(1+\beta)\log\frac{\sqrt{2|I|+1}}{\delta} + 4SR\sqrt{\log\frac{2|I|+1}{\delta^2}}. \quad (36)$$

Combining the upper bounds for term (a) and term (b) with (35), we have

$$\sum_{t\in I}\widetilde{L}_t(\widehat{\theta}_t) - \sum_{t\in I}\widetilde{L}_t(\theta)$$

$$\leq 2d\gamma\log\left(1 + \frac{|I|R^2}{\lambda}\right) + \frac{2\lambda S^2}{\gamma} + 48(1+\beta)\log\frac{\sqrt{2|I|+1}}{\delta} + 4SR\sqrt{\log\frac{2|I|+1}{\delta^2}}$$

$$+ \underbrace{\left(\frac{1}{2\gamma} + \frac{1}{24(1+\beta)}\right)\sum_{t\in I}(\nabla \widehat{L}_t(\widehat{\theta}_t)^\top(\widehat{\theta}_t - \theta))^2 - \frac{5}{24(1+\beta)}\sum_{t\in I}\mathbb{E}_t\left[(\nabla \widehat{L}_t(\widehat{\theta}_t)^\top(\widehat{\theta}_t - \theta))^2\right]}_{\texttt{term (c)}}$$

$$(37)$$

Then, setting $\gamma = 6(1+\beta)$, with probability at least $1 - \delta$, we can bound term (c) by

$$\texttt{term (c)} \leq \frac{1}{6(1+\beta)}\left(\frac{3}{4}\sum_{t\in I}\nabla \widehat{L}_t(\widehat{\theta}_t)^\top(\widehat{\theta}_t - \theta))^2 - \frac{5}{4}\sum_{t\in I}\mathbb{E}_t\left[(\nabla \widehat{L}_t(\widehat{\theta}_t)^\top(\widehat{\theta}_t - \theta))^2\right]\right)$$

$$\leq \frac{6S^2R^2}{1+\beta}\log\frac{\sqrt{2|I|+1}}{\delta} + \frac{S^2R^2}{1+\beta}\sqrt{\log\frac{2|I|+1}{\delta^2}},$$

where the last inequality is due to Lemma 9. Combining the upper bound for term (c) and (37) and setting $\lambda = 1$, we have

$$\sum_{t\in I}\widetilde{L}_t(\widehat{\theta}_t) - \sum_{t\in I}\widetilde{L}_t(\theta)$$

$$\leq 12d(1+\beta)\log\left(1+|I|R^2\right) + \frac{S^2}{3\beta+3}$$

$$+ \left(48(\beta+1) + \frac{6S^2R^2}{1+\beta}\right)\log\frac{\sqrt{2|I|+1}}{\delta} + \left(4SR + \frac{S^2R^2}{\beta+1}\right)\sqrt{\log\frac{2|I|+1}{\delta^2}}$$

$$= \mathcal{O}(d\beta\log(|I|/\delta)),$$

with probability at least $1 - 2\delta$, which completes the proof. $\qquad\square$

**Lemma 5.** *When all the minimizers $\theta_t^* = \arg\min_{\theta\in\mathbb{R}^d} L_t(\theta)$ lie in the interior of $\Theta$, then we have*

$$\|\nabla\widetilde{L}_t(\theta_t^*)\|_2^2 = \frac{2R^2\ln((d+1)/\delta)}{N_0} + o\left(\frac{\log(d/\delta)}{N_0}\right).$$

*Proof of Lemma 5.* Since all the minimizers $\theta_t^* = \arg\min_{\theta\in\mathbb{R}^d} L_t(\theta)$ lie in the interior of $\Theta$, then we have $\nabla L_t(\theta_t^*) = 0$ and the term $\|\nabla\widetilde{L}_t(\theta_t^*)\|_2^2$ can be rewritten as

$$\|\nabla\widetilde{L}_t(\theta_t^*)\|_2^2 = \|\nabla\widetilde{L}_t(\theta_t^*) - \nabla L_t(\theta_t^*)\|_2^2$$

$$= \frac{1}{4}\|\mathbb{E}_{\mathbf{x}\sim S_0}[(\sigma(\mathbf{x}^\top\theta_t^*) - 1)\phi(\mathbf{x})] - \mathbb{E}_{\mathbf{x}\sim\mathcal{D}_0}[(\sigma(\mathbf{x}^\top\theta_t^*) - 1)\phi(\mathbf{x})]\|_2^2$$

$$= \frac{1}{4}\left\|\frac{1}{N_0}\sum_{n=1}^{N_0} Z_n\right\|_2^2, \tag{38}$$

where $\{Z_n\}_{n=1}^{N_0}$ with $Z_n = (\sigma(\phi(\mathbf{x}_n)^\top\theta_t^*) - 1)\phi(\mathbf{x}_n) - \mathbb{E}_{\mathbf{x}\sim\mathcal{D}_0}[(\sigma(\phi(\mathbf{x})^\top\theta_t^*) - 1)\phi(\mathbf{x})]$ are $N_0$ independent random vectors. We can check that $\mathbb{E}_{\mathbf{x}_n\sim\mathcal{D}_0}[Z_n] = 0$ and $\|Z_n\|_2 \leq 2R$ for any $\mathbf{x}_n \in \mathcal{X}$. Then by the Hoeffding's inequality for the random vector [69, Theorem 6.1.1], with probability at least $1 - \delta$, we have

$$\left\|\frac{1}{N_0}\sum_{n=1}^{N_0} Z_n\right\|_2 \leq \frac{2R\ln((d+1)\delta)}{3N_0} + \sqrt{\frac{8R^2\ln((d+1)/\delta)}{N_0}}. \tag{39}$$

Plugging (39) into (38) obtains $\|\nabla\widetilde{L}_t(\theta_t^*)\|_2^2 = \frac{2R^2\ln((d+1)/\delta)}{N_0} + o\left(\frac{\log(d/\delta)}{N_0}\right).$ $\qquad\square$

**Lemma 6.** *Let $\nabla\widehat{L}_t(\widehat{\theta}_t) = \frac{1}{2}\mathbb{E}_{\mathbf{x}\sim S_0}[(\sigma(\phi(\mathbf{x})^\top\widehat{\theta}_t) - 1)\phi(\mathbf{x})] + \frac{1}{2}\mathbb{E}_{\mathbf{x}\sim S_t}[\sigma(\phi(\mathbf{x})^\top\widehat{\theta}_t)\phi(\mathbf{x})]$. We have*

$$2\nabla\widehat{L}_t(\widehat{\theta}_t)\nabla\widehat{L}_t(\widehat{\theta}_t)^\top \preccurlyeq \mathbb{E}_{\mathbf{x}\sim S_0}[\mathbf{x}\mathbf{x}^\top] + \mathbb{E}_{\mathbf{x}\sim S_t}[\phi(\mathbf{x})(\phi(\mathbf{x})^\top].$$

*Proof of Lemma 6.* For notation simplicity, we denote by $\mathbf{a}_t = \mathbb{E}_{\mathbf{x}\sim S_0}[(\sigma(\mathbf{x}^\top\widehat{\theta}_t) - 1)\mathbf{x}]$ and $\mathbf{b}_t = \mathbb{E}_{\mathbf{x}\sim S_t}[\sigma(\mathbf{x}^\top\widehat{\theta}_t)\mathbf{x}]$. Then, according to the definition of $\nabla\widehat{L}_t(\widehat{\theta}_t)$, we have

$$\nabla\widehat{L}_t(\widehat{\theta}_t)\nabla\widehat{L}_t(\widehat{\theta}_t)^\top = \frac{1}{4}(\mathbf{a}_t + \mathbf{b}_t)(\mathbf{a}_t + \mathbf{b}_t)^\top \preccurlyeq \frac{1}{2}(\mathbf{a}_t\mathbf{a}_t^\top + \mathbf{b}_t\mathbf{b}_t^\top),$$

where the last inequality is due to Lemma 16. We can further bound the R.H.S. of the above inequality by

$$\mathbf{a}_t\mathbf{a}_t^\top = \left(\mathbb{E}_{\mathbf{x}\sim S_0}[(\sigma(\phi(\mathbf{x})^\top\widehat{\theta}_t) - 1)\phi(\mathbf{x})]\right)\left(\mathbb{E}_{\mathbf{x}\sim S_0}[(\sigma(\phi(\mathbf{x})^\top\widehat{\theta}_t) - 1)\phi(\mathbf{x})]\right)^\top$$

$$\preccurlyeq \mathbb{E}_{\mathbf{x}\sim S_0}[(\sigma(\mathbf{x}^\top\widehat{\theta}_t) - 1)^2\phi(\mathbf{x})\phi(\mathbf{x})^\top] \preccurlyeq \mathbb{E}_{\mathbf{x}\sim S_0}[\phi(\mathbf{x})\phi(\mathbf{x})^\top].$$

In above the first inequality is due to the fact that $\mathbb{E}[\mathbf{a}]\mathbb{E}[\mathbf{a}]^\top \preccurlyeq \mathbb{E}[\mathbf{a}\mathbf{a}^\top]$ for any random vector $\mathbf{a} \in \mathbb{R}^d$. The second inequality comes from $(\sigma(\mathbf{x}^\top \widehat{\theta}_t) - 1)^2 \leq 1$ since the output value of the sigmoid function $\sigma$ is bounded in $(0,1)$. A similar arguments shows $\mathbf{b}_t \mathbf{b}_t^\top \leq \mathbb{E}_{\mathbf{x} \sim S_t}[\phi(\mathbf{x})\phi(\mathbf{x})^\top]$.

Combining it with above displayed inequalities completes the proof. $\qquad\square$

**Lemma 7.** *The predictions returned by the ONS algorithm satisfies,*

$$\sum_{t \in I} \|\nabla \widehat{L}_t(\widehat{\theta}_t)\|_{A_t^{-1}}^2 \leq d \log \left(1 + \frac{|I|R^2}{\lambda}\right).$$

*Proof of Lemma 7.* The proof follows the standard arguments in the analysis of the online Newton step algorithm [13, Theorem 4.3]. We present the proof here for self-containedness.

$$
\begin{aligned}
\sum_{t \in I} \|\nabla \widehat{L}_t(\widehat{\theta}_t)\|_{A_t^{-1}}^2 &= \sum_{t \in I} \nabla \widehat{L}_t(\widehat{\theta}_t)^\top A_t^{-1} \nabla \widehat{L}_t(\widehat{\theta}_t) \\
&= \sum_{t \in I} \operatorname{trace}\left(A_t^{-1} \nabla \widehat{L}_t(\widehat{\theta}_t) \nabla \widehat{L}_t(\widehat{\theta}_t)^\top\right) \\
&= \sum_{t \in I} \operatorname{trace}\left(A_t^{-1}(A_t - A_{t-1})\right) \\
&\leq \sum_{t \in I} \log \frac{|A_t|}{|A_{t-1}|} = \log \frac{|A_e|}{|A_s|},
\end{aligned}
$$

where the last inequality is due to the fact that $\operatorname{trace}(A^{-1}(A - B)) \leq \log(|A|/|B|)$ for any positive definite matrix satisfying $A \succcurlyeq B \succ 0$ [13, Lemma 4.5].

By definition, we have $A_s = \lambda I_d \in \mathbb{R}^{d \times d}$ and we have $A_e = \lambda I_d + \sum_{t \in I} \nabla \widehat{L}_t(\widehat{\theta}_t) \nabla \widehat{L}_t(\widehat{\theta}_t)^\top$. Furthermore, since $\|\nabla \widehat{L}_t(\widehat{\theta}_t)\|_2 \leq R$, we have $|A_e| \leq (|I|R^2 + \lambda)^d$. In such a case, we can bound the above inequality by

$$\sum_{t \in I} \|\nabla \widehat{L}_t(\widehat{\theta}_t)\|_{A_t^{-1}}^2 \leq \log \frac{|A_e|}{|A_s|} \leq \log \frac{(|I|R^2 + \lambda)^d}{\lambda^d} \leq d \log \left(1 + \frac{|I|R^2}{\lambda}\right)$$

and finish the proof. $\qquad\square$

**Lemma 8.** *Let $Z_t = \langle \nabla \widetilde{L}_t(\widehat{\theta}_t) - \nabla \widehat{L}_t(\widehat{\theta}_t), \widehat{\theta}_t - \theta \rangle$. Then, with probability at least $1 - \delta$, we have*

$$
\begin{aligned}
\sum_{t \in I} Z_t \leq {} & 2\nu \sum_{t \in I} \left(\nabla \widehat{L}_t(\widehat{\theta}_t)^\top (\widehat{\theta}_t - \theta)\right)^2 + 2\nu \sum_{t \in I} \mathbb{E}_t \left[\left(\nabla \widehat{L}_t(\widehat{\theta}_t)^\top (\widehat{\theta}_t - \theta)\right)^2\right] \\
& + \frac{1}{\nu} \log \frac{\sqrt{2|I| + 1}}{\delta} + 4SR \sqrt{\log \frac{2|I| + 1}{\delta^2}}
\end{aligned}
$$

*for any $\delta \in (0,1)$ and $\nu > 0$.*

*Proof of Lemma 8.* We can check that $\{Z_t\}_{t=s}^e$ is a martingale difference sequence such that $\mathbb{E}_t[Z_t] = 0$ since $\widehat{\theta}_t$ is independent of $S_t$. Further noting that $|Z_t| \leq \|\nabla \widetilde{L}_t(\widehat{\theta}_t) - \nabla \widehat{L}_t(\widehat{\theta}_t)\|_2 \|\widehat{\theta}_t - \theta\|_2 \leq 4SR$, a direct application of Lemma 17 shows that

$$\sum_{t \in I} Z_t \leq \nu \sum_{t \in I} Z_t^2 + \nu \sum_{t \in I} \mathbb{E}_t \left[Z_t^2\right] + \frac{1}{\nu} \log \frac{\sqrt{2|I| + 1}}{\delta} + 4SR \sqrt{\log \frac{2|I| + 1}{\delta^2}}, \qquad (40)$$

with probability at least $1 - \delta$ for any $\nu > 0$. Then, for any $t \in [T]$, we can further bound the term $Z_t^2$ and $\mathbb{E}_t[Z_t^2]$ by

$$Z_t^2 = \langle \nabla \widetilde{L}_t(\widehat{\theta}_t) - \nabla \widehat{L}_t(\widehat{\theta}_t), \widehat{\theta}_t - \theta \rangle^2 \leq 2\langle \nabla \widetilde{L}_t(\widehat{\theta}_t), \widehat{\theta}_t - \theta \rangle^2 + 2\langle \nabla \widehat{L}_t(\widehat{\theta}_t), \widehat{\theta}_t - \theta \rangle^2 \qquad (41)$$

and

$$\mathbb{E}_t[Z_t^2] = \mathbb{E}_t[\langle \nabla \widehat{L}_t(\widehat{\theta}_t), \widehat{\theta}_t - \theta \rangle^2] - \langle \nabla \widetilde{L}_t(\widehat{\theta}_t), \widehat{\theta}_t - \theta \rangle^2,$$

where the first inequality is due to the fact $(a-b)^2 \le 2a^2 + 2b^2$ for any $a, b \in \mathbb{R}$. Then, combining the above two displayed inequalities and taking the summation over $T$ iterations, we have

$$\sum_{t\in I} Z_t^2 + \sum_{t\in I} \mathbb{E}_t[Z_t^2] \le \sum_{t\in I} Z_t^2 + 2\sum_{t\in I} \mathbb{E}_t[Z_t^2] \le 2\langle \nabla \widehat{L}_t(\widehat{\theta}_t), \widehat{\theta}_t - \theta \rangle^2 + 2\mathbb{E}_t[\langle \nabla \widehat{L}_t(\widehat{\theta}_t), \widehat{\theta}_t - \theta \rangle^2].$$

(42)

We complete the proof by plugging (42) into (40). $\qquad\square$

**Lemma 9.** *With probability at least $1 - \delta$, we have*

$$\frac{3}{4}\sum_{t\in I}(\nabla \widehat{L}_t(\widehat{\theta}_t)^\top(\widehat{\theta}_t - \theta))^2 - \frac{5}{4}\sum_{t\in I}\mathbb{E}_t\left[(\nabla \widehat{L}_t(\widehat{\theta}_t)^\top(\widehat{\theta}_t - \theta))^2\right] \le \Lambda_T(\delta)$$

*for any $\delta \in (0,1)$, where $\Lambda_I(\delta) = 32S^2R^2\log\frac{\sqrt{2|I|+1}}{\delta} + 4S^2R^2\sqrt{\log\frac{2|I|+1}{\delta^2}}$*

*Proof of Lemma 9.* We can check $\{Y_t\}_{t=s}^e$ is a martingale difference sequence such that $|Y_t| \le 4S^2R^2$. As a consequence, a direct application of Lemma 17 implies,

$$\sum_{t\in I} Y_t = \sum_{t\in I}(\nabla \widehat{L}_t(\widehat{\theta}_t)^\top(\widehat{\theta}_t - \theta))^2 - \sum_{t\in I}\mathbb{E}_t\left[(\nabla \widehat{L}_t(\widehat{\theta}_t)^\top(\widehat{\theta}_t - \theta))^2\right]$$

$$\le 2\nu\sum_{t\in I}\left((\nabla \widehat{L}_t(\widehat{\theta}_t)^\top(\widehat{\theta}_t - \theta))^4 + \mathbb{E}_t[(\nabla \widehat{L}_t(\widehat{\theta}_t)^\top(\widehat{\theta}_t - \theta))^4]\right)$$

$$+ \frac{1}{\nu}\log\frac{\sqrt{2|I|+1}}{\delta} + 4S^2R^2\sqrt{\log\frac{2|I|+1}{\delta^2}}$$

$$\le \frac{1}{4}\sum_{t\in I}\left((\nabla \widehat{L}_t(\widehat{\theta}_t)^\top(\widehat{\theta}_t - \theta))^2 + \mathbb{E}_t[(\nabla \widehat{L}_t(\widehat{\theta}_t)^\top(\widehat{\theta}_t - \theta))^2]\right)$$

$$+ 32S^2R^2\log\frac{\sqrt{2|I|+1}}{\delta} + 4S^2R^2\sqrt{\log\frac{2|I|+1}{\delta^2}},$$

where the first inequality follows the proof of Lemma 8 and the second inequaliuty is due to the parameter setting $\nu = 1/(32S^2R^2)$ and the condition that $(\nabla \widehat{L}_t(\widehat{\theta}_t)^\top(\widehat{\theta}_t - \theta))^2 \le 4S^2R^2$. Rearranging the above inequality, we have

$$\frac{3}{4}\sum_{t\in I}(\nabla \widehat{L}_t(\widehat{\theta}_t)^\top(\widehat{\theta}_t - \theta))^2 - \frac{5}{4}\sum_{t\in I}\mathbb{E}_t\left[(\nabla \widehat{L}_t(\widehat{\theta}_t)^\top(\widehat{\theta}_t - \theta))^2\right]$$

$$\le 32S^2R^2\log\frac{\sqrt{2|I|+1}}{\delta} + 4S^2R^2\sqrt{\log\frac{2|I|+1}{\delta^2}},$$

which completes the proof. $\qquad\square$

### D.3 Regret Guarantee for Meta-Algorithm (Lemma 10)

We have the following guarantee on the meta-algorithm, which ensures that for any interval $I_i \in \mathcal{C}$, the final prediction is comparable with the prediction of the base-learner $\mathcal{E}_i$ with an $\mathcal{O}(\log T)$ cost.

**Lemma 10.** *For any interval $I_i \in \mathcal{C}$ and $\delta \in (0,1)$, with probability at least $1 - \delta$, the final prediction returned by our meta-learner satisfies*

$$\sum_{t\in I_i}\widetilde{L}_t(\widehat{\theta}_t) - \sum_{t\in I_i}\widetilde{L}_t(\widehat{\theta}_{t,i}) \le \mathcal{O}\left(\beta\log(T/\delta) + \beta\ln K\right),$$

*where $K = |\mathcal{C}|$ is the number of the intervals contained in the set $\mathcal{C}$.*

### D.3.1 Main Proof

This part presents the proof of Lemma 10. Our analysis for the meta-learner is based on that of the strongly adaptive algorithm [70], where the problem of tracking base-learners on the corresponding intervals is converted to a sleeping expert problem [71]. The main challenge of our problem is that the feedback function is noisy. Thus, we require to provide tight analysis on the generalization gap between $\widehat{L}_t$ and $\widetilde{L}_t$ without ruining the $\mathcal{O}(\log T)$ regret guarantee. By twisting the meta-algorithm from Hedge [42] used in the previous work to Adapt-ML-Prod [43], we show that the generalization gap can be cancelled by the negative term introduced by the exp-concavity of the loss functions (see how to bound term (b) in (48)).

*Proof of Lemma 10.* To prove Lemma 10, we show that the final prediction $\widehat{\theta}_t$ generated by rule $\widehat{\theta}_t = \sum_{\mathcal{E}_i \in \mathcal{A}_t} p_{t,i} \widehat{\theta}_{t,i}$ with the weight update procedure (25), (26) and (27) is identical to an algorithm learning in the standard prediction with expert advice (PEA) setting [72], where all base-learners submit their predictions at each time $t \in [T]$.

**Lemma 11.** *For any base-learner $\mathcal{E}_i$, let $\mathcal{E}_i'$ be a surrogate of $\mathcal{E}_i$, whose prediction $\bar{\theta}_{t,i} = \widehat{\theta}_{t,i}$ is the same as that of $\mathcal{E}_i$ for any $t \in I_i$ and $\bar{\theta}_{t,i} = \widehat{\theta}_t$ for other iterations. Then, for any time $t \in [T]$, the surrogate meta-learner predicts as*

$$\bar{\theta}_t = \sum_{i \in [K]} \bar{p}_{t,i} \bar{\theta}_{t,i} \quad and \quad \bar{p}_{t,i} = \frac{\bar{\varepsilon}_{t-1,i} \bar{v}_{t-1,i}}{\sum_{i \in [K]} \bar{\varepsilon}_{t-1,i} \bar{v}_{t-1,i}}, \tag{43}$$

*where the potential $\bar{v}_{t,i}$ and the learning rate $\bar{\varepsilon}_{t,i}$ is defined as*

$$\bar{v}_{t,i} = \left( \bar{v}_{t-1,i} \cdot \left( 1 + \bar{\varepsilon}_{t-1,i} \bar{m}_{t,i} \right) \right)^{\frac{\bar{\varepsilon}_{t,i}}{\bar{\varepsilon}_{t-1,i}}} for all t \in [T], \tag{44}$$

*and*

$$\bar{\varepsilon}_{t,i} = \min \left\{ \frac{1}{2}, \sqrt{\frac{\ln K}{1 + \sum_{\tau=1}^{t} \bar{m}_{\tau,j}^2}} \right\} for all t \in [T] \tag{45}$$

*with $\bar{m}_{t,i} = \langle \widehat{L}_t(\bar{\theta}_t), \bar{\theta}_t - \bar{\theta}_{t,i} \rangle$, $\bar{v}_{0,i} = 1/K$ and $\bar{\varepsilon}_{0,i} = \min\{1/2, \ln K\}$ for all $i \in [K]$. Then, we have $\bar{\theta}_t = \widehat{\theta}_t$ for any $t \in [T]$.*

We have the following guarantee for the final prediction generated by the surrogate algorithm defined by (43).

**Lemma 12.** *With probability at least $1 - 2\delta$, for any $\delta \in (0,1)$, the prediction returned by the surrogate meta-algorithm defined as (43) ensures,*

$$\sum_{t=1}^{T} \widetilde{L}_t(\bar{\theta}_t) - \sum_{t=1}^{T} \widetilde{L}_t(\bar{\theta}_{t,j}) \leq C_T^{(2)}(\delta) = \mathcal{O}\left( \beta \log(T/\delta) + \beta \ln K \right)$$

*for any base-algorithm $\mathcal{E}_i'$, $i \in [K]$, where $C_T^{(2)}(\delta) = \frac{16(1+\beta)S^2 R^2 (C_T')^2}{\ln K} + \frac{1}{6(1+\beta)} + 2C_T' + \left( 24(1+\beta) + \frac{11 S^2 R^2}{1+\beta} \right) \log \frac{\sqrt{2T+1}}{\delta} + \left( 4SR + \frac{S^2 R^2}{2(1+\beta)} \right) \sqrt{\log \frac{2T+1}{\delta^2}}$ and $C_T' = 3 \ln K + \ln \left( 1 + \frac{K}{2e}(1 + \ln(T+1)) \right) = \mathcal{O}(\ln K + \ln \ln T)$.*

Then, according to the relationship between $\mathcal{E}_i$ and $\mathcal{E}_i'$, we can bound the regret of the final prediction $\widehat{\theta}_t$ with respect to that of any base-learner $\mathcal{E}_i$ over the interval $I_i$ by

$$\sum_{t \in I_i} \widetilde{L}_t(\widehat{\theta}_t) - \sum_{t \in I_i} \widetilde{L}_t(\widehat{\theta}_{t,j}) = \sum_{t=1}^{T} \widetilde{L}_t(\bar{\theta}_t) - \sum_{t=1}^{T} \widetilde{L}_t(\bar{\theta}_{t,j}) \leq C_T^{(2)}(\delta) = \mathcal{O}(\beta \log(T/\delta) + \beta \ln K).$$

In above, the first equality is due to Lemma 11 and the definition of the surrogate base-learner $\mathcal{E}_i'$ such that $\bar{\theta}_{t,i} = \widehat{\theta}_{t,i}$ for any $t \in I_i$ and $\bar{\theta}_{t,i} = \widehat{\theta}_t$ for any $t \in [T]/I_i$. The first inequality is due to Lemma 12. In the last equality, we treat double logarithmic factors in $T$ as a constant, following previous studies [73, 74]. We have completed the proof. $\square$

### D.3.2 Useful Lemma

*Proof of Lemma 11.* We show that the predictions $\widehat{\theta}_t$ and $\bar{\theta}_t$ by the two algorithm are exactly the same for any $t \in [T]$ by induction. At iteration $t \in [T]$, we have the following two induction hypotheses:

- *IH1:* the final predictions satisfy $\widehat{\theta}_\tau = \bar{\theta}_\tau$ for all $\tau \le t$.

- *IH2:* any base-algorithms $\mathcal{E}_i \in \mathcal{A}_\tau$ satisfy $p_{\tau,i} = \bar{p}_{\tau,i}/(\sum_{i \in \mathcal{A}_\tau} \bar{p}_{\tau,i})$ for all $\tau \le t$.

**Base case:** For the base case of $t = 1$, there is only one active base-algorithm. Denote by $i_1$ the index of the active base-algorithm, we have $\widehat{\theta}_1 = \widehat{\theta}_{1,i_1}$, $p_{1,i_1} = 1$ and $\mathcal{A}_1 = \{i_1\}$. Moreover, for the surrogate algorithm, we have $\bar{p}_{1,i} = 1/K$ for any $i \in [K]$ and $\bar{\theta}_1 = (\sum_{i \in [K]/\{i_1\}} \widehat{\theta}_1 + \widehat{\theta}_{1,i_1})/|K| = \widehat{\theta}_1$. We can check that $\widehat{\theta}_1 = \bar{\theta}_1$ and $p_{1,i} = \bar{p}_{1,i}/\sum_{i \in \mathcal{A}_1} \bar{p}_{1,i}$ for any $i \in \mathcal{A}_1$.

**Induction step:** For the induction step, we first show

$$p_{t+1,i} = \frac{\bar{p}_{t+1,i}}{\sum_{i \in \mathcal{A}_{t+1}} \bar{p}_{t+1,i}} \tag{46}$$

for all $i \in \mathcal{A}_{t+1}$. By the definition of $\bar{p}_{t+1,i}$, it is sufficient to prove

$$p_{t+1,i} = \frac{\bar{\varepsilon}_{t,i} \bar{v}_{t,i}}{\sum_{i \in \mathcal{A}_{t+1}} \bar{\varepsilon}_{t,i} \bar{v}_{t,i}} \quad \text{for any } i \in \mathcal{A}_{t+1}. \tag{47}$$

For the base-algorithm $\mathcal{E}_i$ active at iteration $t + 1$ and his/her corresponding surrogate base-algorithm $\mathcal{E}_i'$, we can decompose the time horizon until $t + 1$ into two intervals: asleep interval $\mathcal{I}_i^{\mathsf{slp}} = [1, s_i - 1]$ and active part $\mathcal{I}_i^{\mathsf{act}} = [s_i, t + 1]$. For any time stamp belonging to the asleep interval $\tau \in \mathcal{I}_i^{\mathsf{slp}}$, we can check that the loss for the base-algorithm $\mathcal{E}_i'$ satisfies

$$\bar{m}_{\tau,i} = \langle \widehat{L}_\tau(\bar{\theta}_\tau), \bar{\theta}_\tau - \bar{\theta}_{\tau,i} \rangle = \langle \widehat{L}_\tau(\bar{\theta}_\tau), \widehat{\theta}_\tau - \widehat{\theta}_\tau \rangle = 0,$$

where the first equality is by the definition of $\bar{m}_{\tau,i}$. The second equality is due to *IH1* such that $\bar{\theta}_\tau = \widehat{\theta}_\tau$ for any $\tau \le t$ and the definition of the surrogate base-learner $\mathcal{E}_i'$ such that the prediction $\bar{\theta}_{\tau,i} = \widehat{\theta}_\tau$ on the asleep interval. For any time stamp belonging to the active interval $\tau \in \mathcal{I}_i^{\mathsf{act}}$, similar argument shows that

$$\bar{m}_{\tau,i} = \langle \widehat{L}_\tau(\bar{\theta}_\tau), \bar{\theta}_\tau - \bar{\theta}_{\tau,i} \rangle = \langle \widehat{L}_\tau(\widehat{\theta}_\tau), \widehat{\theta}_\tau - \widehat{\theta}_{\tau,i} \rangle = m_{\tau,i},$$

Thus, by definition, we can draw the conclusion that $\bar{v}_{t,i} = v_{t,i}$ and $\bar{\varepsilon}_{t,i} = \varepsilon_{t,i}$ for any active base-algorithm with index $i \in \mathcal{A}_{t+1}$, which would finally lead to (47).

Then, we show $\widehat{\theta}_{t+1} = \bar{\theta}_{t+1}$ based on *IH1* and *IH2*. We have

$$\bar{\theta}_{t+1} = \sum_{i \in \mathcal{A}_{t+1}} \bar{p}_{t+1,i} \widehat{\theta}_{t+1,i} + \sum_{i \in [K]/\mathcal{A}_{t+1}} \bar{p}_{t+1,i} \widehat{\theta}_{t+1}$$

$$= \left( \sum_{i \in \mathcal{A}_{t+1}} \bar{p}_{t+1,i} \right) \left( \sum_{i \in \mathcal{A}_{t+1}} p_{t+1,i} \widehat{\theta}_{t+1,i} \right) + \left( 1 - \sum_{i \in \mathcal{A}_{t+1}} \bar{p}_{t+1,i} \right) \widehat{\theta}_{t+1}$$

$$= \left( \sum_{i \in \mathcal{A}_{t+1}} \bar{p}_{t+1,i} \right) \widehat{\theta}_t + \left( 1 - \sum_{i \in \mathcal{A}_{t+1}} \bar{p}_{t+1,i} \right) \widehat{\theta}_{t+1} = \widehat{\theta}_t,$$

where the first equality is due to (46) and the second inequality is due to the definition of $\widehat{\theta}_{t+1}$. We have finished the induction steps and completed the proof. □

*Proof of Lemma 12.* Since $\widehat{\theta}_t$ and $\widehat{\theta}_{t,i}$ are both independent of $S_t$, using the same arguments for obtaining (33), we can obtain that

$$\sum_{t=1}^T \widetilde{L}_t(\widehat{\theta}_t) - \sum_{t=1}^T \widetilde{L}_t(\widehat{\theta}_{t,i}) \le \underbrace{\sum_{t=1}^T \langle \nabla \widehat{L}_t(\widehat{\theta}_t), \widehat{\theta}_t - \widehat{\theta}_{t,i} \rangle}_{\texttt{term (a)}} + \underbrace{\sum_{t=1}^T \langle \nabla \widetilde{L}_t(\widehat{\theta}_t) - \nabla \widehat{L}_t(\widehat{\theta}_t), \widehat{\theta}_t - \widehat{\theta}_{t,i} \rangle}_{\texttt{term (b)}}$$

$$-\sum_{t=1}^{T}\frac{1}{2(\beta+1)}\mathbb{E}_t\left[\left(\nabla\widehat{L}_t(\widehat{\theta}_t)^\top(\widehat{\theta}_t-\widehat{\theta}_{t,i})\right)^2\right] \tag{48}$$

For term (a), we have

$$\mathtt{term\,(a)}\leq\frac{2SRC_T'}{\sqrt{\ln K}}\sqrt{1+\sum_{t=1}^{T}\left(\nabla\widehat{L}_t(\widehat{\theta}_t)^\top(\widehat{\theta}_t-\widehat{\theta}_{t,i})\right)^2}+2C_T'$$

$$\leq\frac{24(\beta+1)S^2R^2(C_T')^2}{\ln K}+\frac{1}{6(\beta+1)}+\frac{1}{6(\beta+1)}\sum_{t=1}^{T}\left(\nabla\widehat{L}_t(\widehat{\theta}_t)^\top(\widehat{\theta}_t-\widehat{\theta}_{t,i})\right)^2+2C_T', \tag{49}$$

for any $i\in[K]$, where $C_T'=3\ln K+\ln\left(1+\frac{K}{2e}(1+\ln(T+1))\right)=\mathcal{O}(\ln K+\ln\ln T)$. In above, the first inequality is a direct application of [43, Corollary 4] with the linearized loss $M_t(\theta)=\langle\widehat{L}_t(\widehat{\theta}_t),\theta\rangle/(2SR)+1/2$ and the second inequality is due to the AM-GM inequality.

For term (b), by Lemma 8 with $\nu=6(1+\beta)$ and a similar argument in (36), we have

$$\mathtt{term\,(b)}\leq\frac{1}{12(1+\beta)}\sum_{t=1}^{T}\left(\left(\nabla\widehat{L}_t(\widehat{\theta}_t)^\top(\widehat{\theta}_t-\widehat{\theta}_{t,i})\right)^2+\mathbb{E}_t\left[\left(\nabla\widehat{L}_t(\widehat{\theta}_t)^\top(\widehat{\theta}_t-\widehat{\theta}_{t,i})\right)^2\right]\right)$$

$$+24(1+\beta)\log\frac{\sqrt{2T+1}}{\delta}+4SR\sqrt{\log\frac{2T+1}{\delta^2}} \tag{50}$$

Combining the upper bound for term (a) and term (b) with (48) yields,

$$\sum_{t=1}^{T}\widetilde{L}_t(\widehat{\theta}_t)-\sum_{t=1}^{T}\widetilde{L}_t(\widehat{\theta}_{t,i})$$

$$\leq\underbrace{\frac{1}{4(1+\beta)}\sum_{t=1}^{T}\left(\nabla\widehat{L}_t(\widehat{\theta}_t)^\top(\widehat{\theta}_t-\widehat{\theta}_{t,i})\right)^2-\frac{5}{12(1+\beta)}\mathbb{E}_t\left[\left(\nabla\widehat{L}_t(\widehat{\theta}_t)^\top(\widehat{\theta}_t-\widehat{\theta}_{t,i})\right)^2\right]}_{\mathtt{term\,(c)}}$$

$$+\frac{24(1+\beta)S^2R^2(C_T')^2}{\ln K}+\frac{1}{6(1+\beta)}+2C_T'+24(1+\beta)\log\frac{\sqrt{2T+1}}{\delta}+4SR\sqrt{\log\frac{2T+1}{\delta^2}}.$$

Then, according to Lemma 9, we can further bound term (c) by

$$\mathtt{term\,(c)}=\frac{1}{4(1+\beta)}\sum_{t=1}^{T}\left(\nabla\widehat{L}_t(\widehat{\theta}_t)^\top(\widehat{\theta}_t-\widehat{\theta}_{t,i})\right)^2-\frac{5}{12(1+\beta)}\mathbb{E}_t\left[\left(\nabla\widehat{L}_t(\widehat{\theta}_t)^\top(\widehat{\theta}_t-\widehat{\theta}_{t,i})\right)^2\right]$$

$$\leq\frac{11S^2R^2}{1+\beta}\log\frac{\sqrt{2|I|+1}}{\delta}+\frac{4S^2R^2}{3(1+\beta)}\sqrt{\log\frac{2T+1}{\delta^2}},$$

which implies

$$\sum_{t=1}^{T}\widetilde{L}_t(\widehat{\theta}_t)-\sum_{t=1}^{T}\widetilde{L}_t(\widehat{\theta}_{t,i})$$

$$\leq\frac{24(1+\beta)S^2R^2(C_T')^2}{\ln K}+\frac{1}{6(1+\beta)}+2C_T'+C_1'\log\frac{\sqrt{2T+1}}{\delta}+C_2'\sqrt{\log\frac{2T+1}{\delta^2}}$$

$$=\mathcal{O}(\beta\log(T/\delta)+\beta\ln K).$$

In above, $C_1'=24(1+\beta)+11S^2R^2/(1+\beta)$ and $C_2'=4SR+4S^2R^2/(3(1+\beta))$. In the last equality, we treat double logarithmic factors in $T$ as a constant, following previous studies [73, 74]. We have completed the proof. □

### D.4 Proof of Theorem 2

This part presents the proof for Theorem 2, which is based on the dynamic regret for the base-learner (Lemma 3) and that for the meta-learner (Lemma 10) presented in Appendix D.1.

### D.4.1 Main Proof

*Proof of Theorem 2.* When choosing the Bregman divergence function as $\psi(t) = \psi_{\mathsf{LR}}(t) \triangleq t\log t - (t+1)\log(t+1)$ and the hypothesis space $\mathcal{H}_\theta^{\mathsf{LR}} = \{\mathbf{x} \mapsto \exp(-\theta^\top \phi(\mathbf{x})) \mid \|\theta\|_2 \le S\}$. The expected loss function $L_t(\theta)$ as defined by (4) is equal to

$$L_t(\theta) = \frac{1}{2}\Big(\mathbb{E}_{\mathcal{D}_0}[\log(1 + e^{-\phi(\mathbf{x})^\top \theta})] + \mathbb{E}_{\mathcal{D}_t}[\log(1 + e^{\phi(\mathbf{x})^\top \theta})]\Big).$$

Let $\theta_t^* \triangleq \arg\min_{\theta \in \mathbb{R}^d} L_t(\theta)$. Since the ground-truth density ratio $r_t^*(\mathbf{x}) = \mathcal{D}_t(\mathbf{x})/\mathcal{D}_0(\mathbf{x})$ function is contained in the hypothesis space as $\mathcal{D}_t(\mathbf{x})/\mathcal{D}_0(\mathbf{x}) \in \mathcal{H}_\theta^{\mathsf{LR}}$. One can show that $\mathcal{D}_t(\mathbf{x})/\mathcal{D}_0(\mathbf{x}) = \exp(-(\theta_t^*)^\top \phi(\mathbf{x}))$ and $\theta_t^* \in \Theta = \{\theta \in \mathbb{R}^d \mid \|\theta\|_2 \le S\}$.

Then, for the base-learner $\mathcal{E}_i$ running on any interval $I_i$, Lemma 3 in Appendix D.2 shows that its prediction $\widehat{\theta}_{t,i}$ is comparable with the best model at every iteration $\theta_t^*$ with an $\mathcal{O}(|I_i|(V_{I_i}^\theta)^2)$ cost:

$$\sum_{t \in I_i} \widetilde{L}_t(\widehat{\theta}_{t,i}) - \sum_{t \in I_i} \widetilde{L}_t(\theta_t^*) \le \mathcal{O}\left(d\beta\log\left(|I_i|/\delta\right) + |I_i|(V_{I_i}^\theta)^2 + \frac{|I_i|\log(dT/\delta)}{N_0}\right), \quad (51)$$

where $V_{I_i}^\theta = \sum_{t=s_i+1}^{e_i}\|\theta_t^* - \theta_{t-1}^*\|_2$ is the path length measuring the fluctuation of the minimizers.

Besides, Lemma 10 in Appendix D.3 ensures that the prediction $\widehat{\theta}_t$ of the meta-learner is comparable with the prediction $\widehat{\theta}_{t,i}$ of each base-learner $\mathcal{E}_i$ on the associated interval $I_i$ with an $\mathcal{O}(\log T)$.

$$\sum_{t \in I_i} \widetilde{L}_t(\widehat{\theta}_t) - \sum_{t \in I_i} \widetilde{L}_t(\widehat{\theta}_{t,i}) \le \mathcal{O}\Big(\beta\log(T/\delta) + \beta\ln K\Big). \quad (52)$$

A direct combination of (51) and (52) shows that the meta-learner is able to track the best prediction $\theta_t^*$ on any interval $I_i \in \mathcal{C}$. Actually, by exploiting the structure of the geometric covering (24), it can be shown that such a property can be extended to arbitrary interval $I \in [T]$ as the following lemma, whose proof is deferred to Appendix D.4.2.

**Lemma 13.** *With probability at least $1 - 5\delta$ for any $\delta \in (0, 1)$, for any interval $[s, e] \in [T]$, the meta-learner's prediction $\widehat{\theta}_t$ returned by (9) ensures,*

$$\sum_{t \in I} \widetilde{L}_t(\widehat{\theta}_t) - \sum_{t \in I} \widetilde{L}_t(\theta_t^*)$$
$$\le \frac{R^2}{4}|I|(V_I^\theta)^2 + 2\log|I|\left(C_T^{(1)}(\delta') + C_T^2(\delta')\right) + \frac{16|I|\ln\big((d+1)\log|I|T/\delta\big)}{N_0} + o\left(\frac{|I|\log(dT/\delta)}{N_0}\right)$$

*where $V_I^\theta = \sum_{t=s+1}^{e}\|\theta_{t-1}^* - \theta_t^*\|_2$ and $\delta' = \delta/2\log|I|$. The coefficient $C_T^{(1)}(\delta) = \mathcal{O}\left(d\beta\log(T/\delta)\right)$ and $C_T^{(2)}(\delta) = \mathcal{O}\Big(\beta\log(T/\delta)\Big)$ is defined in Lemma 4 and Lemma 12, respectively.*

Then, we can use the bin partition [19, Lemma 30] to convert the dynamic regret on each interval to a dynamic regret over $T$ iterations. For self-containedness, we restate the lemma here.

**Lemma 14** (Lemma 30 of [19]). *There exists a partition $\mathcal{P}$ of the time horizon $[T]$ into $M = \mathcal{O}(\max\{dT^{1/3}(U_T^\theta)^{2/3}, 1\})$ intervals ,i.e., $\{I_i = [s_i, e_i]\}_{i=1}^M$ such that for any interval $I_i \in \mathcal{P}$, $U_{I_i}^\theta \le D_{\max}/\sqrt{|I_i|}$, where $U_{I_i}^\theta = \sum_{t=s_i+1}^{e_i}\|\theta_t^* - \theta_{t-1}^*\|_1$ and $D_{\max} = \max_{t \in [T]}\|\theta_t^*\|_\infty \le \max_{t \in [T]}\|\theta_t^*\|_2 \le S$.*

Then, we can decompose the overall dynamic regret of our method into those over the key partition $\mathcal{P}$. Specifically, with probability at least $1 - 5\delta$, we have

$$\sum_{t=1}^T \widetilde{L}_t(\widehat{\theta}_t) - \sum_{t=1}^T \widetilde{L}_t(\theta_t^*)$$
$$= \sum_{I_i \in \mathcal{P}}\left(\sum_{t \in I_i} \widetilde{L}_t(\widehat{\theta}_t) - \sum_{I_i} \widetilde{L}_t(\theta_t^*)\right)$$

$$\leq \sum_{I_i \in \mathcal{P}} \frac{R^2}{4} |I_i| (V_{I_i}^\theta)^2 + \sum_{I_i \in \mathcal{P}} \frac{16|I_i| \ln\left(2(d+1)T \log|I_i| M/\delta\right)}{N_0}$$

$$+ 2 \sum_{I_i \in \mathcal{P}} \log|I_i| \left( C_T^{(1)}\left(\delta/2M \log|I_i|\right) + C_T^2\left(\delta/2M \log|I_i|\right) \right)$$

$$\leq \underbrace{\frac{MS^2 R^2}{4}}_{\texttt{term (a)}} + \underbrace{2M \log T \left( C_T^{(1)}\left(\delta/2M \log T\right) + C_T^{(2)}\left(\delta/2M \log T\right) \right)}_{\texttt{term (b)}} + \underbrace{\frac{16T \ln\left(2(d+1)T \log TM/\delta\right)}{N_0}}_{\texttt{term (c)}},$$

where the second last inequality is due to Lemma 13 and a union bound taken over all intervals in the key partition. The last inequality is due to Lemma 14 such that $V_{I_i}^\theta \leq U_{I_i}^\theta \leq S/\sqrt{I_i}$ for any interval $I_i \in \mathcal{P}$. Lemma 14 shows that we can bound the partition number by $M = \mathcal{O}(\max\{dT^{1/3}(U_T^\theta)^{2/3}), 1\}) = \mathcal{O}(\max\{d^{4/3}T^{1/3}(V_T^\theta)^{2/3}, 1\})$. Then, we can further show that

$$\texttt{term (a)} \leq \mathcal{O}\left(d^{4/3} \max\{T^{1/3}(V_T^\theta)^{2/3}, 1\}\right),$$

$$\texttt{term (b)} \leq \mathcal{O}\left(d^{4/3} \beta \log T \log(T/\delta) \max\{T^{1/3}(V_T^\theta)^{2/3}, 1\}\right),$$

$$\texttt{term (c)} \leq \mathcal{O}\left(\frac{T \ln(dT/\delta)}{N_0}\right),$$

which implies that the overall dynamic regret is bounded by

$$\sum_{t=1}^{T} \widetilde{L}_t(\widehat{\theta}_t) - \sum_{t=1}^{T} \widetilde{L}_t(\theta_t^*) \leq \mathcal{O}\left( d^{\frac{4}{3}} \beta \log T \log\left(T/\delta\right) \cdot \max\{T^{\frac{1}{3}}(V_T^\theta)^{\frac{2}{3}}, 1\} + \frac{T \ln(dT/\delta)}{N_0}\right).$$

Hence, we complete the proof by showing $\widetilde{L}_t(\theta_t^*) = \widetilde{L}_t(r_t^*)$ under the realizable assumption and converting the variation of the model parameter $V_T^\theta = \sum_{t=2}^{T}\|\theta_t^* - \theta_{t-1}^*\|_2$ to the variation of the feature distribution $V_T = \sum_{t=2}^{T}\|\mathcal{D}_t(\mathbf{x}) - \mathcal{D}_{t-1}(\mathbf{x})\|_1$ by Lemma 15. $\qquad\square$

### D.4.2 Useful Lemmas

**Lemma 13.** *With probability at least $1 - 5\delta$ for any $\delta \in (0,1)$, for any interval $[s,e] \in [T]$, the meta-learner's prediction $\widehat{\theta}_t$ returned by* (9) *ensures,*

$$\sum_{t \in I} \widetilde{L}_t(\widehat{\theta}_t) - \sum_{t \in I} \widetilde{L}_t(\theta_t^*)$$

$$\leq \frac{R^2}{4}|I|(V_I^\theta)^2 + 2\log|I| \left( C_T^{(1)}(\delta') + C_T^2(\delta') \right) + \frac{16|I| \ln((d+1)\log|I|T/\delta)}{N_0} + o\left(\frac{|I| \log(dT/\delta)}{N_0}\right),$$

*where $V_I^\theta = \sum_{t=s+1}^{e}\|\theta_{t-1}^* - \theta_t^*\|_2$ and $\delta' = \delta/2\log|I|$. The coefficient $C_T^{(1)}(\delta) = \mathcal{O}\left(d\beta \log(T/\delta)\right)$ and $C_T^{(2)}(\delta) = \mathcal{O}\left(\beta \log(T/\delta)\right)$ is defined in Lemma 4 and Lemma 12, respectively.*

*Proof of Lemma 13.* A direct combination of Lemma 3 and 10 shows that, with probability at least $1 - 5\delta$, we have the following dynamic regret guarantee on any interval $I_i = [s_i, e_i] \in \mathcal{C}$.

$$\sum_{t \in I_i} \widetilde{L}_t(\widehat{\theta}_t) - \sum_{t \in I_i} \widetilde{L}_t(\theta_t^*)$$

$$\leq C_{I_i}^{(1)}(\delta) + C_T^{(2)}(\delta) + \frac{R^2}{4}|I_i|(V_{I_i}^\theta)^2 + \frac{16|I_i| \ln((d+1)|I_i|/\delta)}{N_0} + o\left(\frac{|I_i| \log(d|I_i|/\delta)}{N_0}\right), \quad (53)$$

where $V_{I_i}^\theta = \sum_{t=s_i+1}^{e_i}\|\theta_{t-1}^* - \theta_t^*\|_2$. The coefficient $C_{I_i}^{(1)} = \mathcal{O}\left(d\beta \log(|I_i|/\delta)\right)$ is defined in Lemma 4 and $C_T^{(2)}(\delta) = \mathcal{O}\left(\beta \log(T/\delta)\right)$ is defined in Lemma 12.

For any interval $I = [s, e] \subseteq [T]$, [75, Lemma 3] showed that it can be partitioned into two sequences of disjoint and consecutive intervals, denoted by $I_{-p}, \ldots, I_0 \in \mathcal{C}$ and $I_1, \ldots, I_q \in \mathcal{C}$, such that

$$|I_{-i}|/|I_{-i+1}| \leq 1/2, \forall i \geq 1 \quad \text{and} \quad |I_i|/|I_{i-1}| \leq 1/2, \forall i \geq 2. \tag{54}$$

Then, with probability at least $1 - 5\delta$, we can decompose the regret over $I = [s, e] \subseteq [T]$ as

$$\sum_{t \in I} \widetilde{L}_t(\widehat{\theta}_t) - \sum_{t \in I} \widetilde{L}_t(\theta_t^*) = \sum_{i=-p}^{q} \left( \sum_{t \in I_i} \widetilde{L}_t(\widehat{\theta}_t) - \sum_{t \in I_i} \widetilde{L}_t(\theta_t^*) \right)$$

$$\leq (p+q)(C_T^{(1)}(\delta/p+q) + C_T^{(2)}(\delta/p+q)) + \frac{R^2}{4} \sum_{i=-p}^{q} |I_i|(V_{I_i}^\theta)^2$$

$$+ \sum_{i=-p}^{q} \frac{16|I_i| \ln((d+1)(p+q)T/\delta)}{N_0} + \sum_{i=-p}^{q} o\left( \frac{|I_i| \log(dT(p+q)/\delta)}{N_0} \right).$$

where the last inequality is due to (53) and a union bound taken over all intervals. Then, due to the partition of the intervals (54), we can check that $p + q \leq 2 \log|I|$. Then, we can further bound the above displayed inequality by

$$\sum_{t \in I} \widetilde{L}_t(\widehat{\theta}_t) - \sum_{t \in I} \widetilde{L}_t(\theta_t^*) \leq 2\log|I| \left( C_T^{(1)}(\delta/(2\log|I|)) + C_T^{(2)}(\delta/(2\log|I|)) \right) + \frac{R^2}{4}|I|(V_I^\theta)^2$$

$$+ \frac{16|I| \ln(2(d+1)\log|I|T/\delta)}{N_0} + o\left( \frac{|I| \log(dT\log|I|/\delta)}{N_0} \right)$$

$$= \frac{R^2}{4}|I|(V_I^\theta)^2 + \mathcal{O}\left( d\beta \log|I| \log(T\log|I|/\delta) + \frac{|I| \log(dT\log|I|/\delta)}{N_0} \right),$$

where the first inequality is due to the fact that $\sum_{i=-p}^{q} V_{I_i}^2 \leq V_I^2$. This completes the proof. $\square$

**Lemma 15.** *Under the condition that $\mathcal{D}_t(\mathbf{x})/\mathcal{D}_0(\mathbf{x}) \in \mathcal{H}_\theta^{\mathsf{LR}} \triangleq \{\exp(-\theta^\top \phi(\mathbf{x})) \mid \|\theta\|_2 \leq S\}$ for any $t \in [T]$ and $\mathbb{E}_{\mathbf{x} \sim \mathcal{D}_0}[|\phi(\mathbf{x})^\top \mathbf{a}|] \geq \alpha \|\mathbf{a}\|_2$ for any $\mathbf{a} \in \mathbb{R}^d$ with a certain $\alpha > 0$. Then, we have $\|\theta_{t-1}^* - \theta_t^*\|_2 \leq \beta \|\mathcal{D}_{t-1}(\mathbf{x}) - \mathcal{D}_t(\mathbf{x})\|_1/\alpha$.*

*Proof of Lemma 15.* Under the realizable assumption such that $\mathcal{D}_t(\mathbf{x})/\mathcal{D}_0(\mathbf{x}) \in \mathcal{H}_\theta^{\mathsf{LR}}$. One can verify that $\exp(-\phi(\mathbf{x})^\top \theta_t^*) = \mathcal{D}_t(\mathbf{x})/\mathcal{D}_0(\mathbf{x})$. Then, we can convert the variation of the model $\|\theta_{t-1}^* - \theta_t^*\|_2$ to the variation of the distribution $\|\mathcal{D}_{t-1}(\mathbf{x}) - \mathcal{D}_t(\mathbf{x})\|_1$.

$$\|\theta_{t-1}^* - \theta_t^*\|_2 \leq \frac{1}{\alpha} \mathbb{E}_{\mathbf{x} \sim \mathcal{D}_0} \left[ |\phi(\mathbf{x})^\top (\theta_{t-1}^* - \theta_t^*)| \right]$$

$$\leq \frac{\beta}{\alpha} \mathbb{E}_{\mathbf{x} \sim \mathcal{D}_0} \left[ |e^{\phi(\mathbf{x})^\top \theta_{t-1}^*} - e^{\phi(\mathbf{x})^\top \theta_t^*}| \right] \leq \frac{\beta}{\alpha} \mathbb{E}_{\mathbf{x} \sim \mathcal{D}_0} \left[ \left| \frac{\mathcal{D}_t(\mathbf{x})}{\mathcal{D}_0(\mathbf{x})} - \frac{\mathcal{D}_{t-1}(\mathbf{x})}{\mathcal{D}_0(\mathbf{x})} \right| \right] = \frac{\beta}{\alpha} \|\mathcal{D}_t(\mathbf{x}) - \mathcal{D}_{t-1}(\mathbf{x})\|_1,$$

where the first inequality is by the condition $\mathbb{E}_{\mathbf{x} \sim \mathcal{D}_0}[|\phi(\mathbf{x})^\top \mathbf{a}|] \geq \alpha \|\mathbf{a}\|_2$ and the second is by the mean value theorem and the boundedness of $|\phi(\mathbf{x})^\top (\theta_{t-1}^* - \theta_t^*)| \leq 2SR$ for any $\mathbf{x} \in \mathcal{X}$. The last inequality is due to the realizable assumption.

$\square$

## D.5 Proof of Theorem 3

*Proof of Theorem 3.* Then, we can bound the overall excess risk over the model trained by IW-ERM (1) by a combination of Proposition 1 and Theorem 1 as

$$\frac{1}{T} \left( \sum_{t=1}^{T} R_t(\widehat{\mathbf{w}}_t) - \sum_{t=1}^{T} R_t(\mathbf{w}_t^*) \right) \leq \frac{B \cdot C_T(\delta)}{\sqrt{N_0}} + \frac{2L}{T} \sum_{t=1}^{T} \mathbb{E}_{\mathbf{x} \sim S_0} \left[ |\widehat{r}_t(\mathbf{x}) - r_t^*(\mathbf{x})| \right]$$

$$\leq \frac{B \cdot C_T(\delta)}{\sqrt{N_0}} + \frac{4L\beta}{T} \sqrt{T \left[ \sum_{t=1}^{T} \widetilde{L}_t(\widehat{\theta}_t) - \sum_{t=1}^{T} \widetilde{L}_t(\theta_t^*) \right]_+} + \mathcal{O}\left( \frac{\log(T/\delta)\sqrt{d}}{\sqrt{N_0}} \right), \tag{55}$$

where the first inequality is due to Proposition 1 and $C_T(\delta) = 2GDR + 5L\sqrt{2\log(8T/\delta)} = \mathcal{O}(\log(T/\delta))$. The second inequality is due to Theorem 1 with the choice of the logistic regression model $\psi(t) = t\log t - (1+t)\log(1+t)$ and $\widehat{r}_t(\mathbf{x}) = \exp(-\phi(\mathbf{x})^\top \widehat{\theta}_t)$ and the realizability assumption $r_t^*(\mathbf{x}) = \exp(-\phi(\mathbf{x})^\top \theta_t^*)$. Then, by Theorem 2, we have

$$\sum_{t=1}^{T} \widetilde{L}_t(\widehat{r}_t) - \sum_{t=1}^{T} \widetilde{L}_t(r_t^*) \leq \mathcal{O}\left( d^{\frac{4}{3}} \log T \log\left(T/\delta\right) \cdot \max\{T^{\frac{1}{3}} V_T^{\frac{2}{3}}, 1\} + \frac{T\ln(dT/\delta)}{N_0} \right),$$

which implies

$$\sqrt{T \left[ \sum_{t=1}^{T} \widetilde{L}_t(\widehat{\theta}_t) - \sum_{t=1}^{T} \widetilde{L}_t(\theta_t^*) \right]_+} \leq \mathcal{O}\left( d^{\frac{2}{3}} \log\left(T/\delta\right) \cdot \max\{T^{\frac{2}{3}} V_T^{\frac{1}{3}}, \sqrt{T}\} + T\sqrt{\frac{\ln(dT/\delta)}{N_0}} \right).$$

$$(56)$$

We complete the proof by substituting (56) it into (55). $\qquad\square$

### D.6 Discussion on Tightness of the Bound

This part illustrates the tightness of our bound with the case where the labels of testing samples are available after the prediction.

**Continuous Shift with Labeled Feedback.** We consider a $T$-round online learning process. At iteration $t$, the learner submits her prediction $\widehat{\mathbf{w}}_t \in \mathcal{W}$. At the same time, the environments pick the data pair $(\mathbf{x}_t, y_t) \sim \mathcal{D}_t$. Then, the learner updates her model with the loss function $\widehat{R}_t(\mathbf{w}) = \ell(\mathbf{w}^\top \mathbf{x}_t, y_t)$ and obtains the prediction $\widehat{\mathbf{w}}_{t+1} \in \mathcal{W}$ for the next iteration. The goal of the learner is to minimize the cumulative excess risk against the sequence of optimal models $\mathbf{w}_t^* \in \arg\min_{\mathbf{w} \in \mathcal{W}} R_t(\mathbf{w})$, that is,

$$\mathfrak{R}_T(\{\widehat{\mathbf{w}}_t\}_{t=1}^T) = \frac{1}{T}\sum_{t=1}^{T} R_t(\widehat{\mathbf{w}}_t) - \sum_{t=1}^{T} R_t(\mathbf{w}_t^*). \tag{57}$$

The goal is the same as that of standard continuous covariate shift, see the problem setup in (1). The key difference is that now the label of testing data is *available*. Therefore, the noisy feedback $\widehat{R}_t(\mathbf{w})$ observed by the learner is an unbiased estimator with respect to the true risk $R_t(\mathbf{w})$.

**Tightness of Our Bound.** The continuous shift with labeled feedback is essentially a non-stationary stochastic optimization problem [39]. For general convex functions, [39] showed that any gradient-based algorithm will suffer

$$\mathbb{E}[\mathfrak{R}_T(\{\widehat{\mathbf{w}}_t\}_{t=1}^T)] = \Omega\left( T^{-1/3}(V_T^L)^{1/3} \right) \tag{58}$$

in the worst case. In above, $V_T^L = \sum_{t=2}^{T} \max_{\mathbf{w} \in \mathcal{W}} |R_t(\mathbf{w}) - R_{t-1}(\mathbf{w})|$ measures the fluctuation of the risk function. For the same performance measure, our algorithm achieves

$$\mathfrak{R}_T(\{\widehat{\mathbf{w}}_t\}_{t=1}^T) = \widetilde{\mathcal{O}}\left( \frac{1}{\sqrt{N_0}} + \max\{T^{-1/3} V_T^{1/3}, T^{-\frac{1}{2}}\} \right). \tag{59}$$

In the non-stationary case, i.e., $V_T \geq \Theta(T^{-\frac{1}{2}})$, our bound becomes $\widetilde{\mathcal{O}}\left( 1/\sqrt{N_0} + T^{-1/3} V_T^{1/3} \right)$. As discussed below Proposition 1, the first term $\widetilde{\mathcal{O}}(1/\sqrt{N_0})$ is hard to be improved. We thus focus on the tightness of the second term $\widetilde{\mathcal{O}}(T^{-1/3} V_T^{1/3})$.

Although the definition of $V_T$ in our upper bound is different from the that of $V_T^L$ in the lower bound (58), the two bounds share the same dependence on the time horizon $T$, which provides evidence that our bound is hard to improve. Indeed, consider the 1-dimensional case where the underlying distribution only shifts once from $\mathcal{D}_1(x, y)$ to $\mathcal{D}_2(x, y)$ at a certain time $t \in [T]$. In such a case, $V_T = \mathcal{O}(1)$ and our bound implies an $\mathcal{O}(T^{-1/3})$ rate for the second term of (59). On the other hand, the rate in the lower bound (58) becomes $\Omega(T^{-1/3})$. The same dependence on the time horizon $T$ indicates our bound is hard to improve. We leave a precise lower bound argument as the future work.

### D.7 On Other Choices of Bregman Divergence Function

In this section, we discuss how to apply the other two choice of the divergence function $\psi_{\mathsf{LS}} = (t-1)^2/2$ and $\psi_{\mathsf{KL}} = t \log t - t$ in our framework. Our analysis crucially relies on two conditions:

1. the Bregman divergence function $\psi$ satisfies the conditions in Theorem 1.

2. the induced loss function $\widehat{L}_t(\theta)$ is an exp-concave and smooth function with respect to $\theta$.

**Choice of $\psi_{\mathsf{LS}}(t) = (t-1)^2/2$.** Considering the Bregman divergence function $\psi(t) = (t-1)^2/2$ and the hypothesis space $H_\theta = \{\mathbf{x} \mapsto \phi(\mathbf{x})^\top \theta \mid \|\theta\|_2 \leq S\}$, where $\phi : \mathcal{X} \to \mathbb{R}^d$ represents a specific basis function with bounded norm $\|\phi(\mathbf{x})\|_2 \leq R$. Then, the loss $\widehat{L}_t^\psi(\theta)$ as per (7) becomes

$$\widehat{L}_t(\theta) = \frac{1}{2}\mathbb{E}_{S_0}[(\phi(\mathbf{x})^\top \theta)^2] - \mathbb{E}_{S_t}[\phi(\mathbf{x})^\top \theta].$$

The above configuration recovers the unconstrained least-squares importance fitting (uLSIF) method [28]. We can show such a choice of divergence function satisfy the condition required by Theorem 1 and enjoys favorable function properties.

- Conditions in Theorem 1: when Bregman divergence function is chosen as $\psi_{\mathsf{LS}}$, we have $\partial^2 \psi_{\mathsf{LS}}(t) = 1$ and $\partial^3 \psi_{\mathsf{LS}}(t) = 0$. Thus, $\psi_{\mathsf{LS}}$ is a 1-strongly convex function and $t\partial^3\psi_{\mathsf{LS}}(t) = 0$ for $t \in \mathbb{R}$.

- Expconcavity and smoothness: We have $\nabla^2\widehat{L}_t(\theta) = \mathbb{E}_{S_0}[\phi(\mathbf{x})\phi(\mathbf{x})^\top]$. Then, when the input is upper bounded by $\|\phi(\mathbf{x})\|_2 \leq R$ for all $\phi(\mathbf{x}) \in \mathcal{X}$, we have $\nabla^2\widehat{L}_t(\theta) \preccurlyeq R^2 I_d$, which implies the smoothness. Furthermore, if the initial data ensure $\mathbb{E}_{S_0}[\phi(\mathbf{x})\phi(\mathbf{x})^\top] \succcurlyeq \alpha I_d$, the $\widehat{L}_t(\theta)$ is strongly convex and thus is exp-concave as shown by [60, Lemma 2]. We note that the regularity condition on the offline data $\mathbb{E}_{S_0}[\phi(\mathbf{x})\phi(\mathbf{x})^\top] \succcurlyeq \alpha I_d$ is used in the analysis of LSIF algorithm [28, Lemma 1].

**Choice of $\psi_{\mathsf{KL}} = t \log t - t$.** Considering the Bregman divergence function $\psi(t) = t \log t - t$ and the hypothesis $H_\theta = \{\mathbf{x} \mapsto \phi(\mathbf{x})^\top \theta \mid \|\theta\|_2 \leq S\}$, where $\phi : \mathcal{X} \to \mathbb{R}^d_+$ represents a specific basis function with bounded norm $\|\phi(\mathbf{x})\|_2 \leq R$. Then, the loss $\widehat{L}_t^\psi(\theta)$ as per (7) becomes

$$\widehat{L}_t(\theta) = \mathbb{E}_{S_0}[\phi(\mathbf{x})^\top \theta] - \mathbb{E}_{S_t}[\log(\phi(\mathbf{x})^\top \theta)].$$

To ensure $\widehat{L}_t(\theta)$ is well-defined, we further require the density function $\widehat{r}(\mathbf{x}) = \phi(\mathbf{x})^\top \theta > 1/\beta$ for any $\mathbf{x} \in \mathcal{X}$ and $\theta \in \Theta \triangleq \{\theta \mid \|\theta\|_2 \leq S\}$ with a certain positive constant $\beta > 0$. The above configuration recovers the UKL method [37] with the generalized linear model. We can show such a choice of divergence function satisfy the condition required by Theorem 1 and enjoys favorable function properties.

- Conditions in Theorem 1: When Bregman divergence function is chosen as $\psi_{\mathsf{KL}}$, we have $\partial^2 \psi_{\mathsf{KL}}(t) = 1/t$ and $\partial^3 \psi_{\mathsf{KL}}(t) = -1/t^2$. When the input of the loss function satisfies $t \geq \beta$, $\psi_{\mathsf{KL}}(t)$ is a $1/\beta$-strongly convex function, and we can also check that $t\partial^3\psi_{\mathsf{KL}}(t) \leq 0$ for all $t \in \mathbb{R}_+$.

- Expconcavity and smoothness: We have $\nabla^2\widehat{L}_t(\theta) = \mathbb{E}_{S_t}[\phi(\mathbf{x})\phi(\mathbf{x})^\top/(\phi(\mathbf{x})^\top \theta)^2]$. Then, since the output of the density ratio function $\widehat{r}(\mathbf{x}) = \phi(\mathbf{x})^\top \theta \geq 1/\beta$ for any $\mathbf{x} \in \mathcal{X}$ and $\theta \in \Theta$ and the boundedness of the input feature $\|\phi(\mathbf{x})\|_2 \leq R$, we have $\nabla^2\widehat{L}_t(\theta) \preccurlyeq \beta^2 R^2 I_d$ for any $\theta \in \Theta$. Then, the loss function is a smooth function. Besides, under the regularity condition $\mathbb{E}_{S_0}[\mathbf{x}\mathbf{x}^\top] \succcurlyeq \alpha I_d$ of the initial data and again the boundedness of $\Theta$ and $\|\phi(\mathbf{x})\|_2 \leq R$, the loss function is also a strongly convex function by $\nabla^2\widehat{L}_t(\theta) \succcurlyeq \alpha/(SR)^2 I_d$, which indicates the expconcavity of the loss function.

## E  Technical Lemmas

This section presents several useful technical lemmas used in the proof.

**Lemma 16.** *For any* $\mathbf{a}, \mathbf{b} \in \mathbb{R}^d$, *we have* $(\mathbf{a}+\mathbf{b})(\mathbf{a}+\mathbf{b})^\top \preccurlyeq 2(\mathbf{a}\mathbf{a}^\top + \mathbf{b}\mathbf{b}^\top)$, *where for any matrix* $A, B \in \mathbb{R}^{d \times d}$, $A \preccurlyeq B$ *indicates* $B - A$ *is a positive semi-definite matrix.*

*Proof.* For any $\mathbf{x} \in \mathbb{R}^d$, $\mathbf{x}^\top \left(2(\mathbf{a}\mathbf{a}^\top + \mathbf{b}\mathbf{b}^\top) - (\mathbf{a}+\mathbf{b})(\mathbf{a}+\mathbf{b})^\top\right)\mathbf{x} = \mathbf{x}^\top\left(\mathbf{a}\mathbf{a}^\top + \mathbf{b}\mathbf{b}^\top - \mathbf{a}\mathbf{b}^\top - \mathbf{b}\mathbf{a}^\top\right)\mathbf{x}$
$= \mathbf{x}^\top\left((\mathbf{a}-\mathbf{b})(\mathbf{a}-\mathbf{b})^\top\right)\mathbf{x} \geq 0$, which completes the proof. $\qquad\square$

**Lemma 17** (Theorem 4 of [68])**.** *Let* $\{Z_i : i \geq 1\}$ *be a martingale difference with the filtration* $\mathfrak{F} = \{\mathcal{F}_n : n \geq 1\}$ *and suppose* $|Z_i| \leq R$ *for all* $i \geq 1$. *Then, for any* $\delta \in (0,1)$, $r > 0$, *with probability at least* $1 - \delta$,

$$\left|\sum_{i=1}^t Z_i\right| \leq r\left(\sum_{i=1}^t Z_i^2 + \sum_{i=1}^t \mathbb{E}[Z_i^2 \mid \mathcal{F}_{i-1}]\right) + \frac{1}{r}\log\frac{\sqrt{2t+1}}{\delta} + R\sqrt{\log\frac{2t+1}{\delta^2}}.$$

