# OpenReview forum: "Adapting to Continuous Covariate Shift via Online Density Ratio Estimation"
_NeurIPS.cc/2023/Conference — NeurIPS 2023 poster_

### Official Review · Reviewer_KTsd · 2023-07-04

**Soundness:** 3 good
**Presentation:** 3 good
**Contribution:** 3 good
**Rating:** 7
**Confidence:** 3

**Summary:**

This work introduced an online density ratio estimation method that can adaptively update the model to minimize the risk accumulated over time in the continuous covariate shift scenario. This method is able to estimate density ratios between test and training samples when the test set is varying over time. Only a few unlabeled samples are required at each time. A theoretical analysis of the regret bound of the density ratio estimator is provided.

**Strengths:**

The paper is well written with clear justification and useful theoretical analysis. The proposed method relaxes the requirement of unbiasedness of prior work and does not need a large amount of unlabeled data at each time step. The authors also proved the dynamic regret bound of the density ratio estimator. The experiments demonstrate the effectiveness of the proposed method.

**Weaknesses:**

I don't have many complaints about this paper. One weakness is that some important experimental results are in the appendix which is not reasonable as the appendix is optional to read by reviewers. I understand the page limitation but I would suggest moving some preliminary content to the appendix instead.

**Questions:**

I understand covariate shift is a well-established research area but the assumption of covariate shift (p(y|x) not change) is quite strong, so how can we verify or guarantee this assumption is valid in a real-world problem?

**Limitations:**

Since this method is for continuous covariate shift, the inherent limitation would be the assumption of covariate shift.

---

> ### Author Rebuttal · Authors · 2023-08-09
>
> Many thanks for your great appreciation for our work and the helpful comments! In the following, we will address your questions. We will further improve the paper according to your suggestions.
>
> ---
> **Q1:** important experimental results are in the appendix which is not reasonable as the appendix is optional to read by reviewers
>
> **A1:** Many thanks for your careful review and constructive comments. In the next version, we will try to move the empirical results into the main text. We believe this is very feasible, given that an additional page is allowed in the camera-ready version. Thanks!
>
> ---
>
> **Q2:** but the assumption of covariate shift (p(y|x) not change) is quite strong, how can we verify or guarantee this assumption is valid in a real-world problem?
>
> **A2:**  We appreciate your insightful comment on the covariate shift condition and acknowledge that it is somewhat a strong assumption. However, it is important to note that the unsupervised distribution shift adaptation problem is inherently challenging, and it is generally hard to perform the adaptation without any assumptions. The covariate shift condition is one of the most fundamental assumptions in the study of learning with distribution shift, which has also been successfully employed in many real-world applications (e.g. [8,9,10]).
>
> There are several possible directions to relax the requirement on the covariate shift assumption. When no label information about the test distribution is available, we could explore more complex modeling of the distribution shift, such as considering the case where both covariates and labels shift (Chen et al., 2022). Another interesting direction is to study how to efficiently test the covariate shift assumption with limited labeled data from the test distribution, particularly in the continuous shift setting. Our study for the basic covariate shift setting may serve as a foundational step toward addressing more complex real-world distribution shift cases.
>
> Thank you for the comments; we will include a more detailed discussion about potential ways to generalize the covariate shift assumption in the next version.
>
> **Ref**: Chen et al. Estimating and Explaining Model Performance When Both Covariates and Labels Shift. In NeurIPS 2022.

---

### Official Review · Reviewer_Ls7D · 2023-07-05

**Soundness:** 4 excellent
**Presentation:** 3 good
**Contribution:** 2 fair
**Rating:** 7
**Confidence:** 4

**Summary:**

This paper proposes an online density ratio estimation method to adaptively train a predictor in the scenario of continuous covariate shift. The proposed method estimates the density ratio between the training and testing distributions using a small number of unlabelled samples and updates the predictor using a weighted empirical risk minimisation algorithm. The paper provides a clear and comprehensive explanation of the problem of continuous covariate shift and the challenges it poses. The paper also provides a thorough review of related work and explains how the proposed method builds on and improves existing methods. The authors also give a dynamic regret bound, which finally leads to an excess risk guarantee for the predictor. The regret bound is minimised by designing an online optimisation process to minimise the dynamic regret defined over the observed loss. The paper explains how to optimise the dynamic regret with the online ensemble framework developed in recent studies of non-stationary online convex optimisation. The proposed method is validated through empirical results on synthetic and real-world datasets, demonstrating its effectiveness compared to other algorithms.

Overall the paper appears technically sound, is easy to follow, and gives theoretical guarantees. The paper is less strong from the empirical perspective. Further, the setting is a little esoteric (offline training followed by online adaptation for models where density ratios can be used), and may be of limited appeal to a wider audience.

----

Post rebuttal increasing my score from 6 to 7

**Strengths:**

Formulating the problem using the Bregman divergence was a useful tool to be able to generalise some existing methods, and also unlocked the analysis that follows.

Disentangling the model training and importance weight estimation also allowed for adaptation to the continuous shift setting and abiility to change importance weight estimators.

**Weaknesses:**

In terms of empirical study, this is limited to a study using synthetic data, and a small study on the yearbook dataset. For the synthetic scenario, the nature of the shift - sinusoidal and square waves altering a convex combination of distributions for the first two, then linear and “Ber” (which is not fully described in the main text, but is samples from a Bernoulli distribution) seem rather simplistic and artificial. For the single real world experiment, it’s not clear how the hyperparameter settings for the various methods were chosen, so it’s hard to know if this result is cherry-picked or robust.

Another thing that’s not clear is whether the intervals chosen for the ensemble members match up with the period of change in the synthetic experiments. If they do, it’s not a surprise that they do well, but would require knowledge ahead of time.

Figures 2-5 are extremely small and hard to read when printed.

**Questions:**

How sensitive is the method to the intervals in the ensemble members?

How crucial is the ensemble to the method overall?

Am I right in thinking that the bound does not take account for the presence of the ensemble. Does this mean that the bound actually holds for the algorithm as employed in experiments?

Could you plot the bound for one of the synthetic examples to show how tight it is?

**Limitations:**

The authors did not address limitations or potential negative social impact.

---

> ### Author Rebuttal · Authors · 2023-08-09
>
> Thanks for your insightful comments. We will address your questions below and improve the paper according to your suggestions. To better present additional experimental results, we include them in a PDF attached with the global response.
>
> ---
>
> **Q1:** empirical study is limited to a study using synthetic data, and a small study on the yearbook
>
> **A1:** We understand the reviewer's concern regarding the distribution shift generating process and scale-up issue to large data in experiments. However, we would note that the main contribution of this work is to propose the **first theoretically grounded solution** for continuous distribution shift problem. Our experiments are designed to verify the effectiveness of the method, particularly its ability to reuse historical information properly.
>
> Our experimental design follows the **same** settings used in existing works on continuous label shift [1,2], where $\mathcal{D}_t$ is taken as a mixture of two distributions with different shift patterns. We also follow the convention to conduct experiments in three parts: **(1) synthetic data**, to provide controlled illustrations; **(2) four benchmark datasets**, to enable comparative evaluation (Regrettably, the results are reported in Tables 1 and 2 in Appendix A due to limited space); and **(3) real-world scenarios**, to demonstrate real-life applicability. While we acknowledge that more extensive real-world validation would be ideal, we believe that the experiments conducted have already shown the effectiveness of our method and supported the theoretical findings. In the future, we will find more real-world scenarios satisfying covariate shift assumptions and explore further results.
>
> [1]  Online adaptation to label distribution shift. NeurIPS'21
>
> [2]  Adapting to online label shift with provable guarantees. NeurIPS'22
>
> ---
>
> **Q2:** hyperparameter settings for the various methods
>
> **A2:** For each individual algorithm, we use **the same rule in all experiments** to select the parameters.
>
> - For the contenders (DANN, KMM, KLIEP, and uLSIF), the parameters are set to the default settings as detailed in the corresponding ADAPT python package.
>
> - As for our algorithm (Accous), we provide detailed parameter configurations in lines 670-688, Appendix A.2. Specifically, we need to specify the parameters $R$ and $S$ for the logistic regression. Here, $R$ is set as the maximum feature norm in the offline data. Meanwhile, $S$ is the maximum norm of density functions; we set it as $S = d/2$, where $d$ is the dimension of the feature. OLRE uses the same parameter configuration as Accous.
>
> In the revision, we will include the parameter setup in the main text for clearer accessibility. Thanks!
>
> ---
>
> **Q3:** Figures 2-5 are extremely small
>
> **A3:**  Thanks for the comment. We will enlarge the figures to enhance readability in the revision.
>
> ---
>
> **Q4:** whether the intervals chosen for the ensemble members match up with the period of change
>
> **A4:** In all performance comparison experiments, both on synthetic (Figure 2&4) and benchmark data (Table 1&2), the distribution change period **does not align with** the intervals of the ensemble members. These intervals are determined by the theoretical guidance, i.e., $|I_k| = 2^k$. By contrast, the period of distribution change is set as $M = \sqrt{T} = 100$ for $T =10000$, as mentioned in lines 612-614, Appendix A.
>
> **Figure A** in the PDF reports the additional results for the weight assign experiment (Figure 3 in the paper) in the case where the ensemble member interval does not match the shift period $M$. The results show our method can still assign larger weights to the "right'' ensemble members whose interval lengths are close to $M$.
>
> ---
>
> **Q5:** How sensitive is the method to the intervals in the ensemble members?
>
> **A5:** We conducted additional experiments to investigate the sensitivity of ensemble members' intervals length, increasing them by factors of 3 and 5, i.e., $\vert I_k\vert = 3^k$ and $\vert I_k\vert = 5^k$. The results, presented in **Table A** of the PDF, show that our algorithm is generally robust to variations in the interval length.
>
> ---
>
> **Q6:** How crucial is the ensemble to the method overall?
>
> **A6:** The ensemble structure is the core component to ensure our method's adaptivity. Intuitively, one of the main challenges of the continuous covariate shift is the unknown shift intensity $V_T$. To address the problem, we maintain multiple ensemble members to account for possible shift intensities of the environments and employ a meta-algorithm to combine them. The intuition is **supported by our theoretical analysis**, as shown by Lemma 10 in Appendix D.3, the ensemble structure ensures us to track the best ensemble members on each interval, which finally leads to the dynamic regret bound. **From an empirical standpoint,** Figure 3 further shows the importance of our ensemble structure, illustrating how it helps to selectively reuse the right amount of historical information according to the (unknown) shift intensity.
>
> ---
>
> **Q7:** ...the bound does not take account for the presence of the ensemble?
>
> **A7:** Our bound explicitly accounts for the presence of ensemble structure, encompassing both the meta-algorithm and the ensemble members with interval schedule  $|I_k| = 2^k$. As discussed in Q6, the ensemble structure is the core component to achieve our theoretical guarantees (Theorem 2&3).
>
> ---
>
> **Q8:** plot bound for one of the synthetic examples to show how tight it is
>
> **A8:** Thank you for the comment. We have provided a detailed discussion about the tightness of our results in lines 311-321, with further details in Appendix D.6. From a theoretical view, we show that the rate can hardly be improved, even if one can receive labels of the test stream after prediction. We further conducted experiments to show the consistency between our theory and empirical results. Please refer to **Figure** **B** in the PDF for more details. Thanks!

---

### Official Review · Reviewer_UHuD · 2023-07-06

**Soundness:** 3 good
**Presentation:** 4 excellent
**Contribution:** 3 good
**Rating:** 7
**Confidence:** 4

**Summary:**

This work studies the continuous covariate shift problem, where there exists an initial labelled dataset and in every subsequent round, a new unlabelled dataset is revealed. One needs to adapt the model for every round to achieve good performance. The paper uses importance-weighted ERM where the weights are estimated based on Bregman divergence minimization (Sec.3.3) and online ensemble (Sec.4.1). Theoretical analysis shows that the estimation has a linear dynamic regret (Theorem 2) and corresponding IWERM has an average excess risk that depends on problem instances (Theorem 3). Empirical studies on synthetic and real-world data show that the proposed method can outperform several alternatives.

**Strengths:**

- Clear writing and easy-to-follow presentation
- Strong theoretical guarantees on both the ratio function (Theorem 2) and the learned model (Theorem 3)
- Convincing empirical demonstrations on both synthetic and real-world datasets

**Weaknesses:**

- The algorithm and analysis rely on the assumption of using linear prediction models (see, e.g., Eq.(2) where the prediction is linear in w). Linear models are hardly sufficient in many cases, and more powerful models are needed.
- The work of Baby et al (2023), albeit very recent, should be discussed as it achieves a similar bound to the current work in a different setting.
- Baby, D., Garg, S., Yen, T.C., Balakrishnan, S., Lipton, Z.C. and Wang, Y.X., 2023. Online Label Shift: Optimal Dynamic Regret meets Practical Algorithms. *arXiv preprint arXiv:2305.19570*.
- A somewhat minor point to mention is that in the experiment, the description of the Yearbook dataset is not very clear. The appendix only mentions 10 images per round, but it remains unclear how the images are sequentially sampled.

Some minor comments
- R_t should be defined explicitly after Eq.(1) to avoid confusion (since it is the population version, different from (2), the empirical version)
- L173: Ref [37] doesn’t mention UKL and ref [25] is KLIEP instead of KLLEP
- L262: latter -> later
- L305: minimiers -> minimizers
- L336: KEIEP -> KLIEP

**Questions:**

Q1: Are there any possible ways to extend the current work beyond linear models?

Q2: How are the images being sampled for the Yearbook dataset?

**Limitations:**

The assumption of using linear models should be explicitly mentioned.

---

> ### Author Rebuttal · Authors · 2023-08-09
>
> Many thanks for your great appreciation and bringing the concurrent work to us! In the following, we will address your questions. We will further improve the paper according to your suggestions.
>
> ---
>
> **Q1:** Are there any possible ways to extend the current work beyond linear models.
>
> **A1**: We believe it is quite feasible to extend the current work beyond linear models. Our algorithm consists of two integral components: IWERM for predictor training and an online ensemble for density ratio estimation.
>
> - **For the predictor training part,** since this is an ERM-based method, we can extend our approach to learn within a more complex hypothesis space beyond the linear class, as long as Equation (2) can be minimized.
> - **For the density ratio estimation part,** if theoretical guarantees are not a primary concern, our ensemble algorithm can assuredly be extended to learn with more intricate models, such as deep neural networks.  When considering theoretical guarantees, the extension is not as straightforward, given that our method is based on the online convex optimization framework.  However, we believe there are many viable opportunities. For instance, Example 1 demonstrates that our model can already be implemented using a generalized linear model. Besides, it is feasible to extend our online ensemble framework to learn within the Reproducing Kernel Hilbert Space (RKHS), by leveraging the recent advances in online kernel learning [Calandriello et al., 2017; Zhang et al., 2019]. Furthermore, for more complex models like DNNs, although analyzing the nonlinear neural networks is always challenging, it could be an intriguing direction to consider combining the neural tangent kernel theory into the online learning process.
>
> Thank you for raising this point. We will add more discussion in the next version!
>
> Reference:
>
> [1] Daniele Calandriello, Alessandro Lazaric, and Michal Valko. Efficient second-order online kernel learning with adaptive embedding. In NeurIPS 2017.
>
> [2] Xiao Zhang and Shizhong Liao. Incremental randomized sketching for online kernel learning. In ICML 2019.
>
> ---
>
> **Q2**: about the online label shift paper from Baby et al., 2023
>
> **A2:** Thank you for bringing the concurrent paper on the online label shift to us. We will add a discussion to the paper in the next version.
>
> ---
>
> **Q3:** How are the images being sampled for the Yearbook dataset?
>
> **A3:** We generate the unlabeled data stream from the Yearbook dataset based on the inherent timestamps. Within the Yearbook dataset, each image is associated with a specific year. We meticulously reorganized the data to align with the chronological sequence of image years, introducing controlled randomness among images from the same year. Based on this process, we can generate coherent, sequential data that faithfully represents the passage of time with natural distribution shifts.
>
> We will provide a more detailed description of the yearbook dataset in the revision. Many thanks!

---

> > ### Comment · Reviewer_UHuD · 2023-08-16
> >
> > Thanks for the explanations. I would suggest adding the discussion about expansions beyond linear models to the paper to inspire future research.

---

> > > ### Author Response · Authors · 2023-08-17
> > >
> > > Thank you for the constructive comment! We will add a discussion about the expansions beyond linear models in the next version.

---

### Official Review · Reviewer_ZW3t · 2023-07-09

**Soundness:** 3 good
**Presentation:** 3 good
**Contribution:** 3 good
**Rating:** 5
**Confidence:** 3

**Summary:**

This paper focuses on deriving theoretical bounds for online density ratio when there exists continuous covariate shift. The formulation is based on the importance-weighted empirical risk minimization, which is a conventional one for covariate shift adaptation. The paper chooses the Bregman Divergence Density Ratio Matching as the method and tries to bound the regret of the to a dynamically changing optimal density ratio. The results are first established for a general convex function class and then instantiated to a logistic regression model. The online ensemble method proposed mimics the previous continuous label shift work. Experiments are conducted on four different synthetic shift patterns and mint/cifar datasets.








**Strengths:**

Pro:

The paper studies an important problem, is very clearly written, and has solid results.


**Weaknesses:**

Con:
I am not sure how much the first part of the analysis adds to our knowledge about the continuous covariate shift. This is maybe my bias. To my understanding, if we are in this kind of online learning scenario (shift is changing continuously), the minimization of the cumulative dynamic regret is a very straightforward choice. The question is how to minimize it and whether the theory guides the algorithm design. So, bounding the empirical estimation error using the approximated cumulated regret (theorem 1) is not very informative to me.

Also, the paper shows that many density estimator function satisfies the assumption and can achieve the bounds. So the main novel part of the proposed methodology seems to be the twist to the FLH algorithm?

Theoretically, even though it is nice to see a general and relatively standard algorithm can achieve the minimax optimal guarantee for the online density ratio estimation under continuous covariate shift, the theoretical contribution seems to be a bit limited: the high probability result (instead of the expected), and the greatly simplified analysis.

The experimental results can be presented in a better way.

**Questions:**

Questions:

It would be nice to see different shift change patterns and how they affect the learning results. But it seems the errors are averaged before showing? Figure 2 only shows different methods but covers 4 kinds of shifts?



**Limitations:**

There can be more discussions about limitations.

---

> ### Author Rebuttal · Authors · 2023-08-09
>
> Thank you for the detailed comments. In the following, we will first highlight the contribution of our work (Q1 and Q2) and then address your concern about the experiments (Q3). We will improve our paper according to your comments.
>
> ---
>
> **Q1：**“I am not sure how much the first part of the analysis adds to our knowledge about the continuous covariate shift…”
>
> **A1:**  While the conversion from density ratio estimation to regret minimization might appear intuitive (for those who are very familiar with both IW-ERM and density ratio estimation), we believe our first part of the analysis is *still novel and interesting to the community*, given that the study presents the **first theoretically grounded framework** for the continuous covariate shift problem.
>
> We emphasize the contribution of our framework through the following two points:
>
> - **Novel framework:** It is noteworthy to mention that existing approaches (e.g., [18], the closest work to ours proposed for continuous label shift) *cannot be applied to the continuous covariate shift problem*.  As discussed in line 63-67, the previous work typically follows the framework of first constructing an unbiased risk estimator $\hat{R}_t(\mathbf{w})$  and then conducting online optimization over $\hat{R}_t(\mathbf{\mathbf{w}})$ to learn the predictor. However, the construction of $\hat{R}_t(\mathbf{w})$ relies on the unbiasedness of the importance weight function such that that $\mathbb{E}[\hat{r}_t] = r_t$, which is hard to satisfy in the continuous covariate shift problem (e.g.,  $\mathbb{E}[\hat{r}_t] \neq r_t$ in Example 1). The first part of our analysis provides us a novel framework to handle continuous covariate shift.
>
> - **Flexibility:** Our framework is very flexible and holds the potential to serve as a principled way to manage more general continuous distribution shift scenarios. For instance, applying our framework to the continuous label shift problem immediately yields a new approach that *does not require the construction of an unbiased estimator*. Besides, with the disentanglement of predictor training and the IW estimation process, our framework supports the use of more complex hypotheses to train the predictor $\mathbf{w}$, while the model used in [18] is limited to a linear model.
>
> We thank the reviewer for the comments. In the next version, we will further emphasize the novelty and flexibility of our framework for the continuous covariate shift problem.
>
> ---
>
> **Q2:** the theoretical contribution seems to be a bit limited: the high probability result (instead of the expected), and the greatly simplified analysis.
>
> **A2:** We respectfully disagree with the comments and would like to take this opportunity to emphasize our contributions in the regret analysis part.
>
> - First of all, we would like to highlight that the performance measure we investigate serves as an intermediate case between worst-case dynamic regret and universal dynamic regret as detailed in lines 750-765, Appendix B.1. The intermediated regret receives **less attention in standard OCO literature**, but is very **importance for the continuous distribution shift problem.** Using the previous result on universal dynamic regret [19] in a black-box way can only imply an expected bound. In contrast, We contribute to provide a high-probability bound with greatly simplified analysis (without involving KKT conditions) for this less-explored yet practically significant intermediate setting.
> - Furthermore, our high-probability bound is obtained **non-trivially**. As outlined in lines 766-783, Appendix B.1, we strategically modify the Hedge algorithm into Adapt-ML-Prod. This alteration enables us to effectively control the generalization gap between $\hat{L}_t$ and $\tilde{L}_t$ by leveraging the negative term introduced by the exp-concavity of the loss function. Additionally, while the primary focus of this paper is on the continuous covariate shift problem, our simplified analysis is also applicable for general OCO purposes when the minimizers lie in the interior of the decision set. This approach can be of independent interest for other online learning problems.
>
> Due to page limits, we have to place much of the content in the appendix (especially the dynamic regret discussion in Appendix B.1). However, we greatly value your comments and will carefully revise the paper, making sure to emphasize those points more clearly in the main text.
>
> ---
>
> **Q3**: It would be nice to see different shift change patterns and how they affect the learning results. But it seems the errors are averaged before showing? Figure 2 only shows different methods but covers 4 kinds of shifts?
>
> **A3**: We have detailed the performance of all contenders across four different types of distribution shifts in Tables 1, 2, and 3. The results show that our algorithm can adapt to different kinds of shifts. Regrettably, due to space limitations, we have to place these results in Appendix A.1 for this version. In the next version, we will include some of these empirical findings within the main text. We believe this will be quite feasible, especially considering that an additional page is permitted in the camera-ready version. Thank you!

---

### Author Rebuttal · Authors · 2023-08-09


We sincerely appreciate insightful comments and the positive feedback from all reviewers for this paper. In the rebuttal period, we conducted additional experiments to further support our claim (particularly to address the concerns from Review Ls7D), as presented in the attached PDF file.  The experimental setup and results are listed as follows. To provide a better readability of the results, we have also included the text in the PDF files. Thanks!

---

**Experiment Setup for Figure A.** In response to Q4 (Reviewer Ls7D), we conduct weight assignment experiments in the case where the interval lengths of ensemble members **do not match** the period of distribution shift. The experimental setup and performance measure is the same as that in Figure 3 except that the distribution shift period is changed to $M = 10, 50, 100, 200, 400, 800$.

**Experiment Result for Figure A.** Figure A shows that our meta-algorithm can still assign larger weight to the "right'' ensemble members whose interval length is close to the distribution shift period ($M$) even they are not exactly matched.

---

**Experiment Setup for Table A.** In response to Q5 (Reviewer Ls7D), we provide the performance comparison on the synthetic dataset. The experimental setup and measure is the same as that in Figure 1 except we additionally report the performance of Accous when the interval length of the ensemble members are increased by the factor of 3 and 5, i.e., $\vert I_k\vert = 3^k$ and $\vert I_k\vert = 5^k$, respectively.

**Experiment Result for Table A.** Table A shows the averaged error over $T=10000$ iterations. The results show that our method is generally robust to the interval lengths of ensemble members.

---

**Experiment Setup for Figure B.** In response to Q8 (Reviewer Ls7D), we empirically plot the order of the averaged excess risk suffered by our algorithm and compare it with the result established in Theorem 3. We conduct experiments on the synthetic dataset with the $\texttt{Lin}$ shift ($V_t = \Theta(1)$). To empirically measure the order of the averaged excess risk, we introduce the **"risk ratio''** $\rho_t(p)$, defined as $\rho_t(p) = \mathfrak{R}_t\times t^{p}$ for each time $t$, where $p\in(0,1)$ is a constant, and $\mathfrak{R}_t$ is the averaged excess risk at $t$. If the risk ratio of an algorithm decreases over the horizon, it means the convergence rate of risk is at least $O(T^{-p})$. Otherwise, the risk order is slower than $\Theta(T^{-p})$.

**Experiment Result for Figure B.** Figure B shows the risk ratio of our algorithm. As suggested by Theorem 3, the excess risk of our algorithm should converge at least at the rate of $O(T^{-1/3})$ for the $\texttt{Lin}$ shift. The green line shows that the risk ratio $\rho_t(1/3) = \mathfrak{R}_t \times T^{1/3}$ decreases in the $\texttt{Lin}$ shift scenario, indicating that the algorithm empirically attains a convergence rate faster than $O(T^{-1/3})$, which is consistent with our theory.

We further note that the risk ratio $\rho_t({7}/{12})$ (red line) also slightly decreases along the horizon, indicating that the algorithm converges slightly faster than $O(T^{-7/12})$ empirically, thus quicker than $O(T^{-1/3})$. However, this result does not imply that our bound is loose. Theorem 3 essentially provides the **worst-case guarantee**: our algorithm is guaranteed to attain the $O(T^{-1/3})$ rate in any pattern of distribution shift when $V_T = \Theta(1)$. But, it is also possible to achieve a better convergence rate in benign environments. In the online learning literature, a bound that can adapt to benign environments is known as a problem-dependent bound, which requires more advanced techniques to attain. In this paper, we primarily consider the worst-case guarantee and will address how to achieve the problem-dependent bound in future work.

---

### Decision · Program_Chairs · 2023-09-21

**Decision:**

Accept (poster)

**Comment:**

This paper studies the problem of online learning in the presence of continuous covariance shifts. This paper was globally well received. Reviewer ZW3t had some concerns about how impactful the theoretical analysis was. The other reviewers mostly had questions about the experiments, concerns that were answered in the rebuttal. While I agree with reviewer ZW3t that some of the theoretical derivations are not the most impactful ones, I think that this work passes the bar for publication at Neurips. It is the first work providing a method for handling continuous covariance shifts with sound theoretical guarantees. The paper is well written, the work is novel and correct.